# Urea use drives niche separation between dominant marine ammonia oxidizing archaea

Joerdis Stuehrenberg [1,9], Katharina Kitzinger [1,2,9], Jan N. von Arx [1], Jon S. Graf[1], Gaute Lavik[1], Sten Littmann[1], Jana Milucka [1], William D. Orsi [3,4], Sina Schorn [1,7], Daan R. Speth [1,2], Aurèle Vuillemin [3,8], Siqi Wu [1,5], Hannah K. Marchant [1,6] ✉ & Marcel M. M. Kuypers [1]

Ammonia-oxidizing archaea (AOA) are among the most abundant microorganisms in the ocean and play a critical role in marine nitrogen cycling. Recently, urea has been shown to serve as an additional substrate for marine AOA, with substantial urea use in the ammonium-depleted open-ocean. Yet, the mechanisms that control urea use and potentially maintain high AOA abundances remain unclear. Here, we investigate urea and ammonia use by AOA in three contrasting marine environments, from coastal, ammonium-rich to open-ocean, ammonium-poor waters. Our combined results indicate that distinct substrate utilization strategies of *Nitrosopumilus* and *Nitrosopelagicus* control their environmental distribution. The more coastal AOA genus, *Nitrosopumilus*, primarily uses ammonium. In contrast, enhanced urea utilization in ammonium-limited waters is linked to the activity and growth of *Nitrosopelagicus*. Thus, the use of urea, and potentially other organic-N compounds by *Nitrosopelagicus* plays a major role in fueling open-ocean nitrification and sustaining primary productivity in these vast regions.

The process of nitrification plays a critical role in marine nitrogen (N) cycling, as it converts ammonia, the most reduced form of inorganic N, into nitrate, the most oxidized and abundant form. By altering the balance between reduced and oxidized forms of inorganic N, nitrification controls the availability of the substrates required for N-removing processes and impacts the structure of phytoplankton communities[1,2]. Thus, nitrification is intimately linked to marine productivity.

In the oceans, nitrification occurs in a stepwise fashion, where first, ammonium is oxidized to nitrite, largely by ammonia oxidizing archaea (AOA)[3,4], after which, nitrite is further oxidized to nitrate by nitrite oxidizing bacteria (NOB). AOA are ubiquitous throughout the oceans, where they constitute around 20% of the total microbial community[5], even though ammonium is generally present at vanishingly low concentrations, particularly in the vast open ocean regions. Thus, the ability to use urea as an additional N and energy source to ammonium has been hypothesized to be a factor that drives the success and abundance of AOA in the oceans. Urea is ubiquitously available throughout the oceans[6–8], where it is released during remineralization of organic matter and as nitrogenous waste from both prokaryotes and eukaryotes, as well as being an intracellular

[1]Max Planck Institute for Marine Microbiology, Bremen, Germany. [2]Division of Microbial Ecology, Centre for Microbiology and Environmental Systems Science, University of Vienna, Vienna, Austria. [3]Department of Earth and Environmental Sciences, Palaeontology & Geobiology, Ludwig-Maximilian-University, Munich, Germany. [4]GeoBio-Center LMU, Ludwig-Maximilian-University, Munich, Germany. [5]State Key Laboratory of Marine Environmental Science, College of Ocean and Earth Sciences, Xiamen University, Xiamen, Fujian, China. [6]MARUM - Centre for Marine Environmental Sciences University of Bremen, Bremen, Germany. [7]Present address: Department of Marine Sciences, University of Gothenburg, Gothenburg, Sweden. [8]Present address: GFZ Helmholtz Centre for Geosciences Potsdam, Section Geomicrobiology, Potsdam, Germany. [9]These authors contributed equally: Joerdis Stuehrenberg, Katharina Kitzinger. ✉e-mail: hmarchan@mpi-bremen.de

metabolite[9,10]. Environmental AOA communities were first shown to directly use urea and cyanate, another dissolved organic nitrogen compound, as additional sources of ammonia for energy and growth in the coastal Gulf of Mexico[6]. In these nutrient rich waters, the AOA community used nitrogen from urea directly, via import into the cytoplasm and intracellular hydrolysis of urea with urease enzymes to ammonia and subsequently, oxidation via ammonia monooxygenase. This hydrolysis also releases carbon dioxide which may also be assimilated by AOA. Additionally, AOA in the Gulf of Mexico also utilized urea indirectly by cross feeding on ammonia released by other urease-containing community members. However, it appeared that the AOA community in the Gulf of Mexico, which is largely comprised of the AOA genus *Nitrosopumilus*, only supplemented a small fraction of their ammonia demand via this strategy[6].

Recent evidence has suggested that urea might be a more important substrate for nitrification in the open ocean, where AOA-affiliated *ureC*, the marker gene encoding the alpha subunit of urease, is prevalent[11] and highly expressed[12]. Moreover, rates of urea-derived oxidation (hereafter urea oxidation) can be in the same range as ammonia oxidation rates[13,14], and the bulk microbial community can assimilate urea to a similar extent as ammonium[11]. It has been hypothesized that this increase in the importance of urea as a substrate might be linked to the higher availability of urea compared to ammonium in oligotrophic ocean regions[6,13,15].

The relative increase in urea utilization in the open ocean coincides with differences in the biogeography of the two AOA genera that dominate ocean waters (*Nitrosopumilus* and *Nitrosopelagicus*). *Nitrosopumilus* are more abundant in coastal regions, while *Nitrosopelagicus* (which is commonly split into two ecotypes referred to as Water Column A and Water Column B[14,16,17]) are more abundant in open ocean waters[4,14,18,19]. The factors that drive this separation in the environment are currently unclear, as to date most comparisons between the two genera have been made at the genomic level, where, other than differences in phosphorus transporters[20], *Nitrosopumilus* and *Nitrosopelagicus* have similar characteristics. Members of both genera have small genomes (<2 Mb)[20–22], the same highly energy-efficient version of the 3-Hydroxypropionate/4-hydroxybutyrate (3-HP/4-HB) carbon fixation pathway[23–25], and the same enzymatic repertoire for ammonia oxidation and assimilation[20,26,27]. Additionally, culture experiments show that both *Nitrosopumilus* and *Nitrosopelagicus* isolates (*Nitrosopumilus ureiphilus*[28], *Nitrosopumilus piranensis*[29] and *Ca*. Nitrosopelagicus brevis U25[24]) can grow on urea that is hydrolyzed intracellularly to ammonia. So far, however, in situ comparisons between the urea and ammonia utilization strategies of both genera under environmentally relevant conditions are lacking, and the use of urea as an N-source for AOA growth has not been investigated in the open ocean.

In this study, we address whether differential substrate response to urea and ammonium availability drives the niche differentiation between AOA genera. We use a combination of ship-board isotope labeling experiments, single cell imaging, and -omics to compare overall urea and ammonia oxidation rates as well as the substrate-specific assimilation rates of *Nitrosopumilus* and *Nitrosopelagicus* in three contrasting marine environments; the Gulf of Mexico, the Angola Gyre and the Black Sea. The Gulf of Mexico is characterized by eutrophic conditions and high primary productivity, while the surface waters of the open ocean Angola Gyre are more oligotrophic and have low primary productivity. The Western basin of the Black Sea is mesotrophic and therefore lies between these two extremes. Moreover, in the Black Sea, there is a high flux of ammonium from deeper anoxic waters, which is consumed in the stratified water column, leading to sharp ammonium concentration gradients over depth. This leads to a situation where both *Nitrosopelagicus* and *Nitrosopumilus* coexist at a similar abundance, allowing us to test the environmental response of both *Nitrosopelagicus* and *Nitrosopumilus* to urea and ammonium in the same samples, without confounding changes in other environmental conditions.

## Results

### Increasing importance of urea oxidation in the open ocean

The productivity regimes of the Gulf of Mexico, Black Sea and Angola Gyre differed substantially, as indicated by the yearly averaged chlorophyll *a* concentrations in the three regions, which spanned from 0.06 to 47.6 mg m$^{-3}$ (Fig. 1a). In line with the different productivity regimes, we observed differences in ammonium concentrations during our expeditions to the three regions (Supplementary Data 1, and Fig. 1c). As expected, median ammonium concentrations were much higher in the eutrophic Gulf of Mexico than the oligotrophic surface waters of the Angola Gyre (320 nM-N and 11 nM-N, respectively; Fig. 1c, Supplementary Fig. 1, and Supplementary Data 1). Median ammonium concentrations in the Western basin of the Black Sea were also low (51 nM-N), but had a much broader range, from below detection in shallower waters up to 7.4 μM-N at the deepest sampling depth (Fig. 1c, Supplementary Figs. 1, and 2, and Supplementary Data 1). While chlorophyll *a* and ammonium concentrations differed by more than an order of magnitude, urea concentrations were much more comparable across the three study regions (median 138 nM-N, 52 nM-N, 130 nM-N in the Gulf of Mexico, the Angola Gyre and the Black Sea, respectively; Fig. 1d). Thus, the fraction of urea-N to combined ammonium-N + urea-N (i.e., the relative availability of urea-N) was two times higher in the Angola Gyre (0.69 ± 0.04 (s.e.)) and the Black Sea (0.67 ± 0.02 (s.e.)) compared to the Gulf of Mexico (0.33 ± 0.04 (s.e.)) (Fig. 1b), similar to previous observations of higher urea availability compared to ammonium in oligotrophic ocean regions[13].

To determine whether the difference in relative availability of urea-N and ammonium-N affected the extent to which ammonia oxidizers used urea as an energy source, we quantified ammonia and urea-derived oxidation rates after addition of $^{15}$N-labeled substrates (Supplementary Table 1 and Supplementary Note 1). As reported in Kitzinger et al. 2019 (ref. 6), ammonia oxidation rates in the Gulf of Mexico ranged between 80 and 2500 nM-N day$^{-1}$ (average 804 nM-N day$^{-1}$) (Supplementary Fig. 2, and Supplementary Data 1). In comparison, ammonia oxidation rates in the surface waters of the Angola Gyre were two orders of magnitude lower (from below detection to 1.98 nM-N day$^{-1}$) (Supplementary Fig. 2, and Supplementary Data 1). Rates in the Black Sea were also low (range 0.6 to 41.4 nM-N day$^{-1}$) compared to the Gulf of Mexico but spanned a broader range than the Angola Gyre (Supplementary Fig. 2, and Supplementary Data 1). Thus, ammonia oxidation rates decreased as expected along with chlorophyll *a* and ammonium availability. Linear and significant rates of urea oxidation were also observed at all three regions (Supplementary Fig. 2, and Supplementary Data 1) and showed the same trends as ammonia oxidation rates. However, the differences between sites were not as pronounced (Fig. 2b, and Supplementary Data 1). As such, in the Gulf of Mexico, urea oxidation only accounted for around ~3.5% (1.8 to 7.1%) of the total nitrification rate (ammonia plus urea oxidation), rising to ~12% (0.2 to 34.0%) in the Black Sea and to ~44% (2.4 to 90.0%) in the Angola Gyre, which is significantly higher than the Gulf of Mexico and the Black Sea (two-sided Mann–Whitney–Wilcoxon test; $W = 74$, $P$ value = 0.002; $W = 157$, $P$ value = 0.0009, for Gulf of Mexico and Black Sea, respectively) (Fig. 2a). Therefore, urea oxidation appeared to be of minor overall importance to total nitrification in the nutrient-rich Gulf of Mexico. In contrast, in the nutrient-deplete Angola Gyre, urea oxidation rates were equal to ammonia oxidation rates (two-sided Mann–Whitney–Wilcoxon test, $W = 14$, $P$ value = 0.83) and in some samples even exceeded ammonia oxidation. The Angola Gyre results support the few previous studies carried out in oligotrophic open ocean waters, which showed that urea oxidation rates can be in the same range as ammonia oxidation rates[13,14,30].

### Persistent open ocean urea oxidation in presence of ammonium

The initial ammonia and urea oxidation rate data indicated that the ammonia-oxidizing communities in the different regions have

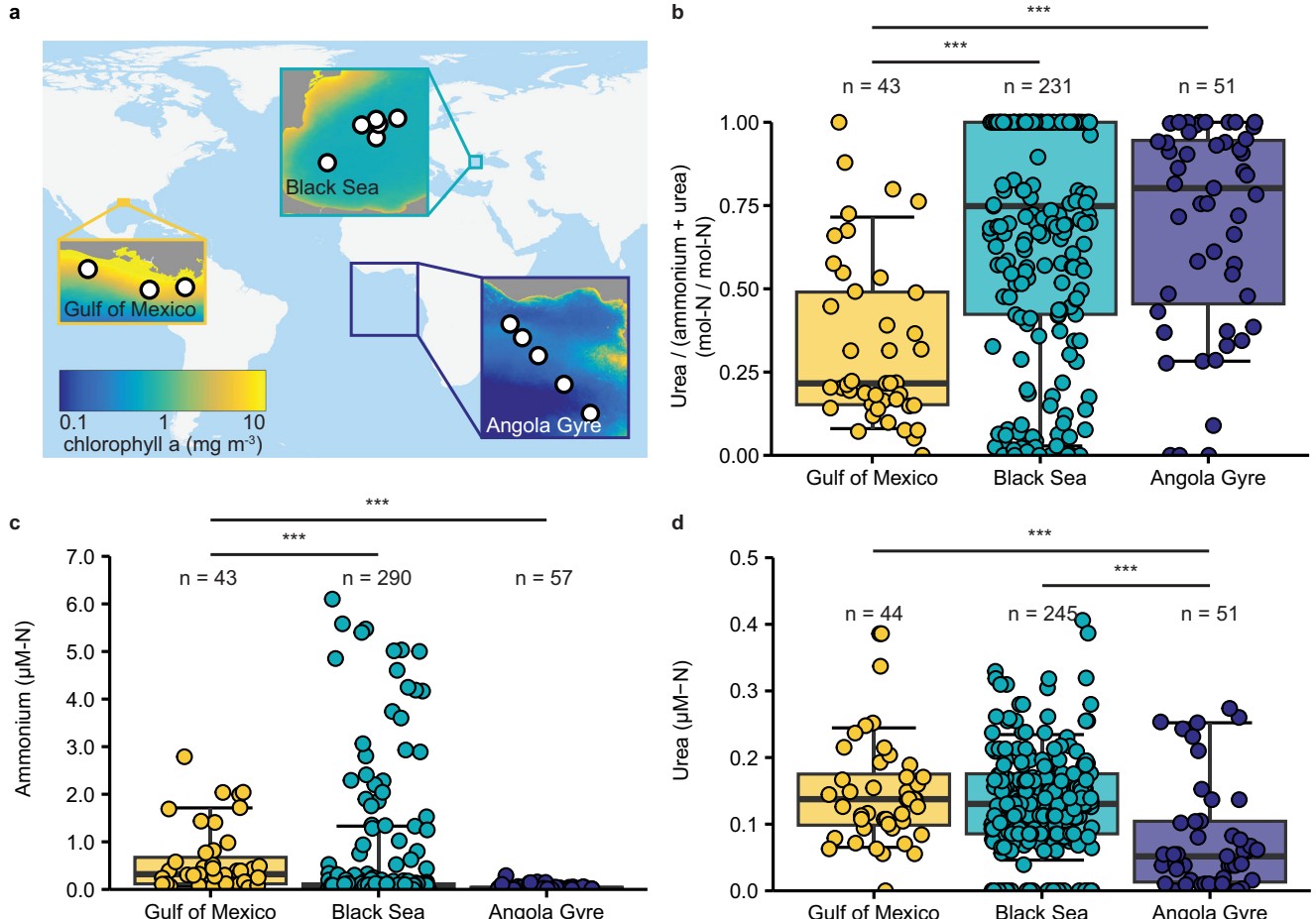

**Fig. 1 | Productivity regimes and relative importance of urea in the three different environments. a** World map with inserts showing the three different environments and respective surface chlorophyll *a* values as a proxy for productivity (averaged over the time period between January 1st, 2016 and December 31st, 2019[75], (https://doi.org/10.5067/AQUA/MODIS/L3M/CHL/2022). **b** The fraction of urea-N to combined ammonium-N + urea-N, concentrations of ammonium-N (**c**) and urea-N (**d**) in the upper water column of the Gulf of Mexico (<18.5 m, yellow), Black Sea (<145 m, turquoise) and Angola Gyre (<150 m, dark blue) (Supplementary Data 1). Boxplots depict the 25–75% quantile range, with the center line depicting the median (50% quantile); whiskers encompass data points within 10 to 90 percentiles. *n* = number of measurements per region. Four urea values from the Angola Gyre (0.59, 0.97 μM-N) and the Black Sea (1.42, 0.77 μM-N) are not depicted; but were included in all calculations. \*\*\* indicates significant difference between regions in two-sided

Mann–Whitney–Wilcoxon tests. The fraction of urea-N to combined ammonium-N plus urea-N in the Gulf of Mexico is significantly lower than in the Black Sea and the Angola Gyre ($W = 2462.5$, *P* value = $7.4 \times 10^{-8}$; $W = 413.5$, *P* value = $2.2 \times 10^{-7}$; respectively). The fraction of urea-N to combined ammonium-N + urea-N did not differ significantly between the Black Sea and Angola Gyre ($W = 6353$, *P* value = 0.4). Ammonium concentrations in the Gulf of Mexico were significantly higher than those in the Black Sea and the Angola Gyre ($W = 9876.5$, *P* value = $1.4 \times 10^{-10}$; $W = 413.5$, *P* value = $2.2 \times 10^{-7}$, respectively), where ammonium concentrations did not differ significantly ($W = 8647.5$, *P* value = 0.6). Urea concentrations in the Angola Gyre were significantly lower than in the Black Sea and the Gulf of Mexico ($W = 8886$, *P* value = $2.1 \times 10^{-6}$; $W = 1709$, *P* value = $1.2 \times 10^{-5}$, respectively), where urea concentrations did not differ significantly ($W = 5671$, *P* value = 0.6). All statistical parameters and source data are provided as a Source Data file.

different substrate preferences with regards to ammonia and urea. As urea must first be broken down intracellularly to ammonia in order to be oxidized via ammonia monooxygenase, we wanted to determine whether this preference for urea persisted when ammonium was replete, or whether the AOA would inherently prefer ammonia when it was available in the environment. Thus, we examined whether urea oxidation rates changed after a large background pool of ammonium was added (see also Supplementary Note 2 on differentiation between direct and indirect urea use). In the Gulf of Mexico, the already low urea oxidation rates decreased substantially in the presence of ammonium, indicating that the community preferentially used ammonia[6] (Fig. 2b). In contrast, in the Black Sea and Angola Gyre, there was no significant change in urea oxidation rate when ammonium was added (two-sided Mann–Whitney–Wilcoxon test, $W = 135$, *P* value = 0.56 and $W = 42$, *P* value = 0.93, respectively). This strongly suggested that urea-utilizing ammonia oxidizers were insensitive to short term (12–24 h)

increases in ammonium and continued to oxidize urea regardless of the external and in situ ammonium concentration (Fig. 2b). This seeming preference for urea is surprising considering that all known ammonia oxidizing archaea rely on the same enzymatic machinery for ammonia oxidation to nitrite[24,27,31] - even when the ammonia is derived from intracellular breakdown of urea. Yet, urea oxidation has also been shown to persist in the presence of excess ammonia in the oligotrophic Pacific[13]. Moreover, a similar preference for urea was recently observed for ammonia-oxidizing bacterial isolates of the class Betaproteobacteria, which were shown to have a higher affinity for urea than for ammonia and to repress ammonia transport in the presence of urea[32]. Taken together these results confirm previous findings[6,13,14] indicating that urea oxidation becomes an increasingly more important metabolism for marine ammonia oxidizers in oligotrophic regions where ammonium concentrations are low. However, to date, this relative increase in urea utilization has not been linked to potential differences in substrate

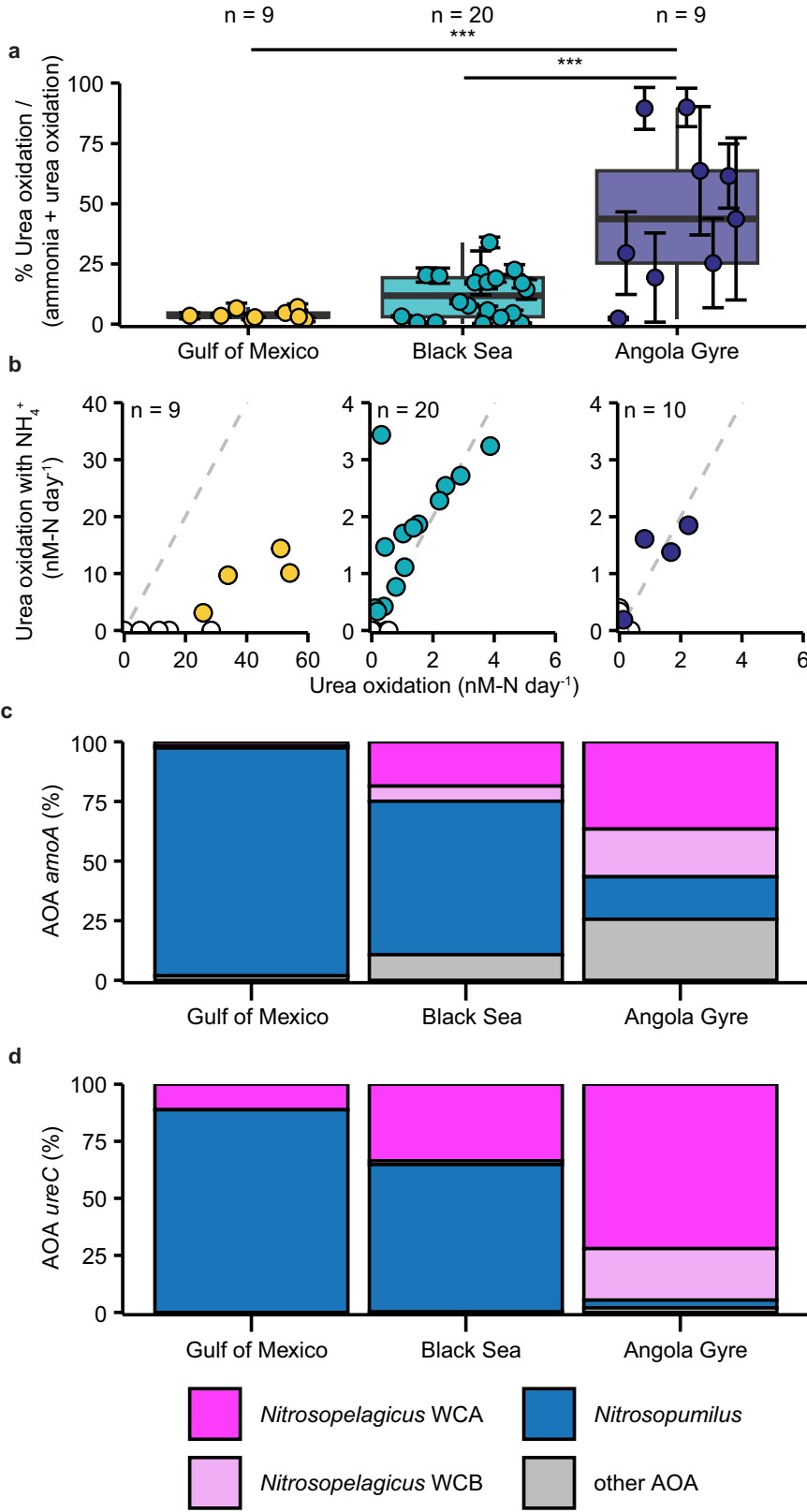

preference by the two dominant AOA genera *Nitrosopumilus* and *Nitrosopelagicus*.

## Genetic repertoire for urea utilization is enhanced in open ocean AOA

To identify the ammonia oxidizers responsible for urea utilization and gain further insights into the potential factors that lead to the different

substrate preferences in the three study regions, we assessed the composition and metabolic repertoire of the ammonia-oxidizing community. Analysis of small-subunit 16S rRNA genes and *amoA* (the gene encoding the alpha subunit of ammonia monooxygenase) in metagenomic datasets revealed that AOA dominated the ammonia-oxidizing community in all three regions (Supplementary Data 1 and 2). Overall however, there was a much lower abundance of ammonia

**Fig. 2 | Comparison between urea oxidation rates and key AOA ammonia and urea utilization genes in the three different environments. a** Urea oxidation rates as a percent of combined ammonia and urea oxidation rates in the Gulf of Mexico (yellow), the Black Sea (turquoise) and Angola Gyre (dark blue). Percentages (one value per depth) are calculated for all depths from the Gulf of Mexico and the Black Sea and for depths below 80 m in the Angola Gyre. Non-significant rates were set to the respective limit of detection. Error bars represent propagated uncertainties. Boxplots depict the 25–75% quantile range, with the center line depicting the median (50% quantile); whiskers encompass data points within 1.5× the interquartile range. *n* = number of measurements per region. *** indicates significant difference between regions in two-sided Mann–Whitney–Wilcoxon test. The fraction of urea oxidation rates to combined ammonium + urea oxidation rates in the Angola Gyre is significantly different from the Black Sea (*W* = 157, *P* value = 0.0009) and the Gulf of Mexico (*W* = 74, *P* value = 0.002). **b** Urea oxidation rates in

incubations without added ammonium compared to urea oxidation rates in incubations with added ammonium in the Gulf of Mexico (yellow), the Black Sea (turquoise) and Angola Gyre (dark blue). Samples where one urea oxidation rate was zero are filled white. Note the difference in scales between panels. Gray dashed lines show the 1:1 ratio. The difference between urea oxidation rates with or without added ammonium was not significant in the Black Sea and in the Angola Gyre (*W* = 135, *P* value = 0.56 and *W* = 42, *P* value = 0.93, two-sided Mann–Whitney–Wilcoxon test respectively). The urea oxidation rate in the Gulf of Mexico dropped significantly, when additional ammonium was added (*W* = 74, *P* value = 0.0032, two-sided Mann-Whitney-Wilcoxon test). Relative abundance (%) of AOA *amoA* (**c**) and *ureC* (**d**) assigned to *Nitrosopelagicus* (WCA magenta, WCB rose), *Nitrosopumilus* (blue) and other *Nitrososphaeria* (gray) across all metagenome samples in the three environments. All statistical parameters and source data are provided as a Source Data file.

oxidizers in the Angola Gyre ($-0.1 \times 10^5$ cells mL$^{-1}$) and Black Sea ($-0.3 \times 10^5$ cells mL$^{-1}$) than in the Gulf of Mexico ($4.9 \times 10^5$ cells mL$^{-1}$), which is generally consistent with the differences in total oxidation rates (ammonia + urea oxidation) between the three regions (Supplementary Data 1, and Supplementary Note 3). However, the one order of magnitude difference in AOA abundance did not match the two order of magnitude difference in total oxidation rates, indicating differences in per cell oxidation rates across environments. Therefore, we examined whether the difference in rates and substrate preference might be connected to the composition and metabolic repertoire of the AOA community. Like cell abundance, the composition of the AOA community differed between the three regions. The genus *Nitrosopumilus* strongly dominated the AOA community in the coastal Gulf of Mexico (-95% of all AOA), however, its relative abundance dropped in the less productive regions to -65% in the Black Sea and -18% in the Angola Gyre (Fig. 2c). As the relative abundance of *Nitrosopumilus* decreased, the relative abundance of *Nitrosopelagicus* from both the WCA and WCB ecotypes increased (Fig. 2c and Supplementary Methods), matching previously reported distributions of marine AOA (e.g., refs. [14,19]). Additionally, we also observed a decrease of *Nitrosopelagicus* abundance with depth in the Black Sea, which coincided with an increase in ammonium concentrations (Supplementary Fig. 2).

Subsequently, we analyzed the metagenomic and metatranscriptomic datasets for the key urea utilization genes *ureC* (encoding for the catalytic subunit of urease) and *dur3* (encoding the most widespread urea transporter in AOA[33]). In the Gulf of Mexico, most *ureC* in metagenomes and metatranscriptomes were affiliated with microorganisms other than AOA and overall transcription of *ureC* in this ammonium-rich coastal environment was low, indicating that urea was not used as a major N-source (Supplementary Fig. 5b). In contrast, in the Black Sea and the Angola Gyre, AOA-associated *ureC* dominated, and was also highly transcribed (Supplementary Fig. 5b). In all three environments, *Nitrosopumilus* and *Nitrosopelagicus* were the main genera encoding and transcribing AOA-associated *ureC* and *dur3* (Fig. 2d, Supplementary Figs. 4 and 5, and Supplementary Data 2). In the Gulf of Mexico, analogous to the community composition, most *ureC* and *dur3* were encoded and transcribed by *Nitrosopumilus*, however, there was a very low ratio of both *ureC* and *dur3* to the single copy marker gene *amoA* in the metagenomes (on average -0.1:1 for both), suggesting that only a small subset of the *Nitrosopumilus* population (-10%) could use urea (Fig. 2d, and Supplementary Figs. 4 and 5). In the Black Sea, more *Nitrosopumilus* appeared to encode *ureC* and *dur3* (on average 0.6:1 and 0.4:1, respectively). In contrast, all *Nitrosopelagicus* in the Black Sea encoded for *ureC* and *dur3* (on average 1.1:1 and 1.1:1, respectively) (Supplementary Fig. 5). However, the overall distribution of *ureC* and *dur3* genes between *Nitrosopumilus* and *Nitrosopelagicus* reflected the community composition i.e., two thirds were associated with *Nitrosopumilus* and one third with *Nitrosopelagicus* (Fig. 2d, and Supplementary Fig. 5). In the Angola Gyre, *ureC* and *dur3* were mainly encoded and transcribed by

*Nitrosopelagicus* (Fig. 2d, and Supplementary Figs. 4 and 5). Furthermore, in the Angola Gyre metagenomes, *Nitrosopelagicus ureC:amoA* ratio was around 0.7:1 and the *dur3:amoA* gene ratio was even higher at 1:1, suggesting that most *Nitrosopelagicus* encoded *ureC*, and all encoded *dur3* (Supplementary Fig. 5). Taken together, these results show that urea utilization genes are more prevalent in *Nitrosopelagicus* compared to *Nitrosopumilus* in both the strongly nutrient-deplete Angola Gyre and the mesotrophic Black Sea. This suggests that despite the presence of both genera in the Black Sea, the *Nitrosopelagicus* population might be more adapted to utilize urea as energy and N-source. However, as both genera transcribed urea utilization genes, these results alone could not unequivocally link urea use to a specific genus.

## Different N-assimilation strategies of coastal and open ocean AOA

In addition to utilizing urea as an energy source, AOA may also use it as an N-source for assimilation. To assess whether the *Nitrosopumilus* and *Nitrosopelagicus* communities differed in their use of ammonium and urea as an N-source, nanoscale secondary-ion mass spectrometry (NanoSIMS) was used to determine the assimilation of $^{15}$N-labeled ammonium and urea into single cells (Fig. 3, and Supplementary Fig. 6), which, unlike bulk measurements, enables substrate utilization to be directly linked to specific AOA genera. We focused our analysis on the Black Sea, as we were unable to visualize the AOA in the Angola Gyre using catalyzed reporter deposition fluorescence in situ hybridization (CARD-FISH), most likely due to their small cell sizes (Supplementary Note 3). Moreover, the mixture of *Nitrosopumilus* and *Nitrosopelagicus* in the Black Sea offered the unique possibility to directly compare these dominant AOA groups in the same samples without additional confounding factors due to differences in environmental conditions. To distinguish between the two groups, we designed specific 16S rRNA-targeted probes to target the *Nitrosopumilus* and the more abundant water column A (WCA) ecotype of *Nitrosopelagicus* using CARD-FISH (Supplementary Methods, Supplementary Data 3 and 4, and Supplementary Table 2). The new probes allowed the two populations to be visualized separately and showed distribution patterns of *Nitrosopumilus* and of *Nitrosopelagicus* that were consistent with the metagenomic results (Supplementary Data 1 and 2).

The single-cell N-assimilation data showed that almost all measured *Nitrosopumilus* cells from the Black Sea (17 out of 18) assimilated $^{15}$N-ammonium (average $3.0 \pm 1.9$ (SD) at% excess), equivalent to an N-based growth rate of 0.04 day$^{-1}$ (Fig. 3, and Supplementary Data 5). In contrast to *Nitrosopumilus*, not all of the *Nitrosopelagicus* cells from the Black Sea assimilated ammonium (10 out of 27, average $0.13 \pm 0.55$ (SD) at% excess) (Fig. 3, and Supplementary Data 5). This is equivalent to an average N-based growth rate of 0.002 day$^{-1}$, which is 24 times lower than that of the co-existing *Nitrosopumilus* and 7 times lower than that of other microorganisms in the Black Sea (Fig. 3). The starkly

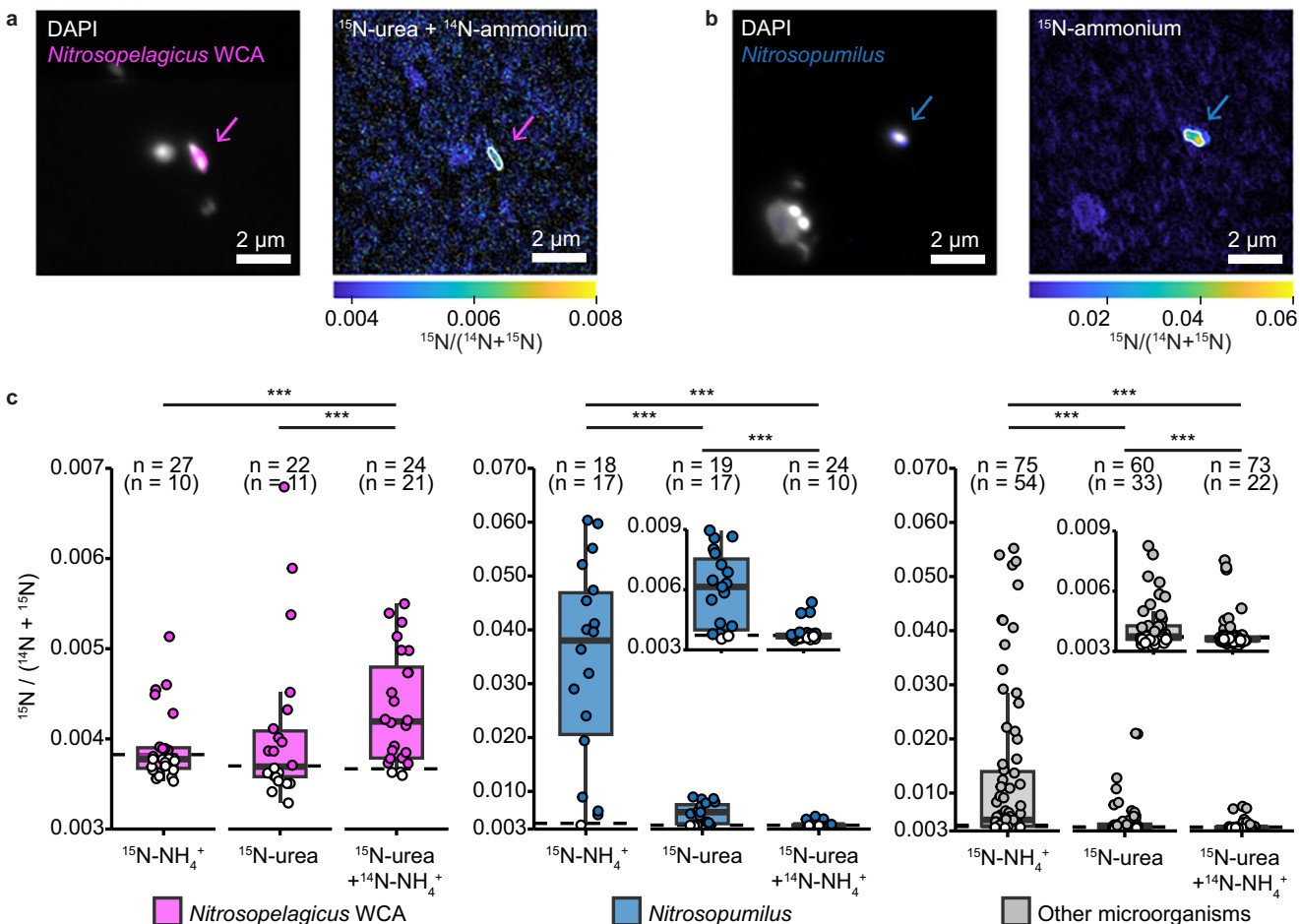

**Fig. 3 | Ammonium and urea assimilation by *Nitrosopelagicus*, *Nitrosopumilus* and other microorganisms.** Representative CARD-FISH images from the Black Sea (Station S3, 104 m) of **a** *Nitrosopelagicus* WCA (targeted by the specific probe Npe_WCA_226 in pink; counterstained by DAPI, white) and corresponding Nano-SIMS image of $^{15}N/(^{14}N + ^{15}N)$ enrichment after addition of $^{15}N$-urea + $^{14}N$-ammonium and **b** *Nitrosopumilus* (targeted by the specific probe Npum_229 in blue; counterstained by DAPI, white) and corresponding NanoSIMS image of $^{15}N/(^{14}N + ^{15}N)$ enrichment after addition of $^{15}N$-ammonium. Target cells are marked by white outlines. Scale bar, 2 μm. Note the different scales for $^{15}N/(^{14}N + ^{15}N)$ enrichment in **a**, **b**. **c** $^{15}N/(^{14}N + ^{15}N)$ enrichment of *Nitrosopelagicus* WCA (magenta), *Nitrosopumilus* (blue) and other microorganisms (gray) after incubation with $^{15}N$-ammonium and urea (+$^{14}N$-ammonium). Boxplots depict the 25–75% quantile range, with the center line depicting the median (50% quantile); whiskers encompass data points within 1.5× the interquartile range. Dashed lines represent the limit of detection. Cells that were below the limit of detection are filled white. Natural abundance of $^{15}N/(^{14}N + ^{15}N)$ is $3.7 \times 10^{-3}$. $^{15}N$ at% enrichment of the substrate pools (ammonium or urea, respectively) was >97%. One value for *Nitrosopelagicus* on ammonium is not depicted ($3.3 \times 10^{-2}$) but was included in all calculations. Note the different y-axis scales. $n$ = number of measurements per group and treatment with number of cells enriched significantly above the detection limit in brackets. *** indicates significant difference between treatments in two-sided Mann–Whitney–Wilcoxon tests. *Nitrosopelagicus* had similar enrichment values from $^{15}N$-ammonium and $^{15}N$-urea ($W = 335$, $P$ value = 0.45). $^{15}N/(^{14}N + ^{15}N)$ enrichment of *Nitrosopelagicus* on $^{15}N$-urea vs $^{15}N$-urea + ammonium was similar, but significantly different ($W = 156$, $P$ value = 0.017). *Nitrosopumilus* and other microorganisms were significantly higher enriched after incubation with $^{15}N$-ammonium compared to $^{15}N$-urea ($W = 305$, $P$ value = $1.2 \times 10^{-5}$ and $W = 3337$, $P$ value = $1.5 \times 10^{-6}$, respectively). Their enrichment dropped significantly, when an additional pool of $^{14}N$-ammonium was present ($W = 397$, $P$ value = $3.8 \times 10^{-5}$ and $W = 2990.5$, $P$ value = $3.0 \times 10^{-4}$, respectively). All statistical parameters and source data are provided as a Source Data file.

different ammonium-based growth rates of the co-occurring *Nitrosopumilus* and the *Nitrosopelagicus* in the Black Sea provided a first indication that the two groups might have different metabolic strategies for N-assimilation.

Incubations with $^{15}N$-urea revealed that both *Nitrosopumilus* and *Nitrosopelagicus*, assimilated urea, however they appeared to have different preferences for this substrate in comparison to ammonium (Fig. 3, and Supplementary Fig. 6). In both the Gulf of Mexico and the Black Sea, *Nitrosopumilus* assimilated very little urea-N compared to ammonium-N (Fig. 3c, and Supplementary Fig. 6). Furthermore, when *Nitrosopumilus* cells were incubated with an additional ammonium pool, the already low urea assimilation rates dropped significantly (Fig. 3c) (two-sided Mann–Whitney–Wilcoxon test, $W = 397$, $p$-value = $3.8 \times 10^{-5}$). These results indicate that although *Nitrosopumilus* can assimilate urea and grow using it, this metabolism is only of minor

importance, particularly when ammonium is available. Similar results, that urea can be used, but ammonium is the preferred substrate were observed for the other microorganisms in the sample (Fig. 3c). In contrast to *Nitrosopumilus*, *Nitrosopelagicus* assimilated ammonium and urea to a similar extent in the Black Sea (Fig. 3c). Moreover, the urea assimilation persisted even when additional ammonium was added (Fig. 3c), and *Nitrosopelagicus* cells showed significantly higher $^{15}N$-enrichment from urea than *Nitrosopumilus* in these incubations (two-sided Mann–Whitney–Wilcoxon test, $W = 452$, $P$ value = $7.5 \times 10^{-4}$). Substrate assimilation and oxidation seem to be tightly linked in ammonia oxidizer pure cultures[32]. Therefore, it can be assumed that the observed assimilation patterns directly translate to substrate oxidation, indicating that *Nitrosopelagicus* is responsible for the unchanged urea oxidation rates in presence and absence of ammonium. Together, these results indicate that although

*Nitrosopelagicus* has lower ammonium-based growth rates than *Nitrosopumilus*, it is metabolically more versatile, utilizing ammonium and urea equally. Furthermore, although we cannot unequivocally show that *Nitrosopelagicus* cells were assimilating ammonium and urea simultaneously, if they do, then this strategy could potentially lead to a doubling of their growth rate when both ammonium and urea are available.

## Discussion

By comparing the urea-utilization strategies of the AOA in the Gulf of Mexico, Black Sea and Angola Gyre, we found that as the fraction of urea to combined ammonium plus urea increased, the proportion of urea oxidation to the total nitrification rates increased, in line with previous observations[13,14]. Furthermore, in oligotrophic waters, urea-driven nitrification occurred at a similar rate regardless of whether additional ammonium was provided (Fig. 2b). These high relative rates of urea oxidation further highlight that approaches which base nitrification rates solely on ammonia oxidation might strongly underestimate open ocean nitrification[13,34].

The increasing importance of urea relative to ammonium as a substrate for AOA was accompanied by an increasing abundance of the genus *Nitrosopelagicus* compared to *Nitrosopumilus*. Although both *Nitrosopumilus* and *Nitrosopelagicus* encoded for and transcribed urease utilization genes, the *Nitrosopelagicus* population displayed a higher prevalence and transcription of these genes. This might suggest that urea utilization is a key factor that allows *Nitrosopelagicus* to maintain high cell abundances in low productivity, ammonium-deplete waters. In fact, by comparing the urea and ammonium assimilation strategies of environmental populations of *Nitrosopumilus* and *Nitrosopelagicus*, we could unequivocally show that the two groups of AOA appear to have very different metabolic strategies for N-utilization. Regardless of the in situ urea-N to ammonium-N ratios, environmental *Nitrosopumilus* cells had high growth rates on ammonium and only supplemented a small fraction of cellular N-demand with urea. In contrast, *Nitrosopelagicus* had much lower overall growth rates, but showed no preference for ammonium or urea. Instead, the *Nitrosopelagicus* population seemed to use both substrates simultaneously, which would potentially allow them to double their population growth rates (Fig. 3). Thus, the different metabolic strategies for N-utilization by *Nitrosopumilus* and *Nitrosopelagicus* may drive the niche differentiation between these two globally important groups of AOA.

Overall, our results show that urea can be equally important as a substrate as ammonia for nitrification by AOA in the open oceans, and reveal that this urea use is particularly associated with the activity and growth of *Nitrosopelagicus*. In this study, we focused on the importance of urea as a substrate for AOA in the open oceans. However, urea is only a minor component of the marine dissolved organic nitrogen (DON) pool, which is comprised of thousands of diverse, but poorly characterized compounds that together represent the largest reservoir of fixed nitrogen in oligotrophic waters[35]. Based on our results, and growing evidence that AOA might be able to use other organic nitrogen compounds such as amino acids and polyamines[36,37], we hypothesize that *Nitrosopelagicus* also obtains ammonia from other DON compounds in the open ocean, at the same time as urea, to further alleviate their substrate limitation. Considering that *Nitrosopelagicus* are among the most abundant microorganisms in open ocean waters, they might therefore exert a considerable control on the wider dissolved organic matter pool by selectively removing nitrogen-containing compounds. The resulting alteration of the DOM pool could be a key factor leading to the formation of carbon-rich refractory dissolved organic matter in the open oceans.

## Methods
### Sampling sites
Sampling permits for the Black Sea were issued by the Romanian authorities on June 12th, 2019 under the number A-2886/12.06.2019

and by the Bulgarian authorities on July 11th, 2019 under the number G5-1/11590. For the Angola Gyre no sampling permit was needed as these are international waters.

The Western Basin of the Black Sea (between 42° 27.05′N 29°15′W and 44° 12.8′N 30°59.6′W) was sampled during cruise POS539 with the R/V Poseidon, from November 6th to November 21st 2019. Seawater above the sulfidic layer was collected with a pump cast conductivity temperature depth system (pCTD) equipped with sensors for temperature and salinity (Seabird SBE49 FastCAT), fluorescence (Turner Designs Cyclops-7F), high- (Pyroscience FSO2-Subport) and low-sensitivity oxygen concentrations (Aanderaa optode 4831). Water was sampled down to 150 m. Depth profile measurements for nutrients were taken at six stations (W1, W2, W3, S2, S3, P1). Stable isotope incubations were carried out at all six stations, and molecular and FISH analyzes were done at two stations (S3, W3).

The Angola Gyre was sampled during cruise M148-2 with the R/V Meteor from July 2nd to 20th 2018. Seawater was collected using a CTD rosette equipped with 21 × 10 L Niskin bottles at all stations (station 210–215). Samples for nutrient concentration measurements were taken at all stations (doi: 10.1594/PANGAEA.931090) from profiling CTD casts. Samples for rate measurements and molecular analyzes were collected from a separate CTD cast from three to four depths from the ocean surface down to 150 m at stations in the open ocean (Angola Gyre) transect (stations 210, 211, 213, 214, 215).

The northern Gulf of Mexico was sampled during cruise PE17-02 with the R/V Pelican, from July 23rd to August 1st, 2016. Stable isotope incubations were carried out at three stations á three depths, and are described in detail in Kitzinger et al. 2019 (ref. 6).

Detailed information on sampling sites can be found in Supplementary Data 1.

### Nutrient analyses and sampling for CARD-FISH
Ammonium concentrations were determined immediately after water collection onboard in unfiltered samples using the fluorometric orthophthaldialdehyde method (limit of detection (LOD) in a 1 cm cuvette 1 nM for Angola Gyre, 42.7 nM for Black Sea cruises)[38]. Concentrations of nitrite (LOD 0.05 μM), nitrate (obtained by subtracting nitrite from combined nitrate and nitrite concentrations, LOD 0.24 μM) were measured either onboard using unfiltered seawater samples (Angola Gyre cruise), or using filtered and frozen samples (Black Sea cruise) spectrophotometrically using the Griess assay as previously described[39] using a QuAAtro39 autoanalyzer (Seal Analytical). Urea concentrations were determined spectrophotometrically in a 5 cm flow-through cuvette (Genesys 6, Thermo Fisher Scientific) using the colorimetric diazetylmonoxime method (Black Sea according to ref. 40 and Angola Gyre according to ref. 41) and had a LOD of 27.6 nM for the Black Sea and 40 nM for the Angola Gyre cruises. For the Gulf of Mexico cruise, nutrient analyses are described in Kitzinger et al. 2019 (ref. 6) (Supplementary Data 1).

For cell counts and catalyzed reporter deposition fluorescence in situ hybridization (CARD-FISH), seawater (50–100 mL for Black Sea; 45–90 mL for Angola Gyre cruises) was fixed with 1% paraformaldehyde (without methanol, EMS) for 12 to 24 h at 4 °C before filtration (<400 mbar) onto 0.22 μm GTTP filters (Millipore) and washing with sterile filtered seawater. Filters were stored frozen at −20 °C until analyzes.

### Process rate experiments
For rate experiments in the Black Sea, seawater incubations were done for three to four depths per station. Incubations were set up headspace-free in 250 mL glass serum bottles or 500 mL Schott bottles (for urea incubations with added ammonium pool) and in both cases closed with degassed rubber stoppers as described by refs. 42,43, and transferred to a light protected cooler until further processing (within 6 h). To minimize oxygen contamination, seawater was then bubbled

with a gentle helium gas stream for 20 min, amended with stable isotopes and aliquoted into 12 mL glass vials (Exetainers; Labco, UK). Ammonia and urea oxidation rates were assessed by using $^{15}$N-stable isotope labeled ammonium and urea, respectively. For ammonia and urea oxidation incubations, the two deepest depths were kept anoxic, while the upper two depths were amended with 10 μM oxygen (Supplementary Table 1). Additionally, we performed incubations where in addition to $^{15}$N-urea, a large unlabeled ($^{14}$N) ammonium pool was added (Supplementary Table 1). These incubations were carried out to assess (i) the potential contribution of biotic and abiotic urea-breakdown to ammonium during the incubations and subsequent use of the resulting extracellular $^{15}$N-ammonium by ammonia oxidizers (see Kitzinger, et al. ref. [6]) and (ii) to test whether urea was still oxidized in the presence of ammonium (Supplementary Table 1 for treatment overview and added concentrations) and (iii) to differentiate between direct and indirect urea utilization by ammonia oxidizers (see Supplementary Note 2). In all incubations, a glass vial each (i.e., one biological replicate incubation each) was sacrificed at five time points (after 0 h, 3 h, 6 h, 12 h, 24 h) by adding a glass bead and 100 μL of dilute HgCl$_2$ (0.7 g/100 mL). For urea incubations with added ammonium, the content of an additional glass vial per time point was sterile filtered and frozen at −80 °C for $^{15}$N-ammonium analyses (see below). After 24 h, 3 × 12 mL glass vials were pooled and the seawater fixed with 1% paraformaldehyde (12–24 h, 4 °C) and filtered onto 0.22 μm gold-sputtered GTTP filters for NanoSIMS analysis (see below).

In the Angola Gyre, for urea-derived oxidation rates both without and with additional added ammonium, 200 mL seawater each was distributed into 250 mL Schott bottles with an oxic headspace (oxygen concentrations at the sampled depths was >60 μM throughout, and thus were not specifically adjusted). $^{15}$N-tracer and $^{14}$N-pool additions were made according to Supplementary Table 1. Per depth and treatment, duplicate bottles were incubated (this was done due to time, water, and incubator space constraints). Four 12 mL subsamples were taken during the incubation (approximately at 0 h, 3 h, 6 h, 12 h), sterile filtered (0.2 μm PES) and frozen at −80 °C until further processing. Ammonia oxidation rate experiments were carried out in glass vials as described for the Black Sea, under ambient oxygen conditions.

In the Gulf of Mexico, the ammonia and urea oxidation rate experiments were conducted as described in Kitzinger et al. 2019 (ref. [6]). Briefly, incubations were performed in triplicate in serum bottles, with oxygen concentrations adjusted to match in situ concentrations. In all three environments, the amount of tracer addition was made taking into consideration the productivity regime, and was adjusted to expected in situ concentrations, aiming for >90% $^{15}$N at%, similar to previous studies[44–47] (Supplementary Data 1, and Supplementary Table 1).

### $^{15}$N-rate measurements

Potential ammonia and urea-derived oxidation rates were determined from the increase in $^{15}$N-nitrite over time after addition of $^{15}$N-ammonium or $^{15}$N-urea, respectively. For all time points, nitrite in 5 mL subsamples was converted to N$_2$O using acetic azide[48]. The resulting $^{45}$N$_2$O was quantified by gas chromatography isotope ratio mass spectrometry (GC-IRMS) with a customized TraceGas coupled to a multicollector IsoPrime100 (Isoprime, Manchester, UK) and the software Ion Vantage for Isoprime (Build1,5,6,0). In $^{15}$N-urea incubations with added $^{14}$N-ammonium, combined biotic and abiotic breakdown rates of urea to ammonium were measured for all time points in 5 mL subsamples from the increase in $^{15}$N-ammonium over time after $^{15}$N-urea addition. This was done by combining hypobromite oxidation of ammonium to nitrite, subsequent neutralization by HCl and reduction to N$_2$O using acetic azide according to ref. [49]. We did not remove the $^{15}$N-nitrite before the hypobromite conversion, and therefore measured combined $^{15}$N-ammonium + $^{15}$N-nitrite. $^{15}$N-ammonium values were obtained by subtracting the separately

measured $^{15}$N-nitrite values for each time point from the combined $^{15}$N-ammonium + $^{15}$N-nitrite values (Supplementary Methods equation 1). Breakdown of $^{15}$N-urea to $^{15}$N-ammonium was negligible in Angola Gyre and Black Sea samples, but was measurable in the Gulf of Mexico samples (see Kitzinger et al. 2019 ref. [6] for extended analyses of these rates) (Supplementary Data 1). In total, 324, 300 and 240 samples were measured using GC-IRMS from the Gulf of Mexico, the Black Sea and the Angola Gyre, respectively.

For the Black Sea and Angola Gyre cruises, initial ammonium $^{15}$N-labeling percentage in the incubations was determined from the in situ ammonium concentrations and the measured $^{15}$N-concentration after tracer addition by converting $^{15}$N-ammonium to N$_2$ using alkaline hypobromite[50] and subsequent GC-IRMS analysis. For the Black Sea cruise, the initial urea $^{15}$N-labeling percentage in the incubations was determined from in situ urea concentrations and measured total urea concentrations after tracer addition[40]. For the Angola Gyre cruise, initial urea $^{15}$N-labeling percentage was determined from in situ urea concentrations and the $^{15}$N-concentration excess above natural abundance after tracer addition by converting $^{15}$N-urea to N$_2$ using alkaline hypobromite[50].

Linear regression slopes were calculated over all time points (24 h for Black Sea and 12 h for Angola Gyre samples) and a one-sided Student T-test was applied to test whether they were significantly different from zero ($P$ value < 0.05). The slopes were corrected for the initial labeling percentage of the added $^{15}$N-substrate. Detection limits were calculated as described in Supplementary Methods (equation 2). The detection limits for ammonia oxidation, urea oxidation and urea oxidation (+NH$_4^+$) in the Angola Gyre were 0.017, 0.040, 0.146 nM day$^{-1}$, respectively. In the Black Sea detection limits for ammonia oxidation (+10 μM O$_2$), ammonia oxidation (no added O$_2$), urea oxidation and urea oxidation (+NH$_4^+$) were 0.15, 0.17, 0.06 and 0.17 nM day$^{-1}$, respectively. In the Black Sea, all significant slopes above the limit of detection were reported as significant process rates. As slopes in the Angola Gyre were very close to the limit of detection, when one of the duplicate incubations showed a significant slope, but the second duplicate not, these non-significant slopes were set to the limit of detection and then the process rates were calculated by averaging duplicates (Supplementary Fig. 2, and Supplementary Data 1). When both replicates had non-significant slopes, the rates were set to zero.

### DNA and RNA sampling, extraction and sequencing

For Black Sea and Angola Gyre metagenome and metatranscriptome analyses, ~4 L of seawater from selected incubation depths and stations (Black Sea Station S3, W3 and Angola Gyre Station 210, 214, 215) was filtered onto 0.22 μm cartridge filters (Sterivex, Millipore) using peristaltic pumps (Masterflex) immediately after sampling. Sterivex cartridges intended for RNA extraction were filled with RNAlater (Invitrogen), sealed and stored at −80 °C, residual seawater from Sterivex cartridges intended for DNA analyzes was expelled and cartridges were directly frozen at −20 °C.

Total DNA and RNA were extracted using the DNeasy and RNeasy PowerWater kits, respectively (Qiagen, Germany) according to the manufacturer instructions, including the recommended extra heating step for lysis of difficult-to-lyse organisms, for RNA extraction, we performed on-column DNase digestion. Concentrations of RNA and DNA in the extracts were 2.5 to 24 ng μL$^{-1}$ and 30 to 57 ng μL$^{-1}$, respectively, for Black Sea samples, below detection to 1.3 ng μL$^{-1}$ and 0.18 to 0.45 ng μL$^{-1}$, respectively, for Angola Gyre samples (Supplementary Data 2). DNA and RNA samples were stored at −80 °C until library preparation.

Metagenomes and metatranscriptomes from the Black Sea were sequenced (paired-end mode, 2 × 150 bps) using Illumina MiSeq at the Max Planck Sequencing Center in Cologne, Germany (Supplementary Data 2). Metagenomes and metatranscriptomes from the Angola Gyre were sequenced (paired end mode, 2 × 150 bps) using Illumina

NextSeq1000 except for the metatranscriptome sample from S214, 85 m, which was sequenced using Illumina MiniSeq (paired end mode, 2 × 150 bps) at the Department of Earth Sciences at Ludwig-Maximilian-University Munich as described in ref. 51 (Supplementary Data 2).

## Abundance of key genes in metagenomes and metatranscriptomes

Illumina adapter sequences were removed from quality-controlled paired-end reads with TrimGalore (v0.6.7)[52] with Cutadapt (v4.0 with Python v3.7.12)[53]. Subsequently, phyloFlash (v3.4, v3.4.1, for Black Sea and Angola Gyre, respectively; Silva database v138.1)[54,55] was run on the trimmed metagenomic reads to assess AOA abundance and community composition based on 16S rRNA gene reads in the metagenomes (Supplementary Data 1).

We assessed the abundance of key genes of interest (translated to amino acid sequences) in metagenomes and metatranscriptomes following ref. 56. Specifically, we assessed abundances of the alpha subunit of ammonia monooxygenase (amoA), the alpha subunit of urease (ureC) and the urea active transporter (urea sodium:solute symporter family (SSSF) transporter, dur3) frequently found in archaeal ammonia oxidizers e.g.,[24,57,58]. We first constructed a global database (compiled from GTDB[59] and GEM[60]) and custom gene databases (see Supplementary Methods).

Our trimmed metagenomic and metatranscriptomic reads were first compared to the compiled reference databases using DIAMOND blastx (v2.0.14)[61]. Reads with hits to the custom database (cutoff "minscore 10") were extracted and additionally compared to all GTDB/GEM proteins using DIAMOND blastx (v2.0.14). The BLAST score ratio (BSR) approach was used to remove false positive hits while keeping divergent sequences by calculating the BSR between a hit against our custom database and a hit against GTDB/GEM proteins. This ensured that sequences with low similarity to both GTDB/GEM proteins and our database were considered as true positives, whereas reads with a much better hit to the GTDB/GEM proteins than to our database were discarded as false positives. We used a minimum score of 100, percentage identity of 90 and a BSR of 0.9 as criteria for true positive hits. This threshold was selected based on a first manual inspection of reads with different BLAST score ratios (Supplementary Fig. 7b). From the extracted amoA read sets, we determined the fraction of reads pertaining to AOA (Nitrososphaeria), by creating a new database containing only the Nitrososphaeria reference dataset sequences, and following the same BSR approach as outlined above (first DIAMOND mapping against the complete AmoA/PmoA database, then all significant hits were mapped against the Nitrososphaeria subset) (Supplementary Fig. 7c). Scripts used for database construction and read searching are available from https://github.com/dspeth/aoa_urea/.

We defined the three marine AOA groups, Nitrosopumilus, Nitrosopelagicus WCA and Nitrosopelagicus WCB based on the consensus of the AOA trees based on 16S rRNA genes (Supplementary Fig. 8), genome (Supplementary Fig. 9) and amoA (Supplementary Fig. 10, Supplementary Methods). RPKM values as a proxy for abundance were calculated for each sequence in the AOA amoA, ureC and dur3 database in our metagenome and metatranscriptome (Supplementary Methods equation 3). The databases and reported RPKM values for the Black Sea, the Angola Gyre and the Gulf of Mexico can be found in Supplementary Data 2.

Additionally, we assessed the diversity of AOAs and the affiliation to the three groups (Nitrosopumilus, Nitrosopelagicus WCA and WCB) in the three environments based on phylogenetic analyses (amoA, UreC, genome trees) using the metagenome assemblies (Supplementary Figs. 3, 4, 9, Supplementary Methods, Supplementary Table 3).

## AOA visualization by CARD-FISH

We used CARD-FISH to visualize AOA in our fixed seawater samples. AOA visualization in the Gulf of Mexico is described in Kitzinger et al. 2019 (ref. 6). Note that the AOA community in the Gulf of Mexico is dominated by Nitrosopumilus, as such, almost all AOA targeted by general AOA oligonucleotide probes represent this group. As we were unable to perform CARD-FISH for AOA on Angola Gyre samples, most likely due to their small cell sizes in this open ocean region (Supplementary Note 3), we focused our CARD-FISH analyses on the Black Sea.

To differentiate between the co-occurring AOA groups of interest in the Black Sea samples, probes separately targeting Nitrosopumilus, Nitrosopelagicus WCA and Nitrosopelagicus WCB were designed in ARB[62] using the SILVA database v138.1 (Supplementary Data 4). The newly designed probes targeted 95% of the Nitrosopumilus diversity (Npum_229), 89% of the Nitrosopelagicus WCA diversity (Npe_WCA_226) and 95% of the Nitrosopelagicus WCB diversity (Npe_WCB_270) in the Black Sea samples (Supplementary Data 3). This was assessed by first retrieving all 16S rRNA gene reference sequences from the SILVA database which were identified as best hit in the phyloFlash analyses (v.3.4; Silva database v138.1, using the hitstats output)[54,55]. The reference sequences were separated into Nitrosopumilus, Nitrosopelagicus WCA, and WCB (Supplementary Data 4) and queried for perfect probe match using the reverse complement of the newly designed probes (blastn v. 2.6.0+, with –task option "blastn-short")[63]. We then calculated the percentage of reads mapping to group reference sequences with perfect matches to the probe, compared to the total number of reads mapping to group reference sequences. The newly designed probes and competitors (designed for sequences where differences in in silico-predicted formamide melting concentrations of target and non-target organisms were less than 20%[64] were ordered at biomers.net GmbH Ulm, Germany. A formamide concentration series was performed for each probe to determine the optimal hybridization conditions[65] using the Clone-FISH approach[66] for Nitrosopelagicus WCA and WCB probes, and using fixed Nitrosopumilus adriaticus cells for the newly designed Nitrosopumilus probe (Supplementary Fig. 6, Supplementary Methods, Supplementary Table 2).

For target cell quantification, the CARD-FISH protocol[67] using horseradish peroxidase (HRP)-labeled probes was applied onto filtered and fixed seawater samples (probe specifics in Supplementary Table 2). Cell immobilization was performed using 0.2% low-gelling agarose (except for samples prepared for NanoSIMS analysis). Endogenous peroxidases were inactivated in 0.01 M HCl for 10 min. Cell permeabilization was performed using 0.1 M HCl for 1 min, followed by two washing steps with ultrapure water (MilliQ). The hybridization with HRP-labeled probes and respective unlabeled competitors was done at the optimal formamide concentrations for each probe (Supplementary Table 2) at 46 °C for 2 to 3 h. A washing step at 48 °C for 5 min and HRP probe equilibration step in 1× PBS for 15 min was performed before signal amplification with OregonGreen488-labeled tyramides at 48 °C for 30 min. Residual amplification buffer was removed by washing in 48 °C 1× PBS and MilliQ. Cells were counterstained with 4',6-diamidino-2-phenylindole (DAPI, 10 μg mL⁻¹, 5 min at room temperature). Each CARD-FISH experiment included a positive control using probes EUB338 I-III[68,69] and negative controls with the probe NonEUB[70] on separate filter pieces. Counting of target cells was done using an epifluorescence microscope (Axioplan 2, Zeiss, Germany).

## NanoSIMS determination of ¹⁵N-enrichment

NanoSIMS analyses on the Gulf of Mexico samples are described in detail in Kitzinger et al. 2019 (ref. 6). For NanoSIMS analyses on samples from the Black Sea, we used samples from 104 m at station S3 where there were high theoretical per cell ammonia and urea oxidation rates of the whole AOA community, calculated from the measured bulk rates and AOA abundance assessed by CARD-FISH. NanoSIMS filters were marked with the laser microdissection microscope (LMD 6000 B, Leica) prior to CARD-FISH hybridization to enable correlative CARD-FISH and NanoSIMS imaging. CARD-FISH was conducted as described above, however, without the agarose embedding step. Previously, CARD-FISH was shown to dilute the isotopic composition of target and non-target cells depending on their growth stages[71–73]. In our

experiments, we did not account for isotopic dilution of AOA, as it is not well constrained for environmental samples. Thus, we potentially underestimate the $^{15}N$-enrichment of the AOA cells, and our estimates are conservative. The NanoSIMS filter which was amended with $^{15}N$-ammonium and hybridized for *Nitrosopelagicus* WCA was coated with an additional gold layer (7 nm at 30 mA current and a vacuum of $5 \times 10^{-2}$ mbar) with a sputter coater (ACE600, Leica), prior to Nano-SIMS measurement. The NanoSIMS instrument precision for detection of $^{15}N/^{14}N$ and $^{13}C/^{12}C$ isotope ratios was monitored regularly on Graphite Planchet. The prepared samples were pre-sputtered with a cesium (Cs+) primary ion beam of 300 pA at the NanoSIMS 50L (CAMECA). For measurement, the sample surface was rastered with a Cs$^+$ primary ion beam (current: 1.5 pA) with a dwelling time of 1 ms per pixel for 30 planes (raster area $10 \, \mu m \times 10 \, \mu m$) or 20 planes ($6 \, \mu m \times 6 \, \mu m$). The masses of $^{12}C^-$, $^{13}C^-$, $^{19}F^-$, $^{12}C^{14}N^-$, $^{12}C^{15}N^-$, $^{31}P^-$ and $^{32}S^-$ were measured simultaneously, and the instrument was tuned for a mass resolving power >8000. The nominal spot size was <100 nm and the image resolution was 256 px × 256 px. There were 27, 22 and 24 *Nitrosopelagicus* cells from the Black Sea analyzed from incubations with ammonium, urea and urea + ammonium, respectively. As well as 18, 19 and 24 *Nitrosopumilus* cells from the Black Sea from incubations with ammonium, urea and urea + ammonium, respectively. From the Gulf of Mexico there were 58 and 32 *Nitrosopumilus* cells from incubations with ammonium and urea, respectively, analyzed.

Subsequently, the NanoSIMS data were processed using the software Look@NanoSIMS (2023-03-16)[74] in MATLAB (R2021b) (The MathWorks Inc., 2021). The dead-time correction was applied, and the registration-based algorithm was used for aligning NanoSIMS and fluorescence images. Only measurements with a Poisson error <5% across all planes and a $^{15}N/(^{14}N + ^{15}N)$ background between $3.4 \times 10^{-3}$ and $4.0 \times 10^{-3}$ were used for further calculations. The limit of detection (Supplementary Methods equation 4) was $3.82 \times 10^{-3}$, $3.70 \times 10^{-3}$, $3.67 \times 10^{-3}$ $^{15}N/(^{14}N + ^{15}N)$ for *Nitrosopelagicus* on ammonium, urea and urea + ammonium, respectively. For *Nitrosopumilus* the LOD was $4.06 \times 10^{-3}$, $3.72 \times 10^{-3}$, $3.72 \times 10^{-3}$ $^{15}N/(^{14}N + ^{15}N)$ on ammonium, urea and urea + ammonium, respectively. The cellular $^{15}N$-atom% excess and growth rates were calculated as described in the Supplementary Methods equations 5 and 6. To enhance visualization of the NanoSIMS raw enrichment (Fig. 3a, b), we utilized the total counts of the $^{14}N$ signals to generate an alpha mask. This mask was then rescaled from the 1st to the 99th percentile and applied to the $^{15}N/(^{14}N + ^{15}N)$ enrichment ratio. Through this processing step, enrichment values associated with lower counts (i.e., stemming from the filter surface) are dimmed (but still visible), whereas those with higher counts (i.e., associated with biomass) are accentuated.

### Data analysis
Maps depicting sampling sites were generated using QGIS (v3.32.2). Averaged chlorophyll values were downloaded from Giovanni (v4.39)[75] and log10 transformed and visualized in QGIS (v3.32.2). Data was processed and visualized using MS Excel 2016 and RStudio (v2023.12.1) including packages dbplyr (v2.5.0), tidyr (v1.3.1) ggplot2 (v3.5.1), patchwork (v1.2.0), scales (v1.3.0). All statistical tests were done in RStudio.

### Reporting summary
Further information on research design is available in the Nature Portfolio Reporting Summary linked to this article.

### Data availability
All sequence data and AOA MAGs generated in this study have been deposited in NCBI under BioProject number: PRJNA1091553. Nutrient Data for the Angola Gyre cruise has been deposited at https://doi.org/10.1594/PANGAEA.931090; nutrient data for the Black Sea is given in Supplementary Data 1, for the Gulf of Mexico refer to Kitzinger et al.

2019 (ref. 6). Custom gene databases and RPKM values from the Gulf of Mexico, the Black Sea and the Angola Gyre are in Supplementary Data 2. Evaluation of target and non-target hits of newly designed CARD-FISH probes are in Supplementary Data 3. Accession numbers of sequences used for 16S rRNA, *amoA* and UreC trees are in Supplementary Data 4. Processed NanoSIMS data is provided in Supplementary Data 5 and raw data is available upon request. Source data are provided with this paper.

### Code availability
Scripts used for database construction and read searching are available from https://github.com/dspeth/aoa_urea/ (https://doi.org/10.5281/zenodo.17294067).

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

## Acknowledgements

The authors thank the Captain and the crew of R/V Meteor M148/2 EreBUS cruise, and the Captain and crew of the FS Poseidon POS539 cruise. We gratefully acknowledge the Meteor Leitstelle, the German National Science Foundation (DFG), and the Federal Ministry for Education and Research (BMBF) for the organization and financing of Meteor Expedition M148/2 and F/S Poseidon Expedition POS539. We thank T. Ferdelman for organizing and leading M148/2, G. Klockgether, K. Imhoff and N. Rujanski for nutrient and IRMS measurements. V. Mohrholz and S. Beier for CTD support, and D. Merino Benito, M. Philippi and W. Mohr for support during the cruise; S. Ahmerkamp and T. Priest for image processing advice and L. Bristow for support during the planning of the experiments. We thank L. Polerecky for providing additional functions in the look@nanoSIMS software. H.K.M. and J.S. received funding from the DFG under Germany's Excellence Strategy (no. EXC-2077-390741603). S.W. was supported by the Ph.D. Fellowship of the State Key Laboratory of Marine Environmental Science, Xiamen University and the China Scholarship Council. This study was funded by the Max Planck Society.

## Author contributions

G.L., H.K.M., J.M., J.N.v.A, J.S.G., K.K., S.S. carried out ship-board sampling and experiments. G.L., J.S., K.K. analyzed bulk rate data. A.V. and W.D.O. sequenced Angola Gyre samples. D.R.S., J.S., K.K. processed and analyzed metagenomic and metatranscriptomic data. J.S., K.K., S.W. designed and tested FISH probes. J.S. and S.L. conducted nanoSIMS analyses. H.K.M., K.K., and M.M.M.K. designed the study. H.K.M., J.S., K.K., and M.M.M.K. wrote the manuscript with contributions from all coauthors.

## Funding

## Competing interests

The authors declare no competing interests.
