## [Transparent Peer Review file · Nature Communications]

Urea use drives niche separation between dominant marine ammonia oxidizing archaea

Corresponding Author: Dr Hannah Marchant

Version 0:

Reviewer comments:

Reviewer #1

(Remarks to the Author)

Stuehrenberg et al. investigated the role of urea in sustaining AOA in coastal and open ocean regions, with a focus on niche differentiation between *Nitrosopumilus* and *Nitrosopelagicus*, two abundant genotypes. Using a combination of stable isotope labeling, metagenomics, metatranscriptomics, and NanoSIMS analyses, the authors show that urea oxidation becomes increasingly important in oligotrophic environments, particularly for *Nitrosopelagicus*. The results provide valuable insights into the metabolic flexibility of marine AOA and their nitrogen utilization strategies.

The manuscript is well written, with robust data supporting its conclusions. However, my main concern is the novelty of the findings. Several recent studies have already highlighted that urea oxidation can be more significant than ammonia oxidation in oligotrophic open ocean regions (Wan et al., 2024 <https://doi.org/10.1029/2023GB007996>; Arandia-Gorostidi et al., 2024 <https://doi.org/10.1093/ismejo/wrae230>) and that ureC is prevalent in open-ocean AOA based on metagenomic analyses (Arandia-Gorostidi et al., 2024 <https://doi.org/10.1093/ismejo/wrae230>). In addition, previous work has compared the affinities of urea oxidation and ammonia oxidation across different marine regions (Wan et al., 2024 <https://doi.org/10.1029/2023GB007996>; Xu et al., 2019 <https://doi.org/10.1002/lno.11114>) and reported high ureC expression in the open ocean (Zhao et al., 2024 <https://doi.org/10.1038/s41467-024-50867-z>). While this study adds new insights by applying a novel probe with NanoSIMS to distinguish *Nitrosopumilus* and *Nitrosopelagicus* in terms of urea and ammonia uptake and assimilation, the overarching message largely reinforces prior findings.

Given these considerations, the study is probably more suitable for a specialized journal. A stronger case for novelty, perhaps by emphasizing mechanistic insights beyond what is already known, would be necessary for a high-impact submission.

Reviewer #2

(Remarks to the Author)

Review: «Urea use drives niche separation between dominant marine ammonia oxidizing archaea»

This paper reports on substrate usage of AOA in oligotrophic and coastal waters. The strong point of this paper is that urea can potentially explain the varying dominance of *Nitrosopelagicus* versus *Nitrosopumilus*. The genus *Nitrosopelagicus*, which dominates the open ocean, thrives on urea, while *Nitrosopumilus*, found more commonly in coastal waters, predominantly uses ammonium. This discovery suggests that the relative availability of ammonium and urea may be a critical factor in determining niche separation between AOA genera and fuels open ocean nitrification. This study provides valuable insights into the ecological roles of AOA and shifts our understanding of nitrogen cycling in the open ocean. The data interpretation and conclusions presented in this manuscript are generally robust. The experimental design is well thought out, and the results support the hypothesis that urea can sustain AOA in ammonium-limited waters. The observed differentiation between *Nitrosopelagicus* and *Nitrosopumilus* based on their substrate preference is compelling and presents a clear picture of niche differentiation.

This study has significant implications for marine nitrogen cycling and the broader field of microbial ecology. It challenges the traditional view that ammonium is the primary substrate for nitrification, suggesting that urea and other organic nitrogen

substrates may play a critical role, especially in open ocean environments. This finding is important not only for understanding microbial processes in the ocean but also for understanding how marine ecosystems respond to changing nitrogen sources due to anthropogenic impacts. The data are well presented, and the figures support the narrative effectively. However, there is quite some text below each figure, which could be reduced and more focused on the necessary descriptions needed. Additionally, a total of 10 extended figures also seems excessive. Could they be placed into the supplement file?

The approach taken to analyze the role of urea in AOA activity and growth is well-designed and the data appear to be of high quality. There is quite some method development and comparison on the placement of water column clade B hidden in the supplements, but I enjoyed reading and seeing the figures there. The application of NanoSims for cell specific nitrogen metabolism greatly strengthens the study and is the key to support the main findings.

The manuscript is clear, well-written, and accessible. The significance of the study's findings is appropriately emphasized, and the authors do a good job of situating their work within the current state of research on marine nitrification.

This is a highly impactful study that provides novel insights into the role of urea in marine nitrogen cycling. The research is well-conducted, and the findings significantly advance our understanding of how AOA respond to varying nitrogen conditions in the ocean.

However, I have a few recommendations that could strengthen the paper:

Line 40-41: add reference

Line 48-49: ammonium is not the primary substrate of AOA, it is ammonia. Please be more precise.

Line 57-67: Please mention here direct and in-direct urea utilization strategies. Assimilated for C or/and N or just taken up for N oxidation and C excreted?

Line 68: What about the WCB clade? Please clarify in the text. Do you consider it as part of Nitrosopelagicus in your study or not? After reading auxiliary materials, I found that information, please move it into the main text.

Line 72: be more specific. Which DIC fixation pathway do they have?

Line 83 – 84: What about Wan et al 2024 (<https://doi.org/10.1029/2023GB007996>)? – urea oxidation rates from the coast to the open ocean are reported there and compared to ammonia oxidation rates.

Line 91-92: change “ammonium oxidation rates” to “ammonia oxidation rates”

Line 101: Heading is miss-leading, “towards the ocean” suggests that you sampled a transect, but in fact you investigated 3 very different ecosystems

Line 106: Please refer to Fig1c, when you mention ammonium concentrations in the text

Line 115: Add reference to Fig1d into brackets

Line 165 -167: This key finding is not different from your key take home at the end of the previous paragraph (line 141-143). Therefore, I think this section can be shortened. It is also not novel (see Wan et al. 2024), if there is additional insight compared to Wan et al. 2024, focus on that.

Line 211: “utilize urea”? Do you mean for energy and assimilation, you can not distinguish between the two looking at ureC and dur3.

Line 228: You refer to Supplemental Table 1, but there are only DAPI counts and no comparison to the metagenomic results.

Line 253-256: Can the author say something how these results translate into the urea oxidation capacity of Nitrosopelagicus?

Line 316: remove “is”

Line 337 and following: How much substrate was added? Fraction label? Are you reporting potential rates? – I found the data in supplementary table 1, but you only refer to supplementary table 2, please add 1 as well

Line 341-343: Weren't all samples oxic? Why is O₂ contamination an issue here?

Line 349: How do you differentiate between biotic and abiotic urea-breakdown?

Line 371: Despite adjusting for in situ concentration, all tracer additions are above 90% meaning that all rates are potential rates and not in situ rates. Please add.

Suppl. Table 1: LOD for oxidation rates from Gulf of Mexico missing, not in suppl. table

Supplement Discussion Line 33-44: This distinction/ definition of the different clades is missing in the main section and should go into the main text.

Supplement Discussion Line 122: Should it be named CLONE-FISH instead of DOPE-FISH?

Extended data figure 5: How can you exclude that proteobacteria in your sample split the urea and C and N and only use the C and excrete the N, which is then measured as urea-N oxidation or urea-N assimilation?

Reviewer #3

(Remarks to the Author)

please see attachment for comments

Reviewer #4

(Remarks to the Author)

Version 1:

Reviewer comments:

Reviewer #2

(Remarks to the Author)

The authors have thoroughly revised the manuscript and thoughtfully addressed all of the suggested changes. These revisions have significantly clarified the novelty and impact of the study. I find the manuscript to be greatly improved, and I have no further suggestions at this time. In my view, the study is now suitable for publication in its current form.

Reviewer #3

(Remarks to the Author)

Thank you again for updating the manuscript and your thorough response. A significant amount of work went into reframing the abstract and introduction to highlight the niche differentiation between *Nitrosopelagicus* and *Nitrosopumilus*. The manuscript now builds from bulk to single-cell measurements to provide insight into the abundance of *Nitrosopelagicus* in ammonia-replete waters with high urea to ammonia ratios. Their work provides fundamental insights into the ecophysiology of *Nitrosopelagicus* AOA.

The authors have done an excellent job at addressing my comments. There are a few minor recommendations below with the primary recommendation in the major comment 3 response. It has been a pleasure to review this manuscript.

Reviewer #4

(Remarks to the Author)

Answers to Reviewers:

Please note that all line numbers given here refer to the “Track changed” manuscript version.

Reviewer #1 (Remarks to the Author):

Stuehrenberg et al. investigated the role of urea in sustaining AOA in coastal and open ocean regions, with a focus on niche differentiation between *Nitrosopumilus* and *Nitrosopelagicus*, two abundant genotypes. Using a combination of stable isotope labeling, metagenomics, metatranscriptomics, and NanoSIMS analyses, the authors shows that urea oxidation becomes increasingly important in oligotrophic environments, particularly for *Nitrosopelagicus*. The results provide valuable insights into the metabolic flexibility of marine AOA and their nitrogen utilization strategies.

The manuscript is well written, with robust data supporting its conclusions. However, my main concern is the novelty of the findings. Several recent studies have already highlighted that urea oxidation can be more significant than ammonia oxidation in oligotrophic open ocean regions (Wan et al., 2024 <https://doi.org/10.1029/2023GB007996>; Arandia-Gorostidi et al., 2024 <https://doi.org/10.1093/ismejo/wrae230>) and that ureC is prevalent in open-ocean AOA based on metagenomic analyses (Arandia-Gorostidi et al., 2024 <https://doi.org/10.1093/ismejo/wrae230>). In addition, previous work has compared the affinities of urea oxidation and ammonia oxidation across different marine regions (Wan et al., 2024 <https://doi.org/10.1029/2023GB007996>; Xu et al., 2019 <https://doi.org/10.1002/lno.11114>) and reported high ureC expression in the open ocean (Zhao et al., 2024 <https://doi.org/10.1038/s41467-024-50867-z>). While this study adds new insights by applying a novel probe with NanoSIMS to distinguish *Nitrosopumilus* and *Nitrosopelagicus* in terms of urea and ammonia uptake and assimilation, the overarching message largely reinforces prior findings.

Given these considerations, the study is probably more suitable for a specialized journal. A stronger case for novelty, perhaps by emphasizing mechanistic insights beyond what is already known, would be necessary for a high-impact submission.

We thank the reviewer for their positive assessment concerning the manuscript. Based on their comments we have now made a stronger case for the novelty and importance of our work, emphasizing the mechanistic insights gained in the study.

Moreover, we have entirely rewritten and restructured the introduction to the manuscript, to better reflect the current state of the art. This has allowed us to include the recent studies which have been published while this manuscript has been undergoing the submission and review process. We would like to emphasise, that while we agree that these papers indicate that urea is a relevant substrate for nitrifiers in the oceans, our study provides novel insights into which members of the nitrifying community use urea, thus elucidating the mechanism that leads to urea becoming more important in the open ocean. We show that the increasing use of urea in open ocean reasons is due to the differential response to available energy sources of the main

AOA in the ocean - one generalist (*Nitrosopelagicus*) and one specialist (*Nitrosopumilus*) population. Thus, we find that this insight provides a new mechanistic explanation for the different biogeographic distribution of AOA, and reveals how *Nitrosopelagicus* fuel open-ocean nitrification.

For example, although Wan et al. 2024 GBC show that urea oxidation relative to ammonia oxidation increases from the eutrophic coastal zone to the oligotrophic open ocean and investigate the kinetics of urea use in mixed marine communities (similar to Xu et al. 2019 L&O), neither study links the results to the composition of the *in situ* community. While Zhao et al. 2024 Nat. Comms. show that AOA dominate urease expression in ocean waters, they do not link this to uptake/oxidation rates, nor to the different AOA groups in the ocean. Arandia-Gorostidi et al. 2024 ISMEJ show at a single cell level that a large proportion of microorganisms in the ocean take up urea, but do not target AOA specifically and can therefore only indirectly link this single cell activity to AOA by omics approaches. As we show in our study, the genomic capacity to utilise urea in marine AOA provides few insights into which AOA are actually using urea and when, as both *Nitrosopumilus* and *Nitrosopelagicus* encode and transcribe urease. Only the single cell approach, alongside the specific visualization of the two genera, enabled the insights into differential substrate responses of these main AOA.

Reviewer #2 (Remarks to the Author):

Review: «Urea use drives niche separation between dominant marine ammonia oxidizing archaea»

This paper reports on substrate usage of AOA in oligotrophic and coastal waters. The strong point of this paper is that urea can potentially explain the varying dominance of *Nitrosopelagicus* versus *Nitrosopumilus*. The genus *Nitrosopelagicus*, which dominates the open ocean, thrives on urea, while *Nitrosopumilus*, found more commonly in coastal waters, predominantly uses ammonium. This discovery suggests that the relative availability of ammonium and urea may be a critical factor in determining niche separation between AOA genera and fuels open ocean nitrification. This study provides valuable insights into the ecological roles of AOA and shifts our understanding of nitrogen cycling in the open ocean.

The data interpretation and conclusions presented in this manuscript are generally robust. The experimental design is well thought out, and the results support the hypothesis that urea can sustain AOA in ammonium-limited waters. The observed differentiation between *Nitrosopelagicus* and *Nitrosopumilus* based on their substrate preference is compelling and presents a clear picture of niche differentiation.

This study has significant implications for marine nitrogen cycling and the broader field of microbial ecology. It challenges the traditional view that ammonium is the primary substrate for nitrification, suggesting that urea and other organic nitrogen substrates may play a critical role, especially in open ocean environments. This finding is important not only for understanding microbial processes in the ocean but also for understanding how marine ecosystems respond to changing nitrogen sources due to anthropogenic impacts. The data are well presented, and the figures support the narrative effectively. However, there is quite some text below each figure, which could be reduced and more focused on the necessary descriptions needed. Additionally, a

total of 10 extended figures also seems excessive. Could they be placed into the supplement file?

The approach taken to analyze the role of urea in AOA activity and growth is well-designed and the data appear to be of high quality. There is quite some method development and comparison on the placement of water column clade B hidden in the supplements, but I enjoyed reading and seeing the figures there. The application of NanoSims for cell specific nitrogen metabolism greatly strengthens the study and is the key to support the main findings.

The manuscript is clear, well-written, and accessible. The significance of the study's findings is appropriately emphasized, and the authors do a good job of situating their work within the current state of research on marine nitrification.

This is a highly impactful study that provides novel insights into the role of urea in marine nitrogen cycling. The research is well-conducted, and the findings significantly advance our understanding of how AOA respond to varying nitrogen conditions in the ocean.

However, I have a few recommendations that could strengthen the paper:

We thank the reviewer for their kind comments.

Line 40-41: add reference

This comment was referring to: "This microbially-mediated process is associated with substantial emissions of nitrous oxide, a potent greenhouse gas.". During the rewriting of the introduction this sentence was removed.

Line 48-49: ammonium is not the primary substrate of AOA, it is ammonia. Please be more precise.

We double checked the whole manuscript for this mistake.

Line 57-67: Please mention here direct and in-direct urea utilization strategies. Assimilated for C or/and N or just taken up for N oxidation and C excreted?

This information is incorporated into the rewritten introduction at L60-62.

Line 68: What about the WCB clade? Please clarify in the text. Do you consider it as part of Nitrosopelagicus in your study or not? After reading axillary materials, I found that information, please move it into the main text.

We have now introduced both WCA and WCB in this paragraph (L75-76), and refer the reader to the associated Supplementary Methods earlier in the results (L202).

Line 72: be more specific. Which DIC fixation pathway do they have?

We now specifically state that both AOA groups use the highly energy efficient 3-Hydroxypropionate/4-hydroxybutyrate (3-HP/4-HB) carbon fixation pathway.

Line 83 – 84: What about Wan et al 2024 (<https://doi.org/10.1029/2023GB007996>)? – urea oxidation rates from the coast to the open ocean are reported there and compared to ammonia oxidation rates.

We have now introduced the previous literature looking into open ocean urea utilization early on in the restructured introduction (L67-69).

Line 91-92: change “ammonium oxidation rates” to “ammonia oxidation rates”
Changed as suggested.

Line 101: Heading is miss-leading, “towards the ocean” suggests that you sampled a transect, but in fact you investigated 3 very different ecosystems
Thank you for pointing this out, we agree this may have been misleading and have adjusted the heading accordingly.

Line 106: Please refer to Fig1c, when you mention ammonium concentrations in the text
This information has been added.

Line 115: Add reference to Fig1d into brackets
This information has been added.

Line 165 -167: This key finding is not different from your key take home at the end of the previous paragraph (line 141-143). Therefore, I think this section can be shortened. It is also not novel (see Wan et al. 2024), if there is additional insight compared to Wan et al. 2024, focus on that.

We now specifically reference previous literature and highlight additional information provided by our study (linking this observed activity to specific AOA groups). Now L175-178.

Line 211: “utilize urea”? Do you mean for energy and assimilation, you can not distinguish between the two looking at ureC and dur3.

The presence of urea utilization genes enables both use of urea as energy and N-source. We now specify this in the text (L235-237).

Line 228: You refer to Supplemental Table 1, but there are only DAPI counts and no comparison to the metagenomic results.

We apologise for this oversight and have changed the reference in the sentence so that it also includes Supplementary Data 3, where the omics data is presented.

Line 253-256: Can the author say something how these results translate into the urea oxidation capacity of *Nitrosopelagicus*?

In complex environmental communities it is unfortunately not possible to unambiguously link measured oxidation rates to specific subgroups of co-occurring AOA, as there are no known clade-specific inhibitors. However, given the tight link of substrate preference for oxidation and assimilation observed across all tested ammonia oxidizer pure cultures (Qin et al. 2024 Nat. Microbiol.), we are confident that our measured *Nitrosopumilus/Nitrosopelagicus* specific substrate assimilation rates directly translate to their substrate oxidation rates, and we expect *Nitrosopelagicus* to dominate urea oxidation, irrespective of ammonium availability. We have now included this also in the main text (L282-286).

Line 316: remove “is”

Done as suggested.

Line 337 and following: How much substrate was added? Fraction label? Are you reporting potential rates? – I found the data in supplementary table 1, but you only refer to supplementary table 2, please add 1 as well

In the three environments the ^{15}N tracer additions were adjusted according to the respective nutrient concentrations aiming for >90 % ^{15}N at%. We now refer to both Supplementary Data 1 and Supplementary Table 2 as suggested. We are reporting potential rates, this is also stated in the methods L417.

Line 341-343: Weren't all samples oxic? Why is O₂ contamination an issue here?

In the Black Sea we sampled the water with a pumpCTD system, which enables sampling low oxygen waters with minimal oxygen contamination. Since we chose to set up the incubations with *in situ* oxygen concentrations, we first completely degassed the serum bottles with helium (to also remove potential oxygen contamination introduced by sampling). Afterwards we re-aerated oxygen to the two upper depths of each station, while the incubations from the deeper two depths were run without additional oxygen additions.

Line 349: How do you differentiate between biotic and abiotic urea-breakdown?

For the Angola Gyre and Black Sea cruises, we have not included dead (i.e. sterile filtered) controls for rate measurements, and as such, we assessed the combined abiotic and biotic breakdown of urea in our experiments. However, we have previously shown that there is no measurable abiotic urea breakdown in sterile filtered seawater of the Gulf of Mexico, stored under the same conditions. Thus, any measured urea breakdown in our experiments also in the Black Sea and Angola Gyre can very likely be attributed predominantly to biological activity. We added an additional Supplementary Discussion section on this topic.

Line 371: Despite adjusting for in situ concentration, all tracer additions are above 90% meaning that all rates are potential rates and not in situ rates. Please add.

In L417 we state that rates are potential rates.

Suppl. Table 1: LOD for oxidation rates from Gulf of Mexico missing, not in suppl. Table
We apologize for this oversight. We now added the limit of detection for the oxidation rates in the Gulf of Mexico. Briefly, they were 25.1, 3.3 and 2.0 nM day⁻¹ for ammonia oxidation, urea oxidation and urea oxidation + $^{14}\text{NH}_4^+$, respectively. Please note that the ammonia oxidation rates in the Gulf of Mexico are exceptionally high, which is why the calculated limit of detection is higher than for urea oxidation.

Supplement Discussion Line 33-44: This distinction/ definition of the different clades is missing in the main section and should go into the main text.

We now describe the differentiation of *Nitrosopelagicus* WCA and WCB briefly in the introduction (L75-76), but feel that an in-depth discussion is better suited for the Supplementary Information. We now mention this earlier in the results (L201-203).

Supplement Discussion Line 122: Should it be named CLONE-FISH instead of DOPE-FISH? This should read DOPE-FISH - for probe evaluation of the Nitrosopumilus-specific probe Npum_229, we did not have to use the CLONE-FISH approach, as unlike for the *Nitrosopelagicus* WCA and WCB probes, we had a *Nitrosopumilus* pure culture representative (*N. adriaticus*) available for probe optimization. To clarify this, we have changed the wording of the previous paragraph (Supplementary Methods L140-142).

Extended data figure 5: How can you exclude that proteobacteria in your sample split the urea und C and N and only use the C and excrete the N, which is then measured as urea-N oxidation or urea-N assimilation?

In the Gulf of Mexico, where proteobacterial ureases are most abundant and urea-oxidation rates are high, we do see evidence of indirect utilization - i.e. formation of ¹⁵N-ammonia which is subsequently taken up by the AOA (now added to the introduction). However, we can largely rule out that indirect urea-oxidation is significant in the Angola Gyre and Black Sea based on comparison of the ¹⁵N-urea incubations with and without ammonium. Firstly, in the ¹⁵N-urea incubations without ammonium, breakdown and release of ¹⁵N-ammonium from ¹⁵N-urea by other microorganisms would progressively increase the amount of ¹⁵N in the extracellular ammonium pool, which would lead to exponential production of ¹⁵N-nitrite from ammonia oxidation over time, which we did not observe. Secondly, in the incubations with addition of a ¹⁴N-ammonium pool (ammonium pool incubations), the likelihood of any indirect use of ¹⁵N-ammonium would be decreased, which, if indirect utilization were significant, would translate into a lower production of ¹⁵N-nitrite compared to the incubations without added unlabeled ammonium. We also did not observe this in the Black Sea and Angola Gyre. Furthermore, we measured only sporadic and negligible production of ¹⁵N-ammonium in incubations with added ammonium in the Black Sea and Angola Gyre. Analogously, if indirect utilization played a major role in N-assimilation, we would expect to see a strong decrease in ¹⁵N-assimilation in the urea incubations with ¹⁴NH₄⁺ compared to those without. However, *Nitrosopelagicus* did not show decreased urea assimilation rates in presence of ammonium. Based on your comment we realised that we had not included this information in the manuscript and have now added it as Supplementary Discussion.

Reviewer #3 (Remarks to the Author):

Overall judgment: The manuscript Urea use drives niche separation between dominant marine ammonia oxidizing archaea propose ecological niches for *Nitrosopelagicus* and *Nitrosopumilus* — two genera of marine ammonia-oxidizing archaea (AOA) — based on single-cell assimilation of ammonium and urea in the Black Sea. Previous work from the authors has demonstrated distinct ammonium (NH₄⁺) and organic nitrogen utilization strategies by AOA and nitrite-oxidizing bacteria (NOB) in the Gulf of Mexico (Kitzinger et al., 2020, 2019). The dominance of *Nitrosopelagicus* ammonia oxidizing archaea (AOA) in the open ocean and *Nitrosopumilus* AOA in estuarine areas has been well documented (Qin et al., 2020; Santoro et al., 2017). However, physiological traits defining the ecological niche of these ubiquitous marine AOA has remained elusive. Demonstrating distinct nitrogen metabolite preferences dictates the environmental niche of these marine AOA genera — as was recently demonstrated for the major ammonia-oxidizing microorganism lineages (Qin et al., 2024) — would represent a significant contribution to the

field. All experimental and computational approaches were sufficient. However, specific techniques and software packages should be explicitly stated in the methods section and there were major concerns listed below. Care should be taken to update the introduction and discussion to improve clarity. The manuscript provides compelling evidence of genus-level niche differentiation between marine AOA and could shape our understanding of the marine nitrogen cycle. Despite presenting compelling evidence for the distinct metabolite preferences of *Nitrosopelagicus* and *Nitrosopumilus* in the Black Sea, this manuscript would have benefited by constraining the scope to the Black Sea and linking the single-cell observations with the environmental rate and metaomic datasets. The most significant findings of this worked can be captured as following:

Significance #1: Combining single-cell N-assimilation measurements with in-situ genus level phylogeny in samples collected from the Black Sea enabled the authors to present compelling evidence of the distinct strategies *Nitrosopelagicus* and *Nitrosopumilus* to employ their ammonia and urea utilization (Figure 3). Their results suggest *Nitrosopumilus* prefer to utilize ammonia and repress urea utilization in the presence of ammonia, while *Nitrosopelagicus* did not demonstrate a distinct preference toward ammonia or urea. Despite preferring ammonia, the ^{15}N -urea biomass enrichment — and the calculated growth rate — of *Nitrosopumilus* and all other microorganisms appear to be faster than *Nitrosopelagicus* when ^{15}N -urea was supplemented. Yet, *Nitrosopelagicus* may outcompete *Nitrosopumilus* for urea in the presence of excess ammonia based on higher ^{15}N urea biomass enrichments in samples supplemented with ^{15}N -urea and ^{14}N - NH_4^+ . Considering all marine environments are characterized by both ammonium and urea, it would suggest the niche of *Nitrosopelagicus* is defined by urea-utilization in ammonium replete marine environments.

We thank the reviewer for their kind assessment.

Shortfall #1: Unfortunately, the compelling single-cell analysis from the Black Sea could not be extended to the oligotrophic Angola Gyre where *Nitrosopelagicus* comprised a larger proportion of the nitrifier community.

While we agree that it would have been a nice addition to extend the single-cell analysis to the Angola Gyre, we find that the Black Sea single-cell analysis provides the most insightful information - as this is the only environment where we can test the response of environmental communities of both *Nitrosopelagicus* and *Nitrosopumilus* in the same samples, without confounding changes in other environmental conditions. We now also highlight this better in the introduction (L99-103) and the main text (L248-251).

Significance #2: The authors also characterize the relative contribution of ammonium and urea to ammonia oxidation in three distinct marine environments with unique productivity regimes (Figure 1 & 2).

Shortfall #2: Though presented as a novel finding in the manuscript, the increased contribution of urea relative to NH_4^+ to nitrification in the open ocean has been reported several times before and does not challenge the current understanding of marine nitrification (Arandia-Gorostidi et al., 2024; Santoro et al., 2017; Tolar et al., 2017; Wan et al., 2024). The distribution

of urea-dependent ammonia-oxidation (hereafter urea-oxidation) and ammonia-oxidation is well known at the global scale. The urea-dependent ammonia-oxidation rate and ammonia-oxidation rate decrease from copiotrophic eutrophic coastal zones to the oligotrophic open ocean while the contribution of urea to total ammonia-oxidation increases due to decreasing substrate concentrations (Wan et al., 2024). The authors confirm the validity of this framework in non-polar marine regions. However, the isotopic rates measurements were not paired with single-cell analysis, metagenomics, and metatranscriptomics to study the niche differentiation between *Nitrosopelagicus* and *Nitrosopumilus*.

We agree that there are now a number of observations that urea-derived oxidation rates seem to contribute more to nitrification in oligotrophic environments. A number of these studies have been recently published, and we have now rewritten the introduction of the manuscript to reflect this state of the art, and have made this more evident throughout the manuscript. Yet, we think it is important to demonstrate that this observation holds true also for the environments investigated here, as this is key background information for our single cell analyses, which enable us to link specific substrate utilization patterns to specific AOA groups, which is the main finding of our study. As the reviewer points out, this has not been undertaken by any previous studies.

Major Points:

1. The distribution of urea-oxidation and ammonia-oxidation in marine environments is well known at the global scale. Recent studies have also demonstrated the contribution of urea and ammonia to total ammonia oxidation by mesopelagic and bathypelagic AOA is nearly equivalent (Arandia-Gorostidi et al., 2024; Laperriere et al., 2021; Santoro et al., 2017; Tolar et al., 2017; Wan et al., 2024). Therefore, the contribution of urea to nitrification has been demonstrated and the rate measurements presented here do not represent a unique or novel finding. It is important that the authors accurately represent our current understanding.

We agree with the reviewer and have ensured that this is appropriately acknowledged throughout the manuscript.

2. In our opinion the most significant claim in the manuscript is urea utilization in the presence of ammonium defines the niche of *Nitrosopelagicus*. However, this finding is not emphasized throughout the manuscript and must be strengthened as a result. The authors should consider framing the manuscript around this finding as it might explain the distribution of *Nitrosopelagicus* in the open ocean. Compelling evidence that *Nitrosopelagicus* AOA are responsible for urea hydrolysis in the Black Sea or Angola Gyre is not presented and would serve as an indicator of their distinct preference for urea-derived ammonium over ammonium if presented. Neither the environmental rate data in Figure 2a-b nor the meta-omic data in Figure 2c-d and Extended Data Fig. 5 contribute to defining the niche of *Nitrosopelagicus*. Focusing on the depth resolved samples collected from the Black Sea instead of clustering all samples together may provide additional insight and it is suggested that the authors provide these additional data sets. Please comment on how the oxidation rates and meta-omic sampling contribute to understanding the environmental niche of *Nitrosopelagicus* and *Nitrosopumilus* in the Black Sea.

We agree with the reviewer that the continued use of urea in the presence of ammonium by *Nitrosopelagicus* is a major finding of our manuscript and have put more emphasis on this result

in the text. However, we strongly feel that the detailed description of the study sites and bulk rate measurements and -omics analyses are required to set the scene and provide the background for our single cell measurements, which ultimately link *Nitrosopelagicus* to urea utilization. We further are convinced that the presented single cell data (unlike the bulk data, which is ambiguous) unequivocally shows that *Nitrosopelagicus* utilizes urea.

In response to the reviewers' comment we have now included additional panels (Supplementary Figure 2) showing exemplary depth distributions for the Black Sea of ammonium concentrations, oxidation rates and *Nitrosopumilus/Nitrosopelagicus* abundance, which highlights that *Nitrosopelagicus* is more abundant in ammonium-poor waters.

3. The caveats associated with varying experimental methods across samples and the impact on the subsequent statistical comparisons needs to be discussed. Different methods were used to measure oxidation rates with stable isotope tracers on the Gulf of Mexico, Black Sea, and Angola Gyre cruises. There were also experimental variations between sampling depths on Black Sea cruise. Specific differences include the incubation bottles (serum bottle & exetainer), oxygen addition methods (in-situ & He/N₂ sparge, or sparge & O₂ addition), supplemented metabolites (N₂O & DIC), supplemented metabolite concentration, and supplemented isotope percentages differed between the cruises (lines 338-412). A primary concern is the influence of tracer and oxygen additions on the measured rates, which is not discussed in the manuscript. A discussion about methodological differences and subsequent statistical analysis is requested.

The experimental setup was adjusted during each cruise to match *in situ* conditions as accurately as possible while also accommodating best feasibility. The ¹⁵N tracer additions differed due to the different nutrient availability across sites. Similarly, the oxygen concentrations in the incubations were adjusted to match *in situ* oxygen conditions. By mimicking the respective *in situ* conditions we anticipate that our potential oxidation rates reflect the actual rates as closely as possible. The incubation period is short (<24h) and the rates are linear from the beginning of the incubation on, without a lag phase, which indicates that ammonia oxidizers are active *in situ*. While experimental details differed between cruises, the association of *Nitrosopelagicus* with enhanced urea use holds across our study sites, and also other studies in open ocean settings where *Nitrosopelagicus* typically dominates (Santoro et al. 2017 L&O, Wan et al. 2024, GBC). Thus, we are confident that these minor differences in protocols did not impact the outcome of our experiments, or our conclusions. To address this issue, we added a section to the Supplementary Discussion.

4. Ammonium concentrations classically define the environmental niche of ammonia-oxidizing microorganisms. A recent study found the ammonia affinity was higher than the urea affinity in the open ocean providing an explanation for the distribution of urea in this environment (Wan et al., 2024). However, a discussion about the impact of concentrations on niche differentiation is lacking in the current manuscript, especially while discussing direct versus indirect urea oxidation rates (Figure 2b). For example, the axes for the Gulf of Mexico are an order of magnitude greater than the other sampling sites in Figure 2b making it difficult to make a direct comparison between the sampling sites. It is apparent that the addition of ammonia significantly decreased the urea oxidation rate in the Gulf of Mexico, while the tracer has no impact in the Black Sea and Angola Gyre. However, the Black Sea and Angola Gyre urea oxidation rates are

an order of magnitude lower than the presented Gulf of Mexico rates. The different responses may be attributed to the lower *Nitrosopumilus* (0.04 d⁻¹) and *Nitrosopelagicus* (0.002 d⁻¹) NanoSIMS-based growth rates in the Black Sea (line 229-236) compared to the Gulf of Mexico (0.23 d⁻¹). The faster growth rate may enable the Gulf of Mexico AOA population to transition their metabolism away from urea-oxidation over the 24 hour isotope incubations. The Black Sea and Angola Gyre AOA populations may also transition their metabolism towards ammonia-oxidation if the incubation times were corrected for the AOA growth rates. Therefore, we suggest that you consider discussing the impact of concentration and growth rates on direct versus indirect urea oxidation rates.

Generally, we agree that it is possible that the AOA population transitions their metabolism depending on the substrate available. This is the reason why we chose to incubate <24h. Additionally, if as hypothesized the Gulf of Mexico AOA would transition away from urea oxidation, we would see a flattening of the nitrite production from urea over time, however, we see a strong linear trend over time. The same holds true for the hypothesis that the Angola Gyre and Black Sea AOA population transition towards ammonia oxidation. We see a strong linear trend without a lag phase in all our rates. Additionally, in the single cell measurements, we see that even though *Nitrosopumilus* reduces its urea utilization in the presence of ammonium, *Nitrosopelagicus* did not show decreased urea assimilation rates in presence of ammonium. Together with the observed strong correlation of *Nitrosopelagicus* dominating in high urea relative to ammonium availability (concentration based) (and vice versa for *Nitrosopumilus*), we are confident that our observations are a result of inherent physiological traits defining these AOA.

5. There are distinct metabolite distributions with depth presented in Extended Data Figure 1, which are not captured in Figure 1b-d or Figure 2a-b. The ammonium concentrations detected in the Black Sea samples are near the limit of detection until ~120m where the ammonium concentrations increase up to 7.4 μM-NH₄⁺. The distribution of AOA in the Black Sea samples also appears to have an inverse relationship with ammonia availability based on the Thermoproteota ureC RPKM dataset presented in Extended Data Figure 5b. Please, consider presenting the nutrient distributions collected at each sampling station, depth integrated nutrient inventories, or boxplots corresponding to specific depths within the sample site to characterize the environmental gradient within the Black Sea.

We thank the reviewer for the suggestion. We now present ammonium concentration profiles along with ammonia and urea oxidation rates and relative AOA abundance from the Black Sea in Supplementary Figure 2. We also refer to this depth distribution in the main text e.g. L115-116.

6. The difference in AOA cell concentrations might contribute to the rate measurements presented in this manuscript. Please discuss how bulk rates are dependent on cell concentrations and cell specific rates. This is important because the two order of magnitude difference might be attributed to an order of magnitude difference in cell concentration and an order of magnitude difference in the cell specific rate. Additionally, consider presenting the ammonia-oxidizer cell abundance either in the main text, extended data, or supplemental material.

We agree with the reviewer that both the overall abundance of AOA and their specific activity influence the measured oxidation rates. It appears that in the highly productive Gulf of Mexico, cell specific rates are much higher than in the more oligotrophic Black Sea and Angola Gyre. We have already alluded to this mismatch between AOA abundance and oxidation rates in the previous version of the manuscript, and now specifically mention the difference in specific cell rates. AOA cell numbers are provided both in the main text, and in Supplementary Data 1.

7. The axes limits are inconsistent between conditions in many figures making direct comparison difficult. This is especially apparent for the ^{15}N biomass enrichment data of *Nitrosopelagicus* presented in Figure 3c, which is an order of magnitude lower than other genera. Therefore, we recommend adjusting the axis accordingly.

We took your suggestions into consideration. However, we have tried multiple options of data depiction (e.g. same axes limits, log scale) for this particular plot and found that the current formatting showed the data most clearly. As the axes limits are exactly one magnitude different, the comparison is still possible and we also specifically point out the difference in scales in the figure legend.

8. Please remove the statement “thus essentially doubling its growth rate” in line 256. The data presented in this manuscript does not demonstrate the growth rate of *Nitrosopelagicus* was doubled via urea and ammonium co-consumption. Previous observations of co-consumption by *Nitrosococcus oceanus* did not report a statistical difference in the growth rate while consuming both ammonia and urea. The statement is not sufficiently supported by the experimental results presented in Figure 3 and should be removed from the manuscript.

We agree that this sentence was not careful enough given the presented evidence and have revised it. It now says “Furthermore, although we cannot unequivocally show that *Nitrosopelagicus* cells were assimilating ammonium and urea simultaneously, if they do, then this strategy could potentially lead to a doubling of their growth rate when both ammonium and urea are available.” (L288-291).

Minor Points (line-by-line):

Line 29-31: The increased contribution of urea relative to NH_4^+ to nitrification in the open ocean has been reported several times before (Arandia-Gorostidi et al., 2024; Tolar et al., 2017; Wan et al., 2024) and does not challenge the current understanding of marine nitrification. Please tone down novelty claims of your findings.

We have rephrased the abstract taking into consideration the reviewer comments.

Line 31-33: The dominance of *Nitrosopelagicus* ammonia oxidizing archaea (AOA) in the open ocean and *Nitrosopumilus* AOA in estuarine areas was previously reported by (Santoro et al., 2015) and Qin et al., 2020. However, physiological traits defining the ecological niche of these ubiquitous marine ammonia-oxidizing archaea have not yet been reported and would represent a major contribution to the field. Please revise manuscript to integrate this rationale.

We have rephrased the abstract taking into consideration the reviewer comments.

Line 33-36: The trend of increased urea-N oxidation relative to ammonia oxidation from eutrophic coastal zones to the oligotrophic open ocean was previously demonstrated by (Wan et al., 2024) in the Subtropical North Pacific. Their results were further validated by comparing the urea to NH₄⁺ concentration and oxidation rate ratios across the coastal, epipelagic zone, and mesopelagic zone across the global ocean. Please tone down novelty claims of your findings. We have rephrased the abstract taking into consideration the reviewer comments.

Line 40-41: AOA should be presented earlier in the introduction considering the hypothesis presented in the manuscript is the relative availability of ammonium versus urea drives niche differentiation between AOA the Nitrosopelagicus and Nitrosopumilus genus. We have restructured the introduction based on the reviewer comments, and currently introduce AOA already in the second paragraph of the introduction, after a short first paragraph describing why it is important to understand nitrification in the open ocean. In the revised introduction we have added substantial new text into the introduction regarding what is known about niche differentiation.

Line 55-56: While there are still unresolved aspects about the factors shaping the distribution of AOA at the regional scale, the factors shaping the global distribution of urea and ammonia oxidation in the global ocean are known at the global scale (Wan et al., 2024). Although the urea-dependent ammonia-oxidation rate and ammonia-dependent ammonia-oxidation rate decrease from eutrophic coastal zones to the oligotrophic open ocean, the contribution of urea to total ammonia-oxidation increases due to substrate concentrations. Please update the statement in line 55-56 to reflect this understanding. It might be worthwhile to mention areas where this paradigm does not hold. We thank the reviewer for this comment. In the revised introduction, we now specifically refer to patterns of urea utilization and higher urea availability relative to ammonium in the oligotrophic oceans.

Line 57: The mechanism of urea derived utilization should be mentioned while introducing urea as an AOA substrate. Urea must be transported into the cell where it is hydrolyzed by urease to ammonia and carbon dioxide. The intracellularly derived ammonia can then be utilized for ammonia-oxidation by the ammonia monooxygenase. Consider explicitly explaining this mechanism in the introduction. We have added this information to the introduction.

Line 58-59: Consider using a different reference to introduce urea utilization by marine AOA, since the contribution of urea to total ammonia-oxidation was minor as stated in lines 62-64. The contribution of urea was hypothesized based on genomic surveys (Alonso-Sáez et al., 2012) and the capacity of a marine AOA to utilize urea was demonstrated with the marine strain PS0 isolated from Puget Sound (Qin et al., 2014). The contribution of ammonium and urea to total ammonia oxidation were later compared across pelagic environments where it was found to have a major contribution in Antarctic coastal waters and a minor contribution in temperate coastal waters (Tolar et al., 2017).

We have restructured and revised the introduction to include more studies investigating urea utilization. However in this particular section we are referring to Kitzinger et al. 2019 Nat. Microbiol., as we are talking about environmental growth rates of AOA on small organic N-compounds, which have only been reported in this paper.

Line 62-66: Recent studies have demonstrated the contribution of urea and ammonia to total ammonia oxidation by mesopelagic and bathypelagic AOA is nearly equivalent (Arandia-Gorostidi et al., 2024; Laperriere et al., 2021; Santoro et al., 2017; Tolar et al., 2017; Wan et al., 2024). Therefore, the importance of urea no longer remains a hypothesis. Please update the statements in line 62 and 65-66 to reflect the current understanding.

We revised the introduction accordingly.

Line 66: The different AOA genera (i.e. Nitrosopelagicus and Nitrosopumilus) have not yet been introduced. Please refrain from mentioning AOA genera until they have been introduced. Consider moving statement anywhere after line 73.

We have revised and restructured the introduction and now early on introduce the two main genera of AOA (L72-74).

Line 72: Please mention carbon fixation pathway as it is important when comparing carbon assimilation across nitrifying lineages.

This is now included in the introduction.

Line 74-75: The word “many” is not quantitative and does not denote magnitude. Either remove statement or include percentage of Nitrosopelagicus and Nitrosopumilus containing urease and urea transport proteins.

We now specify precisely the isolates that encode and utilise urease.

Line 80-83: This is the crux of the manuscript. The physiological mechanisms underpinning the distribution of Nitrosopelagicus and Nitrosopumilus remain unknown considering their genomic similarity. This should be highlighted earlier in the introduction, especially the lack of in-situ comparisons.

We agree with the reviewer that this is the main point of the manuscript. We now introduce that the factors driving the different distributions of *Nitrosopumilus* and *Nitrosopelagicus* are unknown early on in the revised introduction (L76-80), before describing the similarities on genomic level.

Line 86-94: The use of the phrase “environmental gradient ...” can be easily misconstrued here. Samples were collected from three distinct marine environments, instead of sampling a transect defined by a nutrient gradient. The sampling sites provide a snapshot into the environmental factors driving niche differentiation there, but nothing about transitions between different environmental conditions. Please specify why each site was chosen and the comparisons that were made between sites.

We rephrased this sentence for clarity. The reasoning for choosing these three sites is given in the introduction.

Line 86-94: Contribution of urea to nitrification along a gradient from eutrophic coastal waters to oligotrophic open oceans was recently demonstrated by (Wan et al., 2024). However, the isotopic rates measurements by Wan were not paired with single-cell imaging, metagenomics, and metatranscriptomics. This study provides new insights into the niche differentiation between of Nitrosopelagicus and Nitrosopumilus. Please focus the results on the niche differentiation. We agree that the niche differentiation is the key point of our manuscript and we now put more focus on the single cell results throughout. Nevertheless, the additional data (oxidation rates, metagenomics, metatranscriptomics) are essential to support these results and set the scene.

Line 94: Urea utilization by AOA in the Gulf of Mexico has already been published by the authors (Kitzinger et al., 2020, 2019). Results from the Black Sea and Angola Gyre cruise should be compared to the Gulf of Mexico cruise. Consider focusing novelty claims on the results of the Black Sea and Angola Gyre cruise and referencing the published Gulf of Mexico dataset.

We agree that the Gulf of Mexico dataset has a supporting role in the manuscript, however, we have re-analyzed all -omics data for consistency between environments and therefore find it essential to discuss them explicitly.

Line 106-109: There are distinct metabolite distributions with depth presented in Extended Data Figure 1, which are not captured in Figure 1b-d. The ammonium concentrations detected in the Black Sea samples are near the limit of detection until ~120m where the ammonium concentrations increase up to 7.4 $\mu\text{M-NH}_4^+$. The distribution of AOA in the Black Sea samples also appears to have an inverse relationship with ammonia availability based on the Thermoproteota ureC RPKM dataset presented in Extended Data Figure 5b. Consider presenting the nutrient distributions collected at each sampling station, depth integrated nutrient inventories, or boxplots corresponding to specific depths within the sample site.

We adjusted Supplementary Figure 2 and this information to the main text, as described under major point 5.

Line 106-109: The manuscript aims to link AOA ecotypes to the relative availability of ammonia and urea in distinct marine ecosystems. However, the environmental niche of ammonia-oxidizing organisms as defined by abundance and oxidation rates is ultimately regulated by their substrate affinity and the environmental nutrient concentration. Please consider (1) presenting the total ammonium + urea concentration alongside the fractional urea abundance using a secondary y-axis and (2) presenting the nutrient data alongside the oxidation rate and AOA relative abundance datasets presented in Figure 2.

We have carefully considered adding the suggested additional panels to the main figures. However, as we already provide all relevant data in the Supplementary Table(s) & Data, we instead chose to show two exemplary depth distributions of ammonium concentrations, oxidation rates and abundance of the different AOA groups for the Black Sea as extra panel of Supplementary Figure 2, as suggested by the reviewer in the comment above.

Line 106-109: Consider presenting metabolite concentrations and ratios in Figure 1 with a log axis.

While we have considered this, the resulting plot is very hard to read and thus we have not changed the current presentation.

Line 106-109: The x-axis for individual metabolites differs between sampling locations in Extended Data Figure 1. Please maintain a consistent x-axis to ensure the sampling sites can be easily compared.

We considered adjusting the Supplementary Figure 1. However, if we maintain consistent x-axes, the environment-specific depth distributions are not visible (e.g. ammonium concentrations in the Angola Gyre appear to be zero). Instead we now point out the different axis scales in the figure legend.

Line 115-118: The Angola Gyre has the lowest total ammonium + urea concentrations presented in this study in addition to having the highest fractional urea abundance. This observation is consistent with data collected throughout the Global Ocean that show an increasing urea:NH₄ + ratio as ammonium and urea concentrations decrease (Wan et al., 2024). Please include information about the total concentrations while discussing the fractional urea abundance.

We have adjusted this section as requested.

Line 119: Please define the term relative availability as it relates to urea-N and NH₄ + -N. It would be recommended to additionally define availability in terms of concentration considering the relationship between urea:NH₄ + concentration and rate ratios does not hold for polar marine regions (Damashek et al., 2019; Shiozaki et al., 2021; Tolar et al., 2017). However, there is a distinct correlation in the Global Ocean (Wan et al., 2024).

We now introduce relative urea-N availability in L120, and reference Wan et al. 2024 GBC specifically. However, we feel that a discussion of why this relationship may be less pronounced in polar regions is beyond the scope of the present manuscript.

Line 121-122: The caveats associated with varying experimental methods across samples and the impact on the subsequent statistical comparisons needs to be discussed. Different methods were used to measure oxidation rates with stable isotope tracers on the Gulf of Mexico, Black Sea, and Angola Gyre cruises. There were also experimental variations between sampling depths on Black Sea cruise. Specific differences include the incubation bottles (serum bottle & extainer), oxygen addition methods (in-situ & He/N₂ sparge, or sparge & O₂ addition), supplemented metabolites (N₂O & DIC), supplemented metabolite concentration, and supplemented isotope percentages differed between the cruises (lines 338-412). A primary concern is the influence of tracer and oxygen additions on the measured rates, which is not discussed in the manuscript. Please respond to these caveats as part of this review.

While experimental details differed between cruises, urea use has previously been reported to increase towards oligotrophic regions using different experimental setups, and conducted by different authors (e.g. Santoro et al. 2017 L&O; Wan et al. 2024 GBC). Thus, we are confident

that minor differences in protocols did not impact the outcome of our experiments, or our conclusions.

Specifically, we would like to clarify that within each cruise, we only compare treatments (i.e. ammonia and urea oxidation) that received the same amendments in regard to oxygen and DIC. Oxygen availability plays a key role for AOA activity (as it is required for activation of ammonia into hydroxylamine and as terminal electron acceptor), and oxygen affinities of AOA are exceptionally high (nM range, Bristow et al. 2016 PNAS). In our experiments, oxygen was adjusted to reflect *in situ* conditions, however, we anticipate that even samples from anoxic depths contained some residual oxygen as even careful handling introduces oxygen contamination (DeBrabandere et al. 2012 J. Microbiol. Methods). The linearity of our rates, also in low oxygen samples, indicates that oxygen was present at higher concentrations than required for AOA activity.

We do not anticipate that differences regarding the addition of DIC or N₂O pools between cruises directly affect the measured nitrification rates. DIC additions were only 10% of the ambient DIC pool (which is unlikely to stimulate DIC fixation activity), and N₂O is not the result of an enzymatic reaction in AOA, therefore it should have a negligible effect on the measured rates.

Finally, the ammonia oxidation rates measured differ by orders of magnitude between the three environments. While differences in experimental setup could have small effects on the measured oxidation rates, we do not expect them to cause such an order of magnitude difference.

We added an additional section discussing differences between cruises to the Supplementary Discussion.

Line 122: It is difficult to compare the specific tracer and pool additions in Supplementary Table 2. Please include an individual row or column for each metabolite. Consider adding the total 15N percentages in the table.

Supplementary Table 2 was adjusted as suggested.

Line 122-129: Lower ammonia-oxidation rates are observed in environments with lower measured NH₄⁺ concentrations. However, the ammonia oxidation rates and comparison between ammonium concentration and the ammonia oxidation rate are not presented. Please include and reference these datasets.

Ammonia oxidation rates along with ammonium concentrations as well as urea oxidation rates along with urea concentrations and urea oxidation to total oxidation ratios are now presented in Supplementary Data 1. We now also refer to this in the text.

Line 130: Please define urea-derived ammonia oxidation and the urea oxidation notation in the introduction instead of the results.

We have made this change as suggested.

Line 132: The urea oxidation rates are not explicitly compared in Figure 2b. Please include and reference these comparisons. Consider adding comparison between urea concentration and urea oxidation rate as well.

It is true that the urea oxidation rates are not explicitly compared in Figure 2b, however, the difference in magnitude is visible by comparing the different panels. We now additionally mention this in the figure legend, and also refer to Supplementary Data 1, where urea oxidation rates along with urea concentrations are presented.

Line 133-141: Please present the absolute ammonia and urea oxidation rates in addition to the fractional rates presented in Figure 2a. The oligotrophic Angola Gyre has the lowest measured urea oxidation rates and ammonia concentrations presented in the study. Trends in the Global Ocean demonstrate that decreasing ammonia concentrations in the open ocean correlate with increased the urea:NH₄⁺ concentration and oxidation ratios (Wan et al., 2024). The results presented here further confirm these observations. However, an explicit comparison between concentration, rate, and ratios should be presented here.

The absolute rates can be found in Supplementary Data 1. We now also have an explicit comparison between *in situ* concentrations, rates and ratios in Supplementary Data 1.

Line 147-148: The statement “as urea must ... via ammonia monooxygenase” should be moved to the introduction.

As suggested, this information is now in the introduction.

Line 147: Urea is also broken down to carbon dioxide (CO₂), which can be assimilated. Please mention CO₂ while discussion urea hydrolysis

We now mention that urea is broken down into carbon dioxide in the introduction L59-60.

Line 148-159: The addition of a large background NH₄⁺ to labeling incubation have previously been used to examine whether the ¹⁵N₂O – was produced via direct or indirect oxidation of the ¹⁵N labeled substrates (Kitzinger et al., 2019; Wan et al., 2024). Please update section to reflect the measurement of direct or indirect urea oxidation.

We have now referenced a new Supplementary Discussion paragraph here discussing that these experiments can also be used to examine the direct and indirect oxidation of ¹⁵N-labeled substrates.

Line 152-159: The axes for the Gulf of Mexico are an order of magnitude greater than the other sampling sites in Figure 2b making it difficult to make a direct comparison between the sampling sites. It is apparent that the addition of ammonia significantly decreased the urea oxidation rate in the Gulf of Mexico, while the tracer has no impact in the Black Sea and Angola Gyre.

However, the Black Sea and Angola Gyre urea oxidation rates are an order of magnitude lower than the presented Gulf of Mexico rates. The different responses may be attributed to the lower Nitrosopumilus (0.04 d⁻¹) and Nitrosopelagicus (0.002 d⁻¹) NanoSIMS-based growth rates in the Black Sea (line 229-236) compared to the Gulf of Mexico (0.23 d⁻¹). The faster growth rate may enable the Gulf of Mexico AOA population to transition their metabolism away from urea-oxidation over the 24-hour isotope incubations. The Black Sea and Angola Gyre AOA populations may also transition their metabolism towards ammonia-oxidation if the incubation times were corrected for the AOA growth rates. Consider discussing the impact of concentration and growth rates on direct versus indirect urea oxidation rates.

Please refer to major point 4, where we have addressed this comment in detail.

Line 160: Please change “all known ammonia oxidizers” to “all known ammonia-oxidizing archaea”. All ammonia oxidizers encode an ammonia monooxygenase (AMO) and hydroxylamine oxidoreductase (HAO) to oxidize ammonia to nitric oxide (Caranto and Lancaster, 2017). However, ammonia-oxidizing bacteria (AOB) and ammonia oxidizing archaea (AOA) have different responses to the nitric oxide (NO) scavenger PTIO, suggesting the mechanism converting nitric oxide (NO) to nitrite may differ between AOB and AOA (Martens-Habbenha et al., 2015).

We agree and apologise for the imprecise wording, which we have now changed accordingly in the revised manuscript.

Line 162-165: Please specify that only the Betaproteobacteria ammonia-oxidizing bacteria isolates demonstrated a clear preference for urea, while the marine Gammaproteobacteria AOB isolate did not demonstrate a clear preference for urea or ammonium (Qin et al., 2024). The Gammaproteobacteria are probably most relevant for this study.

We now specify that betaproteobacterial ammonia oxidizers show a preference for urea.

Line 165-167: Please indicate the increasing importance of urea in oligotrophic regions has been demonstrated several times before with ¹⁵N tracer studies. These results further confirm the hypothesis in different marine environments.

We revised the entire manuscript to refer to other studies more frequently.

Line 174: The referenced Extended Data figure appears to be incorrect. Extended Data Figure 3 is a phylogenetic tree and does not include information about AOA abundance. It appears the intended reference was Extended Data Figure 5.

We apologize for this oversight, neither Supplementary Figure 3 nor Supplementary Figure 5 depict that AOA are the main ammonia oxidizers, but it can be found in the Supplementary Data 1 and 3. We corrected this in the text.

Line 175-178: The difference in cell concentration appears to be driving the measurements presented in this manuscript. Consider presenting the ammonia-oxidizer cell abundance either in the main text, extended data, or supplemental material.

If we understand the reviewer correctly, they are asking for AOA cell numbers that are already given in this section and we now specifically mention differences in single cell activity between environments (L194). The cell numbers can also be found in Supplementary Data 1.

Line 179-181: Please discuss how bulk rates are dependent on cell concentrations and cell specific rates. This is important because the two order of magnitude difference might be attributed to an order of magnitude difference in cell concentration and an order of magnitude difference in the cell specific rate.

Thank you for this comment. We agree that the bulk oxidation rates are influenced by the abundance of AOA cells, and additionally, AOA in nutrient-richer environments may generally

have higher cell specific rates compared to those in nutrient deplete environments. We have now addressed this within the manuscript (L194).

Line 181-188: The contrasting abundance and pangenomics of *Nitrosopumilus* and *Nitrosopelagicus* in coastal estuaries and the open ocean has previously been reported (Qin et al., 2020; Santoro et al., 2017). The results presented in this section further confirm the previous reports of AOA biogeography and genomic inventory. Please reference previous studies. We now reference Qin et al. 2020 ISMEJ as an example in the main text, as this study investigated marine AOA composition globally.

Line 190-191: Consider mentioning the reason for specifically investigating the *dur3* urea transporter, since AOA encode three types of urea transporters (*dur3*-type, ATP dependent, and *yut*-type). We have adjusted the text to clarify that we investigated *dur3* as it is the most widespread urea transporter in AOA.

Line 197: Additional organisms contain ureases and urea transporters. Please include other organisms in the relative abundance comparisons in Figure 2c-d. We now specifically mention that in the Gulf of Mexico (unlike the Angola Gyre and the Black Sea), most *ureC* reads are affiliated with non-AOA, who also dominate *ureC* transcription, and also early on refer to Supplementary Figure 5b. However, we have refrained from including this data in a main text figure, as we feel it is not a key point to this manuscript.

Line 197: The *ureC* RPKM values presented in Extended Data Figure 5b suggest the Gulf of Mexico AOA population must compete with Cyanobacteria and Proteobacteria for urea. The competition for urea may influence the indirect urea utilization presented in Figure 2b. The Angola Gyre and Black Sea AOA are the dominant urea hydrolyzers and presumably experience minimal urea competition. It is possible these AOA may co-consume ammonia and urea due to the slow growth rates measured with NanoSIMS. It may be worth mentioning that urea competition may influence the observed AOA ecophysiology. The reviewer correctly points out that in the Gulf of Mexico, *ureC* is predominantly encoded and transcribed by microorganisms other than AOA. Additionally, overall *ureC* transcription is very low compared to the other sampled environments, which we interpret as urea simply not being a major N-source used by the microbial community in this ammonia-rich environment. We now address this in the revised manuscript (L288-291).

Line 197: The ratio of urease (*ureC*) and the *dur3*-type urea transporter to the ammonia monooxygenase across AOA lineages is presented in lines 193-211, yet the data is in Extended Data Figure d-g. Consider moving this figure into the main text. We have considered moving this figure to the main text, but find that it does not contribute enough to the main message of the manuscript to include it there.

Line 212: The combined CARD-FISH and NanoSIMS dataset used to evaluate the nitrogen assimilation strategy across marine AOA lineages makes this manuscript unique and furthers

our understanding of AOA ecophysiology. The manuscript should be centered around the single-cell N-assimilation measurements of AOA in contrasting marine environments, instead of presenting it at the end.

Thank you for this comment. We think it is important to first establish that only looking at the water mass as a whole (i.e. by measuring bulk oxidation rates and applying -omics approaches) does not help us in understanding the mechanisms of who is using urea. Only when combined with single cell measurements, the observed patterns start to make sense. We have restructured the abstract and the introduction, as well as made changes throughout the manuscript to better highlight the novelty of our study.

Line 236: Please maintain a consistent y-axis across microorganisms to ensure easy visual comparisons. It may be worthwhile to present the data on a log axis given the order of magnitude difference between the *Nitrosopelagicus* WCA and *Nitrosopumilus* 15N biomass enrichments in Figure 3a-b.

We have tried multiple options of data depiction for this particular plot and think that this best depicts the data. We find that a log scale plot is harder to read and thus obscures the main finding.

Line 236: Please improve contrast of the 15N biomass enrichments in Figure 3a-b to make it easier to distinguish single-cells from the filter background.

We agree that the contrast of the *Nitrosopelagicus* cells is not very high, however the contrast has already been improved by adding an alpha mask (see Methods). The enrichment of *Nitrosopelagicus* cells is relatively close to the natural abundance and because we find it important to also show this visually, we opt for not manipulating the contrast any further.

Line 240-252: The absolute 15N-urea biomass enrichment values in Figure 3 indicate *Nitrosopumilus* and other microorganisms assimilated more urea than *Nitrosopelagicus*. The higher enrichment suggests *Nitrosopumilus* and other microorganisms have higher cell-specific urea assimilation rates and will outcompete *Nitrosopelagicus* in the absence of ammonium. However, the dual substrate (urea + NH₄⁺) dataset suggests *Nitrosopumilus* and other microorganisms repress urea assimilation in the presence of ammonium while *Nitrosopelagicus* continued to assimilate urea at similar rates. The dataset suggests *Nitrosopelagicus* do not have a distinct metabolite preference and may co-consume urea and ammonium, similar to the marine Gammaproteobacteria AOB *Nitrosococcus oceanus* (Qin et al. 2024). Although an additional dual substrate incubation with 15N-NH₄⁺ would be required to confirm this hypothesis. Please comment on this shortfall.

We agree with the reviewer that *Nitrosopelagicus* appears to not have a distinct metabolite preference on the population level. The question of co-consumption of ammonium and urea by the same cell is interesting, but as pointed out, cannot be fully solved by our experiments. To achieve this would require comparison of single cell N-assimilation in additional experiments where both 15N-ammonium + 15N-urea are supplied, in comparison to single labeled experiments. In such experiments, co-consumption would lead to higher 15N enrichment of individual cells in 15N-ammonium + 15N-urea incubations in comparison to single labeled incubations.

In response to this comment, we have now changed the last sentence of this paragraph in the main text (former Line 256, now L288-291).

Line 240-252: Consider comparing urea utilization under the dual substrate incubation across lineages. It appears that *Nitrosopelagicus* may have the greatest assimilation rate under the dual substrate condition. Therefore, *Nitrosopelagicus* may be the predominant urea-oxidizer in the presence of ammonia and urea, which could then explain the direct urea-oxidation measurements in Figure 2b.

It is true that when supplied with ^{15}N -urea and ^{14}N -ammonium *Nitrosopelagicus* is higher enriched in $^{15}\text{N}/(^{14}\text{N}+^{15}\text{N})$ compared to *Nitrosopumilus* and other microorganisms (see Figure A below). We also consider it likely that the consistent urea oxidation that we observe in the incubation amended with ^{14}N -ammonium is performed by *Nitrosopelagicus*.

Figure A: Urea assimilation in the presence of excess ammonium by *Nitrosopelagicus*, *Nitrosopumilus* and other microorganisms. $^{15}\text{N}/(^{14}\text{N} + ^{15}\text{N})$ enrichment of *Nitrosopelagicus* WCA (magenta), *Nitrosopumilus* (blue) and other microorganisms (gray) after incubation with ^{15}N -urea + ^{14}N -ammonium. Boxplots depict the 25–75% quantile range, with the center line depicting the median (50% quantile); whiskers encompass data points within 1.5x the interquartile range. ^{15}N at% enrichment of the substrate pools (ammonium or urea, respectively) was >97%. Note that *Nitrosopumilus* and *Nitrosopelagicus* cells were targeted in separate CARD-FISH NanoSIMS analyses, and thus, the category “other microorganisms” may contain some (not-targeted) *Nitrosopelagicus* and *Nitrosopumilus* cells, respectively. n = number of measurements per group and treatment with number of cells enriched significantly above the detection limit in brackets. *** indicates significant difference between treatments in Mann-Whitney-Wilcoxon tests. *Nitrosopelagicus* was significantly higher enriched in comparison to *Nitrosopumilus* ($W = 452$, P value = 7.5×10^{-4}).

Line 256: Please remove the statement “thus essentially doubling its growth rate”. The data presented in this manuscript does not demonstrate the growth rate of *Nitrosopelagicus* was doubled via urea and ammonium co-consumption. Previous observations of co-consumption by *Nitrosococcus oceanii* did not report a statistical difference in the growth rate while consuming both ammonia and urea (Qin et al., 2024).

Although we do not know whether a single *Nitrosopelagicus* cell is using ammonium and urea simultaneously, the growth rate of the *Nitrosopelagicus* population could be doubled when both ammonium and urea are available. We changed this in the manuscript accordingly.

Line 258-260: This statement disregards the influence of metabolite concentrations. Please revise the statement.

We have considered revising this concluding paragraph, but as we discuss metabolite concentrations in detail in the main manuscript, we feel it is not needed to bring this up again at this point.

Line 263-264: The underestimation of marine nitrification by approaches only considering ammonia-oxidation has been reported before. Please differentiate this study by highlighting single-cell N-assimilation across AOA lineages.

We adjusted the wording accordingly (L305-307).

Line 276-277: Please remove the statement “allowing them to double their growth rates”. See comments about Line 256.

We rephrased this sentence. It now says “Furthermore, although we cannot unequivocally show that *Nitrosopelagicus* cells were assimilating ammonium and urea simultaneously, if they do, then this strategy could potentially lead to a doubling of their growth rate when both ammonium and urea are available”. Now L288-291.

Line 281-282: The demonstration of preferred urea utilization by *Nitrosopelagicus* and concluding urea utilization could differentiate the niche of marine AOA lineages is novel. This finding should be highlighted throughout the manuscript, including the introduction and abstract. Done as suggested.

Line 288-290: Consider softening the statement “we hypothesize that *Nitrosopelagicus* obtains ammonium from other DON compounds”. The utilization of DON compounds by ammonia-oxidizers has been limited to urea, cyanate, and guanidine in isolates and in-situ utilization of glutamate has been demonstrated.

We have considered revising this statement, but taking into account that ammonia oxidizers have been shown to use urea, cyanate, polyamines and (as recently discovered) guanidine, we think that there might be additional DON compounds marine AOAs are using. As we explicitly state that we are hypothesising here, we have not adjusted this statement.

Line 325-328: Explicitly mention analytical methods used to measure metabolites in manuscript. We now explicitly mention the used analytical methods, which are all well established standard protocols to measure urea, ammonium, nitrite and nitrate.

Line 330-331: A brief overview of the experimental methods from the Gulf of Mexico cruise should be presented here.

We now briefly mention how incubations were done in the Gulf of Mexico cruise. However, we feel due to space constraints a more elaborate description of previously published protocols is not needed.

Line 337: Different process rate methods were used for the Gulf of Mexico, Angola Gyre, and Black Sea. Please elaborate on differences between methods.

Please refer to our detailed explanation of how the setups differed and which differences we consider to have an impact in our answer to the reviewer's major points above. Briefly, all protocols followed established incubation protocols and were chosen for each cruise taking into consideration the *in situ* conditions (e.g. oxygen concentrations), the feasibility and also the compatibility with other projects/experiments that were run on the same cruise but are not presented here.

Line 338-343: Sampling depths, storage duration prior to experiment, and light exposure are not explicitly mentioned. Please consider including.

We now specifically mention storage time and light exposure in the methods section (L379-380). Sampling depths are already specified in Supplementary Data 1.

Line 353-355: State replication used in tracer experiment.

We have now added the replication in the methods. Per time point, one biological replicate was sacrificed.

Line 360-361: It is unclear how the water column oxygen concentrations correspond to the ambient oxygen concentrations in the Schott bottle. Please elaborate on choice.

In the Angola Gyre, ambient oxygen concentrations were $>60 \mu\text{M}$ at all incubation depths. Thus we chose to incubate in Schott bottles as no oxygen manipulations were required. We have added this to the text at L400-403.

Line 363: Duplicate samples were collected for each condition. Please elaborate on using duplicate as opposed to triplicate samples for downstream statistical analysis.

In the Angola Gyre cruise, this was done due to time, water, and incubator space constraints, which precluded us from performing triplicate incubations. This information is now also included in the methods section.

Line 370: Please mention the isotope supplementation concentration was chosen taking *in situ* concentrations into considering and wanting the overall ^{15}N at% to be greater than 90% in the main text.

We respectfully disagree that this is needed within the main text, however we have included the rationale in the methods and made a clearer reference to them as well as the Supplementary Data 1, where the measured labelling percentages are given.

Line 373-388: Please include all rate calculation equations in main text or supplement to this manuscript.

As suggested, we have now included all rate calculations in the Supplementary Methods of the manuscript.

Line 392-397: Please elaborate on the use of different methods to quantify initial ^{15}N -urea on the Black Sea and Angola cruise.

We are confident that both methods for determining labeling percentage of the substrate pool are well suited to yield robust and accurate results for our experimental setups. In the Black

Sea, *in situ* urea concentrations and urea tracer additions were sufficiently high to confidently quantify with spectrophotometric methods. For the Angola Gyre, the tracer additions were lower, and thus their quantification via the more sensitive IRMS-based approach was better suited.

Line 397: Consider a paragraph break to separate the statistical analysis from the methods.
Done as suggested.

Line 409-412: Setting the non-significant slopes to the limit of detection could lead to an overestimate of the measured rate, especially if the true rate is zero. Consider elaborating on this choice.

We find it unlikely that when one biological replicate shows a significant rate, the true rate in a replicate incubation is zero. Instead, there appears to be some biological variability which leads to some replicates falling below the detection limit in the Angola Gyre samples, especially as all rates measured there are very low. Given that our detection limits are close to zero (0.017, 0.040, 0.146 nM day⁻¹), and that we observe some biological variability, we think that averaging by considering non-significant rates as detection limit (which we opt for), gives us a more realistic averaged rate in comparison to setting non-significant rates to zero. However, we would also like to emphasize that if both replicate rates were non-significant we considered the rate to be zero.

Line 443: The contig assembly and read mapping methods are not explicitly stated here. Please update this section to include package and settings as done in ref. 54. It is implied the metagenomic and metatranscriptomic abundance was evaluated with a read-based method instead of a MAG-based method. However, the method for calculation abundance should be explicitly mentioned.

We updated this section and now explicitly list the versions for the programs used, all further details regarding the scripts used are found on github (https://github.com/dspeth/aoa_urea/tree/main/marker_gene_databases), which we also reference in the text. Additionally, we clarified the calculation of the abundance (RPKM) calculation. We would like to clarify that for assessing abundance of the key genes (*amoA*, *ureC*, *dur3*) pertaining to AOA, we did not use any assembled metagenomic data, but directly used the trimmed and quality filtered reads. Metagenome assemblies were only used to obtain longer sequences of key genes to calculate phylogenetic trees, and bin metagenome assembled genomes. This latter procedure is detailed in the Supplementary Methods.

Line 444-446: It is not clear how marker gene abundance was assessed until line 477. The database curation method presented in lines 447-476 is important, but could be moved to supplement and referenced in the main text.

As suggested, we moved the method section on database curation to the Supplementary Methods and referenced this in the main text.

Line 480: Please define BSR as bit-score ratio when it is first introduced on line 480
The BSR approach is introduced the first time it is mentioned as “BLAST score ratio (BSR) approach” (L526, previously L458).

Line 595: Please include equations used to calculate cellular ^{15}N -atom% excess and growth rate in the main text or supplement of this manuscript.

We now include the equations in the Supplementary Information of the manuscript.

References

- Arandia-Gorostidi, N., A. L. Jaffe, A. E. Parada, B. J. Kapili, K. L. Casciotti, R. S. R. Salcedo, C. M. J. Baumas and A. E. Dekas (2024). "Urea assimilation and oxidation support activity of phylogenetically diverse microbial communities of the dark ocean." The ISME Journal **18**(1).
- Bristow, L. A., T. Dalsgaard, L. Tiano, D. B. Mills, A. D. Bertagnolli, J. J. Wright, S. J. Hallam, O. Ulloa, D. E. Canfield, N. P. Revsbech and B. Thamdrup (2016). "Ammonium and nitrite oxidation at nanomolar oxygen concentrations in oxygen minimum zone waters." Proc Natl Acad Sci U S A **113**(38): 10601-10606.
- De Brabandere, L., B. Thamdrup, N. P. Revsbech and R. Foadi (2012). "A critical assessment of the occurrence and extend of oxygen contamination during anaerobic incubations utilizing commercially available vials." Journal of Microbiological Methods **88**(1): 147-154.
- Kitzinger, K., C. C. Padilla, H. K. Marchant, P. F. Hach, C. W. Herbold, A. T. Kidane, M. Konneke, S. Littmann, M. Mooshammer, J. Niggemann, S. Petrov, A. Richter, F. J. Stewart, M. Wagner, M. M. M. Kuypers and L. A. Bristow (2019). "Cyanate and urea are substrates for nitrification by Thaumarchaeota in the marine environment." Nat Microbiol **4**(2): 234-243.
- Qin, W., S. P. Wei, Y. Zheng, E. Choi, X. Li, J. Johnston, X. Wan, B. Abrahamson, Z. Flinkstrom, B. Wang, H. Li, L. Hou, Q. Tao, W. W. Chlouber, X. Sun, M. Wells, L. Ngo, K. A. Hunt, H. Urakawa, X. Tao, D. Wang, X. Yan, D. Wang, C. Pan, P. K. Weber, J. Jiang, J. Zhou, Y. Zhang, D. A. Stahl, B. B. Ward, X. Mayali, W. Martens-Habbena and M.-K. H. Winkler (2024). "Ammonia-oxidizing bacteria and archaea exhibit differential nitrogen source preferences." Nature Microbiology.
- Qin, W., Y. Zheng, F. Zhao, Y. Wang, H. Urakawa, W. Martens-Habbena, H. Liu, X. Huang, X. Zhang, T. Nakagawa, D. R. Mende, A. Bollmann, B. Wang, Y. Zhang, S. A. Amin, J. L. Nielsen, K. Mori, R. Takahashi, E. Virginia Armbrust, M. H. Winkler, E. F. DeLong, M. Li, P. H. Lee, J. Zhou, C. Zhang, T. Zhang, D. A. Stahl and A. E. Ingalls (2020). "Alternative strategies of nutrient acquisition and energy conservation map to the biogeography of marine ammonia-oxidizing archaea." ISME J **14**(10): 2595-2609.
- Santoro, A. E., M. A. Saito, T. J. Goepfert, C. H. Lamborg, C. L. Dupont and G. R. Ditullio (2017). "Thaumarchaeal ecotype distributions across the equatorial Pacific Ocean and their potential roles in nitrification and sinking flux attenuation." Limnology and Oceanography **62**(5): 1984-2003.
- Wan, X. S., H. X. Sheng, H. Shen, W. Zou, J. M. Tang, W. Qin, M. Dai, S. J. Kao and B. B. Ward (2024). "Significance of Urea in Sustaining Nitrite Production by Ammonia Oxidizers in the Oligotrophic Ocean." Global Biogeochemical Cycles **38**(10).

Xu, M. N., X. Li, D. Shi, Y. Zhang, M. Dai, T. Huang, P. M. Glibert and S. J. Kao (2019).
"Coupled effect of substrate and light on assimilation and oxidation of regenerated nitrogen in
the euphotic ocean." Limnology and Oceanography **64**(3): 1270-1283.

Zhao, Z., C. Amano, T. Reinthaler, F. Baltar, M. V. Orellana and G. J. Herndl (2024).
"Metaproteomic analysis decodes trophic interactions of microorganisms in the dark ocean."
Nature Communications **15**(1).

Answers to Reviewers:

Black initial reviewer remarks

Blue our first answer

Red answering reviewer remarks

Purple our final answer

Reviewer #3 (Remarks to the Author):

Overall judgment: The manuscript Urea use drives niche separation between dominant marine ammonia oxidizing archaea propose ecological niches for Nitrosopelagicus and Nitrosopumilus — two genera of marine ammonia-oxidizing archaea (AOA) — based on single-cell assimilation of ammonium and urea in the Black Sea. Previous work from the authors has demonstrated distinct ammonium (NH_4^+) and organic nitrogen utilization strategies by AOA and nitrite-oxidizing bacteria (NOB) in the Gulf of Mexico (Kitzinger et al., 2020, 2019). The dominance of Nitrosopelagicus ammonia oxidizing archaea (AOA) in the open ocean and Nitrosopumilus AOA in estuarine areas has been well documented (Qin et al., 2020; Santoro et al., 2017). However, physiological traits defining the ecological niche of these ubiquitous marine AOA has remained elusive. Demonstrating distinct nitrogen metabolite preferences dictates the environmental niche of these marine AOA genera — as was recently demonstrated for the major ammonia-oxidizing microorganism lineages (Qin et al., 2024) — would represent a significant contribution to the field. All experimental and computational approaches were sufficient. However, specific techniques and software packages should be explicitly stated in the methods section and there were major concerns listed below. Care should be taken to update the introduction and discussion to improve clarity. The manuscript provides compelling evidence of genus-level niche differentiation between marine AOA and could shape our understanding of the marine nitrogen cycle. Despite presenting compelling evidence for the distinct metabolite preferences of Nitrosopelagicus and Nitrosopumilus in the Black Sea, this manuscript would have benefited by constraining the scope to the Black Sea and linking the single-cell observations with the environmental rate and metaomic datasets. The most significant findings of this worked can be captured as following:

Significance #1: Combining single-cell N-assimilation measurements with in-situ genus level phylogeny in samples collected from the Black Sea enabled the authors to present compelling evidence of the distinct strategies Nitrosopelagicus and Nitrosopumilus to employ their ammonia and urea utilization (Figure 3). Their results suggest Nitrosopumilus prefer to utilize ammonia and repress urea utilization in the presence of ammonia, while Nitrosopelagicus did not demonstrate a distinct preference toward ammonia or urea. Despite preferring ammonia, the ^{15}N -urea biomass enrichment — and the calculated growth rate — of Nitrosopumilus and all other microorganisms appear to be faster than Nitrosopelagicus when ^{15}N -urea was supplemented. Yet, Nitrosopelagicus may outcompete Nitrosopumilus for urea in the presence of excess ammonia based on higher ^{15}N urea biomass enrichments in samples supplemented with ^{15}N -urea and ^{14}N - NH_4^+ . Considering all marine environments are characterized by both ammonium and urea, it would suggest the niche of Nitrosopelagicus is defined by urea-utilization in ammonium replete marine environments.

We thank the reviewer for their kind assessment.

We thank the authors for updating the manuscript and your thorough response and recognize a significant amount of work went into reframing the abstract and introduction to highlight the niche differentiation between *Nitrosopelagicus* and *Nitrosopumilus*. The manuscript now builds from bulk to single-cell measurements to provide insight into the abundance of *Nitrosopelagicus* in ammonia-replete waters with high urea to ammonia ratios. This work provides fundamental insights into the ecophysiology of *Nitrosopelagicus* AOA and will be a major contribution to the field.

The authors have done an excellent job addressing our comments. There are a few minor recommendations below with the primary recommendation in the major comment 3 response. It has been a pleasure to review this manuscript.

We thank the reviewers once again for their assessment and agree that this review process has significantly improved the manuscript. In the following we will only provide answers to the open comments as the other comments have been sufficiently discussed.

3. The caveats associated with varying experimental methods across samples and the impact on the subsequent statistical comparisons needs to be discussed. Different methods were used to measure oxidation rates with stable isotope tracers on the Gulf of Mexico, Black Sea, and Angola Gyre cruises. There were also experimental variations between sampling depths on Black Sea cruise. Specific differences include the incubation bottles (serum bottle & exetainer), oxygen addition methods (in-situ & He/N₂ sparge, or sparge & O₂ addition), supplemented metabolites (N₂O & DIC), supplemented metabolite concentration, and supplemented isotope percentages differed between the cruises (lines 338-412). A primary concern is the influence of tracer and oxygen additions on the measured rates, which is not discussed in the manuscript. A discussion about methodological differences and subsequent statistical analysis is requested.

The experimental setup was adjusted during each cruise to match in situ conditions as accurately as possible while also accommodating best feasibility. The ¹⁵N tracer additions differed due to the different nutrient availability across sites. Similarly, the oxygen concentrations in the incubations were adjusted to match in situ oxygen conditions. By mimicking the respective in situ conditions we anticipate that our potential oxidation rates reflect the actual rates as closely as possible. The incubation period is short (<24h) and the rates are linear from the beginning of the incubation on, without a lag phase, which indicates that ammonia oxidizers are active in situ. While experimental details differed between cruises, the association of *Nitrosopelagicus* with enhanced urea use holds across our study sites, and also other studies in open ocean settings where *Nitrosopelagicus* typically dominates (Santoro et al. 2017 L&O, Wan et al. 2024, GBC). Thus, we are confident that these minor differences in protocols did not impact the outcome of our experiments, or our conclusions. To address this issue, we added a section to the Supplementary Discussion.

We thank the authors for this clarification and including additional details in the Supplementary Discussion. Our primary concern was comparing oxidation rates from an aerobic Schott bottle with a hypoxic exetainer/serum bottle. It was unclear to us if the 500 mL Schott bottles used for the urea incubations with added ammonium pool in the Black Sea were sealed with butyl rubber stoppers in L377-378, especially considering the Schott bottles used in the Angola Gyre were aerobic. However, ref 43 and L380-382 indicate the Black Sea Schott bottles were sealed, sparged, and amended with O₂. Consider removing the description of the urea incubations with

added ammonium pool in the Black Sea from the parentheses and mentioning that both bottles were sealed with butyl rubber stoppers.

We thank the reviewers for explaining their misunderstanding. We modified the description in the methods section of the manuscript and hope that this makes it easier to understand. Now it says: “. Incubations were set up headspace-free in 250 mL glass serum bottles or 500 mL Schott bottles (for urea incubations with added ammonium pool) and in both cases closed with degassed rubber stoppers as described by ref. 42 and 43, and transferred to a light protected cooler until further processing (within six hours).”

The response and description the authors added in L401-403 explaining the rationale for using a Schott bottle in the Angola Gyre address our concerns about the methodological differences. We agree that the authors were able to obtain oxidation rates reflective of actual oxidation rates by mimicking in situ concentrations and the minor differences did not impact experimental outcomes. Upon reading the response of the authors, we see rational for the tracer addition at each site and station was mentioned in L411-414. We had incorrectly assumed the description of “all three environments” in L411-414 was describing the 3 sample sites in Kitzinger et al. 2019 based on the structure of the Process rate experiment methods section. Our apologies if this caused confusion. Consider moving L410-414 and portions of the reviewer response to the beginning of the Process rate experiment methods section before describing site specific methods. This will help to avoid confusion with the methods from Kitzinger et al. 2019, especially for readers unfamiliar with the techniques the authors employed.

Thank you again for explaining the misconception. We chose to keep this description where it is, as it seems out of place at the beginning of the paragraph. Additionally we think that the comprehensive overview of tracer additions in Supplementary Table 1 helps the reader to understand the method.

5. There are distinct metabolite distributions with depth presented in Extended Data Figure 1, which are not captured in Figure 1b-d or Figure 2a-b. The ammonium concentrations detected in the Black Sea samples are near the limit of detection until ~120m where the ammonium concentrations increase up to $7.4 \mu\text{M-NH}_4^+$. The distribution of AOA in the Black Sea samples also appears to have an inverse relationship with ammonia availability based on the Thermoproteota ureC RPKM dataset presented in Extended Data Figure 5b. Please, consider presenting the nutrient distributions collected at each sampling station, depth integrated nutrient inventories, or boxplots corresponding to specific depths within the sample site to characterize the environmental gradient within the Black Sea.

We thank the reviewer for the suggestion. We now present ammonium concentration profiles along with ammonia and urea oxidation rates and relative AOA abundance from the Black Sea in Supplementary Figure 2. We also refer to this depth distribution in the main text e.g. L115-116.

We thank the authors for adding depth data from the Black Sea to Supplementary Figure 2. we noticed the figure label only mentions the Angola Gyre. Consider mentioning the Black Sea in the figure label or consider making Supplemental Figure 2h-m into a new Supplemental Figure.

We apologize for this oversight and have added the Black Sea to the figure description of Supplementary Figure 2.

Line 240-252: Consider comparing urea utilization under the dual substrate incubation across lineages. It appears that Nitrosopelagicus may have the greatest assimilation rate under the dual substrate condition. Therefore, Nitrosopelagicus may be the predominant urea-oxidizer in the presence of ammonia and urea, which could then explain the direct urea-oxidation measurements in Figure 2b.

It is true that when supplied with ^{15}N -urea and ^{14}N -ammonium Nitrosopelagicus is higher enriched in $^{15}\text{N}/(^{14}\text{N}+^{15}\text{N})$ compared to Nitrosopumilus and other microorganisms (see Figure A below). We also consider it likely that the consistent urea oxidation that we observe in the incubation amended with ^{14}N -ammonium is performed by Nitrosopelagicus.

We thank the authors for including a comparison of the $^{15}\text{N}/(^{14}\text{N}+^{15}\text{N})$ enrichment across organisms in the ^{15}N -urea and ^{14}N -ammonium incubation. It was difficult to compare these enrichments in Fig 3c. I find this figure highly compelling. The significantly higher enrichment of Nitrosopelagicus indicates it is the predominant urea-oxidizer in the presence of ammonia and urea, which could then explain the direct urea-oxidation measurements. It would be nice to see this figure in the manuscript. However, the decision should be at the discretion of the authors.

We agree that this figure is highly compelling. However we chose to not include it in the manuscript as it is just a different way of depicting data that is shown in Figure 3, Supplementary Figure 6 and provided in Supplementary Data 5.

Overall judgment: The manuscript *Urea use drives niche separation between dominant marine ammonia oxidizing archaea* propose ecological niches for *Nitrosopelagicus* and *Nitrosopumilus* — two genera of marine ammonia-oxidizing archaea (AOA) — based on single-cell assimilation of ammonium and urea in the Black Sea. Previous work from the authors has demonstrated distinct ammonium (NH_4^+) and organic nitrogen utilization strategies by AOA and nitrite-oxidizing bacteria (NOB) in the Gulf of Mexico (Kitzinger et al., 2020, 2019). The dominance of *Nitrosopelagicus* ammonia oxidizing archaea (AOA) in the open ocean and *Nitrosopumilus* AOA in estuarine areas has been well documented (Qin et al., 2020; Santoro et al., 2017). However, physiological traits defining the ecological niche of these ubiquitous marine AOA has remained elusive. Demonstrating distinct nitrogen metabolite preferences dictates the environmental niche of these marine AOA genera — as was recently demonstrated for the major ammonia-oxidizing microorganism lineages (Qin et al., 2024) — would represent a significant contribution to the field. All experimental and computational approaches were sufficient. However, specific techniques and software packages should be explicitly stated in the methods section and there were major concerns listed below. Care should be taken to update the introduction and discussion to improve clarity.

The manuscript provides compelling evidence of genus-level niche differentiation between marine AOA and could shape our understanding of the marine nitrogen cycle. Despite presenting compelling evidence for the distinct metabolite preferences of *Nitrosopelagicus* and *Nitrosopumilus* in the Black Sea, this manuscript would have benefited by constraining the scope to the Black Sea and linking the single-cell observations with the environmental rate and meta-omic datasets.

The most significant findings of this worked can be captured as following:

Significance #1: Combining single-cell N-assimilation measurements with *in-situ* genus level phylogeny in samples collected from the Black Sea enabled the authors to present compelling evidence of the distinct strategies *Nitrosopelagicus* and *Nitrosopumilus* to employ their ammonia and urea utilization (Figure 3). Their results suggest *Nitrosopumilus* prefer to utilize ammonia and repress urea utilization in the presence of ammonia, while *Nitrosopelagicus* did not demonstrate a distinct preference toward ammonia or urea. Despite preferring ammonia, the ^{15}N -urea biomass enrichment — and the calculated growth rate — of *Nitrosopumilus* and all other microorganisms appear to be faster than *Nitrosopelagicus* when ^{15}N -urea was supplemented. Yet, *Nitrosopelagicus* may outcompete *Nitrosopumilus* for urea in the presence of excess ammonia based on higher ^{15}N -urea biomass enrichments in samples supplemented with ^{15}N -urea and ^{14}N - NH_4^+ . Considering all marine environments are characterized by both ammonium and urea, it would suggest the niche of *Nitrosopelagicus* is defined by urea-utilization in ammonium replete marine environments.

Shortfall #1: Unfortunately, the compelling single-cell analysis from the Black Sea could not be extended to the oligotrophic Angola Gyre where *Nitrosopelagicus* comprised a larger proportion of the nitrifier community.

Significance #2: The authors also characterize the relative contribution of ammonium and urea to ammonia oxidation in three distinct marine environments with unique productivity regimes (Figure 1 & 2).

Shortfall #2: Though presented as a novel finding in the manuscript, the increased contribution of urea relative to NH_4^+ to nitrification in the open ocean has been reported several times before and does not challenge the current understanding of marine nitrification (Arandia-Gorostidi et al., 2024; Santoro et al., 2017; Tolar et al., 2017; Wan et al., 2024). The distribution of urea-dependent ammonia-oxidation (hereafter urea-oxidation) and ammonia-oxidation is well known at the global scale. The urea-dependent ammonia-oxidation rate and ammonia-oxidation rate decrease from copiotrophic eutrophic coastal zones to the oligotrophic open ocean while the contribution of urea to total ammonia-oxidation increases due to decreasing substrate concentrations (Wan et al., 2024). The authors confirm the validity of this framework in non-polar marine regions. However, the isotopic rates measurements were not paired with single-cell analysis, metagenomics, and metatranscriptomics to study the niche differentiation between of *Nitrosopelagicus* and *Nitrosopumilus*.

Major Points:

1. The distribution of urea-oxidation and ammonia-oxidation in marine environments is well known at the global scale. Recent studies have also demonstrated the contribution of urea and ammonia to total ammonia oxidation by mesopelagic and bathypelagic AOA is nearly equivalent (Arandia-Gorostidi et al., 2024; Laperriere et al., 2021; Santoro et al., 2017; Tolar et al., 2017; Wan et al., 2024). Therefore, the contribution of urea to nitrification has been demonstrated and the rate measurements presented here do not represent a unique or novel finding. It is important that the authors accurately represent our current understanding.
2. In our opinion the most significant claim in the manuscript is urea utilization in the presence of ammonium defines the niche of *Nitrosopelagicus*. However, this finding is not emphasized throughout the manuscript and must be strengthened as a result. The authors should consider framing the manuscript around this finding as it might explain the distribution of *Nitrosopelagicus* in the open ocean. Compelling evidence that *Nitrosopelagicus* AOA are responsible for urea hydrolysis in the Black Sea or Angola Gyre is not presented and would serve as an indicator of their distinct preference for urea-derived ammonium over ammonium if presented. Neither the environmental rate data in *Figure 2a-b* nor the meta-omic data in *Figure 2c-d* and *Extended Data Fig. 5* contribute to defining the niche of *Nitrosopelagicus*. Focusing on the depth resolved samples collected from the Black Sea instead of clustering all samples together may provide additional insight and it is suggested that the authors provide these additional data sets. Please comment on how the oxidation rates and meta-omic sampling contribute to understanding the environmental niche of *Nitrosopelagicus* and *Nitrosopumilus* in the Black Sea.

3. The caveats associated with varying experimental methods across samples and the impact on the subsequent statistical comparisons needs to be discussed. Different methods were used to measure oxidation rates with stable isotope tracers on the Gulf of Mexico, Black Sea, and Angola Gyre cruises. There were also experimental variations between sampling depths on Black Sea cruise. Specific differences include the incubation bottles (serum bottle & exetainer), oxygen addition methods (in-situ & He/N₂ sparge, or sparge & O₂ addition), supplemented metabolites (N₂O & DIC), supplemented metabolite concentration, and supplemented isotope percentages differed between the cruises (lines 338-412). A primary concern is the influence of tracer and oxygen additions on the measured rates, which is not discussed in the manuscript. A discussion about methodological differences and subsequent statistical analysis is requested.
4. Ammonium concentrations classically define the environmental niche of ammonia-oxidizing microorganisms. A recent study found the ammonia affinity was higher than the urea affinity in the open ocean providing an explanation for the distribution of urea in this environment (Wan et al., 2024). However, a discussion about the impact of concentrations on niche differentiation is lacking in the current manuscript, especially while discussing direct versus indirect urea oxidation rates (*Figure 2b*). For example, the axes for the Gulf of Mexico are an order of magnitude greater than the other sampling sites in *Figure 2b* making it difficult to make a direct comparison between the sampling sites. It is apparent that the addition of ammonia significantly decreased the urea oxidation rate in the Gulf of Mexico, while the tracer has no impact in the Black Sea and Angola Gyre. However, the Black Sea and Angola Gyre urea oxidation rates are an order of magnitude lower than the presented Gulf of Mexico rates. The different responses may be attributed to the lower *Nitrosopumilus* (0.04 d⁻¹) and *Nitrosopelagicus* (0.002 d⁻¹) NanoSIMS-based growth rates in the Black Sea (line 229-236) compared to the Gulf of Mexico (0.23 d⁻¹). The faster growth rate may enable the Gulf of Mexico AOA population to transition their metabolism away from urea-oxidation over the 24 hour isotope incubations. The Black Sea and Angola Gyre AOA populations may also transition their metabolism towards ammonia-oxidation if the incubation times were corrected for the AOA growth rates. Therefore, we suggest that you consider discussing the impact of concentration and growth rates on direct versus indirect urea oxidation rates.
5. There are distinct metabolite distributions with depth presented in *Extended Data Figure 1*, which are not captured in *Figure 1b-d* or *Figure 2a-b*. The ammonium concentrations detected in the Black Sea samples are near the limit of detection until ~120m where the ammonium concentrations increase up to 7.4 μM-NH₄⁺. The distribution of AOA in the Black Sea samples also appears to have an inverse relationship with ammonia availability based on the *Thermoproteota ureC* RPKM dataset presented in *Extended Data Figure 5b*. Please, consider presenting the nutrient distributions collected at each sampling station, depth integrated nutrient inventories, or boxplots corresponding to specific depths within the sample site to characterize the environmental gradient within the Black Sea.

6. The difference in AOA cell concentrations might contribute to the rate measurements presented in this manuscript. Please discuss how bulk rates are dependent on cell concentrations and cell specific rates. This is important because the two order of magnitude difference might be attributed to an order of magnitude difference in cell concentration and an order of magnitude difference in the cell specific rate. Additionally, consider presenting the ammonia-oxidizer cell abundance either in the main text, extended data, or supplemental material.
7. The axes limits are inconsistent between conditions in many figures making direct comparison difficult. This is especially apparent for the ^{15}N biomass enrichment data of *Nitrosopelagicus* presented in *Figure 3c*, which is an order of magnitude lower than other genera. Therefore, we recommend adjusting the axis accordingly.
8. Please remove the statement “*thus essentially doubling its growth rate*” in line 256. The data presented in this manuscript does not demonstrate the growth rate of *Nitrosopelagicus* was doubled via urea and ammonium co-consumption. Previous observations of co-consumption by *Nitrosococcus oceani* did not report a statistical difference in the growth rate while consuming both ammonia and urea. The statement is not sufficiently supported by the experimental results presented in *Figure 3* and should be removed from the manuscript.

Minor Points (line-by-line):

- Line 29-31: The increased contribution of urea relative to NH_4^+ to nitrification in the open ocean has been reported several times before (Arandia-Gorostidi et al., 2024; Tolar et al., 2017; Wan et al., 2024) and does not challenge the current understanding of marine nitrification. Please tone down novelty claims of your findings.
- Line 31-33: The dominance of *Nitrosopelagicus* ammonia oxidizing archaea (AOA) in the open ocean and *Nitrosopumilus* AOA in estuarine areas was previously reported by (Santoro et al., 2015) and Qin et al., 2020. However, physiological traits defining the ecological niche of these ubiquitous marine ammonia-oxidizing archaea have not yet been reported and would represent a major contribution to the field. Please revise manuscript to integrate this rationale.
- Line 33-36: The trend of increased urea-N oxidation relative to ammonia oxidation from eutrophic coastal zones to the oligotrophic open ocean was previously demonstrated by (Wan et al., 2024) in the Subtropical North Pacific. Their results were further validated by comparing the urea to NH_4^+ concentration and oxidation rate ratios across the coastal, epipelagic zone, and mesopelagic zone across the global ocean. Please tone down novelty claims of your findings.
- Line 40-41: AOA should be presented earlier in the introduction considering the hypothesis presented in the manuscript is the relative availability of ammonium versus urea drives niche differentiation between AOA the *Nitrosopelagicus* and *Nitrosopumilus* genus.
- Line 55-56: While there are still unresolved aspects about the factors shaping the distribution of AOA at the regional scale, the factors shaping the global distribution of urea and ammonia oxidation in the global ocean are known at the global scale (Wan et al., 2024). Although the urea-dependent ammonia-oxidation rate and ammonia-dependent ammonia-oxidation rate decrease from eutrophic coastal zones to the oligotrophic open ocean, the contribution of urea to total ammonia-oxidation increases due to substrate concentrations. Please update the statement in line 55-56 to reflect this understanding. It might be worthwhile to mention areas where this paradigm does not hold.
- Line 57: The mechanism of urea derived utilization should be mentioned while introducing urea as an AOA substrate. Urea must be transported into the cell where it is hydrolyzed by urease to ammonia and carbon dioxide. The intracellularly derived ammonia can then be utilized for ammonia-oxidation by the ammonia monooxygenase. Consider explicitly explaining this mechanism in the introduction.

- Line 58-59: Consider using a different reference to introduce urea utilization by marine AOA, since the contribution of urea to total ammonia-oxidation was minor as stated in lines 62-64. The contribution of urea was hypothesized based on genomic surveys (Alonso-Sáez et al., 2012) and the capacity of a marine AOA to utilize urea was demonstrated with the marine strain PS0 isolated from Puget Sound (Qin et al., 2014). The contribution of ammonium and urea to total ammonia oxidation were later compared across pelagic environments where it was found to have a major contribution in Antarctic coastal waters and a minor contribution in temperate coastal waters (Tolar et al., 2017).
- Line 62-66: Recent studies have demonstrated the contribution of urea and ammonia to total ammonia oxidation by mesopelagic and bathypelagic AOA is nearly equivalent (Arandia-Gorostidi et al., 2024; Laperriere et al., 2021; Santoro et al., 2017; Tolar et al., 2017; Wan et al., 2024). Therefore, the importance of urea no longer remains a hypothesis. Please update the statements in line 62 and 65-66 to reflect the current understanding.
- Line 66: The different AOA genera (i.e. *Nitrosopelagicus* and *Nitrosopumilus*) have not yet been introduced. Please refrain from mentioning AOA genera until they have been introduced. Consider moving statement anywhere after line 73.
- Line 72: Please mention carbon fixation pathway as it is important when comparing carbon assimilation across nitrifying lineages.
- Line 74-75: The word “many” is not quantitative and does not denote magnitude. Either remove statement or include percentage of *Nitrosopelagicus* and *Nitrosopumilus* containing urease and urea transport proteins.
- Line 80-83: This is the crux of the manuscript. The physiological mechanisms underpinning the distribution of *Nitrosopelagicus* and *Nitrosopumilus* remain unknown considering their genomic similarity. This should be highlighted earlier in the introduction, especially the lack of *in-situ* comparisons.
- Line 86-94: The use of the phrase “*environmental gradient ...*” can be easily misconstrued here. Samples were collected from three distinct marine environments, instead of sampling a transect defined by a nutrient gradient. The sampling sites provide a snapshot into the environmental factors driving niche differentiation there, but nothing about transitions between different environmental conditions. Please specify why each site was chosen and the comparisons that were made between sites.

- Line 86-94: Contribution of urea to nitrification along a gradient from eutrophic coastal waters to oligotrophic open oceans was recently demonstrated by (Wan et al., 2024). However, the isotopic rates measurements by Wan were not paired with single-cell imaging, metagenomics, and metatranscriptomics. This study provides new insights into the niche differentiation between of *Nitrosopelagicus* and *Nitrosopumilus*. Please focus the results on the niche differentiation.
- Line 94: Urea utilization by AOA in the Gulf of Mexico has already been published by the authors (Kitzinger et al., 2020, 2019). Results from the Black Sea and Angola Gyre cruise should be compared to the Gulf of Mexico cruise. Consider focusing novelty claims on the results of the Black Sea and Angola Gyre cruise and referencing the published Gulf of Mexico dataset.
- Line 106-109: There are distinct metabolite distributions with depth presented in *Extended Data Figure 1*, which are not captured in Figure 1b-d. The ammonium concentrations detected in the Black Sea samples are near the limit of detection until ~120m where the ammonium concentrations increase up to 7.4 $\mu\text{M-NH}_4^+$. The distribution of AOA in the Black Sea samples also appears to have an inverse relationship with ammonia availability based on the *Thermoproteota ureC* RPKM dataset presented in *Extended Data Figure 5b*. Consider presenting the nutrient distributions collected at each sampling station, depth integrated nutrient inventories, or boxplots corresponding to specific depths within the sample site.
- Line 106-109: The manuscript aims to link AOA ecotypes to the relative availability of ammonia and urea in distinct marine ecosystems. However, the environmental niche of ammonia-oxidizing organisms as defined by abundance and oxidation rates is ultimately regulated by their substrate affinity and the environmental nutrient concentration. Please consider (1) presenting the total ammonium + urea concentration alongside the fractional urea abundance using a secondary y-axis and (2) presenting the nutrient data alongside the oxidation rate and AOA relative abundance datasets presented in *Figure 2*.
- Line 106-109: Consider presenting metabolite concentrations and ratios in *Figure 1* with a log axis.
- Line 106-109: The x-axis for individual metabolites differs between sampling locations in *Extended Data Figure 1*. Please maintain a consistent x-axis to ensure the sampling sites can be easily compared.
- Line 115-118: The Angola Gyre has the lowest total ammonium + urea concentrations presented in this study in addition to having the highest fractional urea abundance. This observation is consistent with data collected throughout the Global Ocean that show an increasing urea: NH_4^+ ratio as ammonium and urea concentrations decrease (Wan

et al., 2024). Please include information about the total concentrations while discussing the fractional urea abundance.

Line 119: Please define the term relative availability as it relates to urea-N and NH_4^+ -N. It would be recommended to additionally define availability in terms of concentration considering the relationship between urea: NH_4^+ concentration and rate ratios does not hold for polar marine regions (Damashek et al., 2019; Shiozaki et al., 2021; Tolar et al., 2017). However, there is a distinct correlation in the Global Ocean (Wan et al., 2024).

Line 121-122: The caveats associated with varying experimental methods across samples and the impact on the subsequent statistical comparisons needs to be discussed. Different methods were used to measure oxidation rates with stable isotope tracers on the Gulf of Mexico, Black Sea, and Angola Gyre cruises. There were also experimental variations between sampling depths on Black Sea cruise. Specific differences include the incubation bottles (serum bottle & exetainer), oxygen addition methods (in-situ & He/ N_2 sparge, or sparge & O_2 addition), supplemented metabolites (N_2O & DIC), supplemented metabolite concentration, and supplemented isotope percentages differed between the cruises (lines 338-412). A primary concern is the influence of tracer and oxygen additions on the measured rates, which is not discussed in the manuscript. Please respond to these caveats as part of this review.

Line 122: It is difficult to compare the specific tracer and pool additions in *Supplementary Table 2*. Please include an individual row or column for each metabolite. Consider adding the total ^{15}N percentages in the table.

Line 122-129: Lower ammonia-oxidation rates are observed in environments with lower measured NH_4^+ concentrations. However, the ammonia oxidation rates and comparison between ammonium concentration and the ammonia oxidation rate are not presented. Please include and reference these datasets.

Line 130: Please define urea-derived ammonia oxidation and the urea oxidation notation in the introduction instead of the results.

Line 132: The urea oxidation rates are not explicitly compared in *Figure 2b*. Please include and reference these comparisons. Consider adding comparison between urea concentration and urea oxidation rate as well.

Line 133-141: Please present the absolute ammonia and urea oxidation rates in addition to the fractional rates presented in *Figure 2a*. The oligotrophic Angola Gyre has the lowest measured urea oxidation rates and ammonia concentrations presented in the study. Trends in the Global Ocean demonstrate that decreasing ammonia concentrations in the open ocean correlate with increased the urea: NH_4^+ concentration and oxidation ratios (Wan et al., 2024). The results presented here

further confirm these observations. However, an explicit comparison between concentration, rate, and ratios should be presented here.

Line 147-148: The statement “*as urea must ... via ammonia monooxygenase*” should be moved to the introduction.

Line 147: Urea is also broken down to carbon dioxide (CO₂), which can be assimilated. Please mention CO₂ while discussion urea hydrolysis

Line 148-159: The addition of a large background NH₄⁺ to labeling incubation have previously been used to examine whether the ¹⁵NO₂⁻ was produced via direct or indirect oxidation of the ¹⁵N labeled substrates (Kitzinger et al., 2019; Wan et al., 2024). Please update section to reflect the measurement of direct or indirect urea oxidation.

Line 152-159: The axes for the Gulf of Mexico are an order of magnitude greater than the other sampling sites in *Figure 2b* making it difficult to make a direct comparison between the sampling sites. It is apparent that the addition of ammonia significantly decreased the urea oxidation rate in the Gulf of Mexico, while the tracer has no impact in the Black Sea and Angola Gyre. However, the Black Sea and Angola Gyre urea oxidation rates are an order of magnitude lower than the presented Gulf of Mexico rates. The different responses may be attributed to the lower *Nitrosopumilus* (0.04 d⁻¹) and *Nitrosopelagicus* (0.002 d⁻¹) NanoSIMS-based growth rates in the Black Sea (line 229-236) compared to the Gulf of Mexico (0.23 d⁻¹). The faster growth rate may enable the Gulf of Mexico AOA population to transition their metabolism away from urea-oxidation over the 24-hour isotope incubations. The Black Sea and Angola Gyre AOA populations may also transition their metabolism towards ammonia-oxidation if the incubation times were corrected for the AOA growth rates. Consider discussing the impact of concentration and growth rates on direct versus indirect urea oxidation rates.

Line 160: Please change “*all known ammonia oxidizers*” to “*all known ammonia-oxidizing archaea*”. All ammonia oxidizers encode an ammonia monooxygenase (AMO) and hydroxylamine oxidoreductase (HAO) to oxidize ammonia to nitric oxide (Caranto and Lancaster, 2017). However, ammonia-oxidizing bacteria (AOB) and ammonia oxidizing archaea (AOA) have different responses to the nitric oxide (NO) scavenger PTIO, suggesting the mechanism converting nitric oxide (NO) to nitrite may differ between AOB and AOA (Martens-Habbena et al., 2015).

Line 162-165: Please specify that only the *Betaproteobacteria* ammonia-oxidizing bacteria isolates demonstrated a clear preference for urea, while the marine *Gammaproteobacteria* AOB isolate did not demonstrate a clear preference for urea or ammonium (Qin et al., 2024). The *Gammaproteobacteria* are probably most relevant for this study.

- Line 165-167: Please indicate the increasing importance of urea in oligotrophic regions has been demonstrated several times before with ^{15}N tracer studies. These results further confirm the hypothesis in different marine environments.
- Line 174: The referenced *Extended Data* figure appears to be incorrect. *Extended Data Figure 3* is a phylogenetic tree and does not include information about AOA abundance. It appears the intended reference was *Extended Data Figure 5*.
- Line 175-178: The difference in cell concentration appears to be driving the measurements presented in this manuscript. Consider presenting the ammonia-oxidizer cell abundance either in the main text, extended data, or supplemental material.
- Line 179-181: Please discuss how bulk rates are dependent on cell concentrations and cell specific rates. This is important because the two order of magnitude difference might be attributed to an order of magnitude difference in cell concentration and an order of magnitude difference in the cell specific rate.
- Line 181-188: The contrasting abundance and pangenomics of *Nitrosopumilus* and *Nitrosopelagicus* in coastal estuaries and the open ocean has previously been reported (Qin et al., 2020; Santoro et al., 2017). The results presented in this section further confirm the previous reports of AOA biogeography and genomic inventory. Please reference previous studies.
- Line 190-191: Consider mentioning the reason for specifically investigating the *dur3* urea transporter, since AOA encode three types of urea transporters (*dur3*-type, ATP dependent, and *yut*-type).
- Line 197: Additional organisms contain ureases and urea transporters. Please include other organisms in the relative abundance comparisons in *Figure 2c-d*.
- Line 197: The *ureC* RPKM values presented in *Extended Data Figure 5b* suggest the Gulf of Mexico AOA population must compete with *Cyanobacteria* and *Proteobacteria* for urea. The competition for urea may influence the indirect urea utilization presented in *Figure 2b*. The Angola Gyre and Black Sea AOA are the dominant urea hydrolyzers and presumably experience minimal urea competition. It is possible these AOA may co-consume ammonia and urea due to the slow growth rates measured with NanoSIMS. It may be worth mentioning that urea competition may influence the observed AOA ecophysiology.
- Line 197: The ratio of urease (*ureC*) and the *dur3*-type urea transporter to the ammonia monooxygenase across AOA lineages is presented in lines 193-211, yet the data is in *Extended Data Figure d-g*. Consider moving this figure into the main text.
- Line 212: The combined CARD-FISH and NanoSIMS dataset used to evaluate the nitrogen assimilation strategy across marine AOA lineages makes this manuscript unique

and furthers our understanding of AOA ecophysiology. The manuscript should be centered around the single-cell N-assimilation measurements of AOA in contrasting marine environments, instead of presenting it at the end.

Line 236: Please maintain a consistent y-axis across microorganisms to ensure easy visual comparisons. It may be worthwhile to present the data on a log axis given the order of magnitude difference between the *Nitrosopelagicus* WCA and *Nitrosopumilus* ¹⁵N biomass enrichments in *Figure 3a-b*.

Line 236: Please improve contrast of the ¹⁵N biomass enrichments in *Figure 3a-b* to make it easier to distinguish single-cells from the filter background.

Line 240-252: The absolute ¹⁵N-urea biomass enrichment values in *Figure 3* indicate *Nitrosopumilus* and other microorganisms assimilated more urea than *Nitrosopelagicus*. The higher enrichment suggests *Nitrosopumilus* and other microorganisms have higher cell-specific urea assimilation rates and will outcompete *Nitrosopelagicus* in the absence of ammonium. However, the dual substrate (urea + NH₄⁺) dataset suggests *Nitrosopumilus* and other microorganisms repress urea assimilation in the presence of ammonium while *Nitrosopelagicus* continued to assimilate urea at similar rates. The dataset suggests *Nitrosopelagicus* do not have a distinct metabolite preference and may co-consume urea and ammonium, similar to the marine *Gammaproteobacteria* AOB *Nitrosococcus oceani* (Qin et al., 2024). Although an additional dual substrate incubation with ¹⁵N-NH₄⁺ would be required to confirm this hypothesis. Please comment on this shortfall.

Line 240-252: Consider comparing urea utilization under the dual substrate incubation across lineages. It appears that *Nitrosopelagicus* may have the greatest assimilation rate under the dual substrate condition. Therefore, *Nitrosopelagicus* may be the predominant urea-oxidizer in the presence of ammonia and urea, which could then explain the direct urea-oxidation measurements in *Figure 2b*.

Line 256: Please remove the statement “*thus essentially doubling its growth rate*”. The data presented in this manuscript does not demonstrate the growth rate of *Nitrosopelagicus* was doubled via urea and ammonium co-consumption. Previous observations of co-consumption by *Nitrosococcus oceani* did not report a statical difference in the growth rate while consuming both ammonia and urea (Qin et al., 2024).

Line 258-260: This statement disregards the influence of metabolite concentrations. Please revise the statement.

- Line 263-264: The underestimation of marine nitrification by approaches only considering ammonia-oxidation has been reported before. Please differentiate this study by highlighting single-cell N-assimilation across AOA lineages.
- Line 276-277: Please remove the statement “*allowing them to double their growth rates*”. See comments about Line 256.
- Line 281-282: The demonstration of preferred urea utilization by *Nitrosopelagicus* and concluding urea utilization could differentiate the niche of marine AOA lineages is novel. This finding should be highlighted throughout the manuscript, including the introduction and abstract.
- Line 288-290: Consider softening the statement “*we hypothesize that Nitrosopelagicus obtains ammonium from other DON compounds*”. The utilization of DON compounds by ammonia-oxidizers has been limited to urea, cyanate, and guanidine in isolates and *in-situ* utilization of glutamate has been demonstrated.
- Line 325-328: Explicitly mention analytical methods used to measure metabolites in manuscript
- Line 330-331: A brief overview of the experimental methods from the Gulf of Mexico cruise should be presented here.
- Line 337: Different process rate methods were used for the Gulf of Mexico, Angola Gyre, and Black Sea. Please elaborate on differences between methods.
- Line 338-343: Sampling depths, storage duration prior to experiment, and light exposure are not explicitly mentioned. Please consider including.
- Line 353-355: State replication used in tracer experiment.
- Line 360-361: It is unclear how the water column oxygen concentrations correspond to the ambient oxygen concentrations in the Schott bottle. Please elaborate on choice.
- Line 363: Duplicate samples were collected for each condition. Please elaborate on using duplicate as opposed to triplicate samples for downstream statistical analysis.
- Line 370: Please mention the isotope supplementation concentration was chosen taking *in-situ* concentrations into consideration and wanting the overall ^{15}N at% to be greater than 90% in the main text.
- Line 373-388: Please include all rate calculation equations in main text or supplement to this manuscript.
- Line 392-397: Please elaborate on the use of different methods to quantify initial ^{15}N -urea on the Black Sea and Angola cruise.
- Line 397: Consider a paragraph break to separate the statistical analysis from the methods.

- Line 409-412: Setting the non-significant slopes to the limit of detection could lead to an overestimate of the measured rate, especially if the true rate is zero. Consider elaborating on this choice.
- Line 443: The contig assembly and read mapping methods are not explicitly stated here. Please update this section to include package and settings as done in ref. 54. It is implied the metagenomic and metatranscriptomic abundance was evaluated with a read-based method instead of a MAG-based method. However, the method for calculation abundance should be explicitly mentioned.
- Line 444-446: It is not clear how marker gene abundance was assessed until line 477. The database curation method presented in lines 447-476 is important, but could be moved to supplement and referenced in the main text.
- Line 480: Please define BSR as bit-score ratio when it is first introduced on line 480
- Line 595: Please include equations used to calculate cellular ^{15}N -atom% excess and growth rate in the main text or supplement of this manuscript.

Reviewer #3 (Remarks to the Author):

Overall judgment: The manuscript Urea use drives niche separation between dominant marine ammonia oxidizing archaea propose ecological niches for *Nitrosopelagicus* and *Nitrosopumilus* — two genera of marine ammonia-oxidizing archaea (AOA) — based on single-cell assimilation of ammonium and urea in the Black Sea. Previous work from the authors has demonstrated distinct ammonium (NH_4^+) and organic nitrogen utilization strategies by AOA and nitrite-oxidizing bacteria (NOB) in the Gulf of Mexico (Kitzinger et al., 2020, 2019). The dominance of *Nitrosopelagicus* ammonia oxidizing archaea (AOA) in the open ocean and *Nitrosopumilus* AOA in estuarine areas has been well documented (Qin et al., 2020; Santoro et al., 2017). However, physiological traits defining the ecological niche of these ubiquitous marine AOA has remained elusive. Demonstrating distinct nitrogen metabolite preferences dictates the environmental niche of these marine AOA genera — as was recently demonstrated for the major ammonia-oxidizing microorganism lineages (Qin et al., 2024) — would represent a significant contribution to the field. All experimental and computational approaches were sufficient. However, specific techniques and software packages should be explicitly stated in the methods section and there were major concerns listed below. Care should be taken to update the introduction and discussion to improve clarity. The manuscript provides compelling evidence of genus-level niche differentiation between marine AOA and could shape our understanding of the marine nitrogen cycle. Despite presenting compelling evidence for the distinct metabolite preferences of *Nitrosopelagicus* and *Nitrosopumilus* in the Black Sea, this manuscript would have benefited by constraining the scope to the Black Sea and linking the single-cell observations with the environmental rate and metaomic datasets. The most significant findings of this worked can be captured as following:

Significance #1: Combining single-cell N-assimilation measurements with in-situ genus level phylogeny in samples collected from the Black Sea enabled the authors to present compelling evidence of the distinct strategies *Nitrosopelagicus* and *Nitrosopumilus* to employ their ammonia and urea utilization (Figure 3). Their results suggest *Nitrosopumilus* prefer to utilize ammonia and repress urea utilization in the presence of ammonia, while *Nitrosopelagicus* did not demonstrate a distinct preference toward ammonia or urea. Despite preferring ammonia, the ^{15}N -urea biomass enrichment — and the calculated growth rate — of *Nitrosopumilus* and all other microorganisms appear to be faster than *Nitrosopelagicus* when ^{15}N -urea was supplemented. Yet, *Nitrosopelagicus* may outcompete *Nitrosopumilus* for urea in the presence of excess ammonia based on higher ^{15}N urea biomass enrichments in samples supplemented with ^{15}N -urea and ^{14}N - NH_4^+ . Considering all marine environments are characterized by both ammonium and urea, it would suggest the niche of *Nitrosopelagicus* is defined by urea-utilization in ammonium replete marine environments.

We thank the reviewer for their kind assessment.

We thank the authors for updating the manuscript and your thorough response and recognize a significant amount of work went into reframing the abstract and introduction to highlight the niche differentiation between *Nitrosopelagicus* and *Nitrosopumilus*. The manuscript now builds from bulk to single-cell measurements to provide insight into the abundance of *Nitrosopelagicus* in ammonia-replete waters with high urea to ammonia ratios. This work provides fundamental

insights into the ecophysiology of *Nitrosopelagicus* AOA and will be a major contribution to the field.

The authors have done an excellent job addressing our comments. There are a few minor recommendations below with the primary recommendation in the major comment 3 response. It has been a pleasure to review this manuscript.

Shortfall #1: Unfortunately, the compelling single-cell analysis from the Black Sea could not be extended to the oligotrophic Angola Gyre where *Nitrosopelagicus* comprised a larger proportion of the nitrifier community.

While we agree that it would have been a nice addition to extend the single-cell analysis to the Angola Gyre, we find that the Black Sea single-cell analysis provides the most insightful information - as this is the only environment where we can test the response of environmental communities of both *Nitrosopelagicus* and *Nitrosopumilus* in the same samples, without confounding changes in other environmental conditions. We now also highlight this better in the introduction (L99-103) and the main text (L248-251).

We thank the authors for explaining that the Black Sea is the only study site where the environmental response of both *Nitrosopelagicus* and *Nitrosopumilus* can be tested with single-cell analysis in the manuscript. Focusing on the unique conditions of the Black Sea to comparing *Nitrosopelagicus* and *Nitrosopumilus* in L99-103 and L248-251 have significantly improved the manuscript. It is a nice complement to the initial NanoSIMS rationale in L248-250.

It is a shame that the authors were unable to obtain sufficient CARD-FISH signals to visualize AOA in the Angola Gyre. These samples could have provided additional evidence for the urea-based niche of *Nitrosopelagicus*. However, the authors provide a compelling rationale for why these samples were not included (L245-248 and Supplementary Discussion). The absence of single-cell data from the Angola Gyre does not detract from the Black Sea NanoSIMS results, especially after highlighting the Black Sea site (L99-103 & L248-251).

Significance #2: The authors also characterize the relative contribution of ammonium and urea to ammonia oxidation in three distinct marine environments with unique productivity regimes (Figure 1 & 2).

Shortfall #2: Though presented as a novel finding in the manuscript, the increased contribution of urea relative to NH_4^+ to nitrification in the open ocean has been reported several times before and does not challenge the current understanding of marine nitrification (Arandia-Gorostidi et al., 2024; Santoro et al., 2017; Tolar et al., 2017; Wan et al., 2024). The distribution of urea-dependent ammonia-oxidation (hereafter urea-oxidation) and ammonia-oxidation is well known at the global scale. The urea-dependent ammonia-oxidation rate and ammonia-oxidation rate decrease from copiotrophic eutrophic coastal zones to the oligotrophic open ocean while the contribution of urea to total ammonia-oxidation increases due to decreasing substrate concentrations (Wan et al., 2024). The authors confirm the validity of this framework in non-polar marine regions. However, the isotopic rates measurements were not paired with single-

cell analysis, metagenomics, and metatranscriptomics to study the niche differentiation between of Nitrosopelagicus and Nitrosopumilus.

We agree that there are now a number of observations that urea-derived oxidation rates seem to contribute more to nitrification in oligotrophic environments. A number of these studies have been recently published, and we have now rewritten the introduction of the manuscript to reflect this state of the art, and have made this more evident throughout the manuscript. Yet, we think it is important to demonstrate that this observation holds true also for the environments investigated here, as this is key background information for our single cell analyses, which enable us to link specific substrate utilization patterns to specific AOA groups, which is the main finding of our study. As the reviewer points out, this has not been undertaken by any previous studies.

We thank the authors for including previous observations of urea-dependent ammonia-oxidation in the rewritten introduction and throughout the results/discussion. The addition of L69-71, L86-88, L122-123, and L169-171 to the manuscript provides a strong background into what is known about urea-dependent ammonia-oxidation, while L62-64, 74-78, 86-88, 89-103 present a clear rationale for studying the copiotrophic, mesotrophic, and oligotrophic environments. It is important for the authors to confirm observations from their sampling sites reflect previous studies and this point has been effectively communicated (L122-123 & L175-176). This background is required to highlight the scope of this study (L178-180). The authors have done an excellent job at putting their work into the context of previous studies.

Major Points:

1. The distribution of urea-oxidation and ammonia-oxidation in marine environments is well known at the global scale. Recent studies have also demonstrated the contribution of urea and ammonia to total ammonia oxidation by mesopelagic and bathypelagic AOA is nearly equivalent (Arandia-Gorostidi et al., 2024; Laperriere et al., 2021; Santoro et al., 2017; Tolar et al., 2017; Wan et al., 2024). Therefore, the contribution of urea to nitrification has been demonstrated and the rate measurements presented here do not represent a unique or novel finding. It is important that the authors accurately represent our current understanding.

We agree with the reviewer and have ensured that this is appropriately acknowledged throughout the manuscript.

We thank the authors for acknowledging these studies throughout the manuscript.

2. In our opinion the most significant claim in the manuscript is urea utilization in the presence of ammonium defines the niche of Nitrosopelagicus. However, this finding is not emphasized throughout the manuscript and must be strengthened as a result. The authors should consider framing the manuscript around this finding as it might explain the distribution of Nitrosopelagicus in the open ocean. Compelling evidence that Nitrosopelagicus AOA are responsible for urea hydrolysis in the Black Sea or Angola Gyre is not presented and would serve as an indicator of their distinct preference for urea-derived ammonium over ammonium if presented. Neither the environmental rate data in Figure 2a-b nor the meta-omic data in Figure 2c-d and Extended Data Fig. 5 contribute to defining the niche of Nitrosopelagicus. Focusing on the depth resolved samples collected from the Black Sea instead of clustering all samples together may provide additional insight and it is suggested that the authors provide these additional data sets. Please

comment on how the oxidation rates and meta-omic sampling contribute to understanding the environmental niche of *Nitrosopelagicus* and *Nitrosopumilus* in the Black Sea.

We agree with the reviewer that the continued use of urea in the presence of ammonium by *Nitrosopelagicus* is a major finding of our manuscript and have put more emphasis on this result in the text. However, we strongly feel that the detailed description of the study sites and bulk rate measurements and -omics analyses are required to set the scene and provide the background for our single cell measurements, which ultimately link *Nitrosopelagicus* to urea utilization. We further are convinced that the presented single cell data (unlike the bulk data, which is ambiguous) unequivocally shows that *Nitrosopelagicus* utilizes urea.

In response to the reviewers' comment we have now included additional panels (Supplementary Figure 2) showing exemplary depth distributions for the Black Sea of ammonium concentrations, oxidation rates and *Nitrosopumilus/Nitrosopelagicus* abundance, which highlights that *Nitrosopelagicus* is more abundant in ammonium-poor waters.

We thank the authors for their response and updating to the manuscript to highlight *Nitrosopelagicus* continues to use of urea in the presence of urea throughout the manuscript. We agree with the assessment provided by the authors. The persistence of urea-dependent ammonia oxidation in the presence of excess ammonia demonstrates a preference for urea in the Black Sea and Angola Gyre microbial communities (Fig 2b). The Black Sea single-cell data presented in Figure 3c unequivocally demonstrates that *Nitrosopelagicus* assimilates urea in the presence/absence of ammonium. The single-cell assimilation data from the labeled urea and unlabeled ammonium presented in Figure A (below) further confirms the ureolytic niche of *Nitrosopelagicus* by demonstrating ^{15}N urea was more enriched in *Nitrosopelagicus* than *Nitrosopumilus*.

We also agree with the assessment provided by the authors about the structure of the manuscript. The site description, bulk rate, and -omic analysis are required background for the single-cell analysis. The addition of Supplementary Figure 2k-m shows how *Nitrosopelagicus* becomes more abundant with ammonia-depletion in the Black Sea, which has been emphasized in L200-205. We had hoped *Nitrosopelagicus* *amoA* or *ureC* abundance would correlate with the urea-dependent ammonia oxidation rates. However, the higher *ureC:amoA* and *dur3:amoA* ratios of *Nitrosopelagicus* presented in Supplementary Figure 5 suggest an ureolytic niche, which has been sufficiently described in L235-238. I find the *ureC:amoA* and *dur3:amoA* ratios in Supplementary Figure 5 more compelling than Figure 2c, especially after the addition of L235-238. However, the authors have addressed why Supplementary Figure 5 subfigures have not been moved to Figure 2c. No further action is required.

3. The caveats associated with varying experimental methods across samples and the impact on the subsequent statistical comparisons needs to be discussed. Different methods were used to measure oxidation rates with stable isotope tracers on the Gulf of Mexico, Black Sea, and Angola Gyre cruises. There were also experimental variations between sampling depths on Black Sea cruise. Specific differences include the incubation bottles (serum bottle & exetainer), oxygen addition methods (in-situ & He/N₂ sparge, or sparge & O₂ addition), supplemented metabolites (N₂O & DIC), supplemented metabolite concentration, and supplemented isotope

percentages differed between the cruises (lines 338-412). A primary concern is the influence of tracer and oxygen additions on the measured rates, which is not discussed in the manuscript. A discussion about methodological differences and subsequent statistical analysis is requested.

The experimental setup was adjusted during each cruise to match *in situ* conditions as accurately as possible while also accommodating best feasibility. The ^{15}N tracer additions differed due to the different nutrient availability across sites. Similarly, the oxygen concentrations in the incubations were adjusted to match *in situ* oxygen conditions. By mimicking the respective *in situ* conditions we anticipate that our potential oxidation rates reflect the actual rates as closely as possible. The incubation period is short (<24h) and the rates are linear from the beginning of the incubation on, without a lag phase, which indicates that ammonia oxidizers are active *in situ*. While experimental details differed between cruises, the association of *Nitrosopelagicus* with enhanced urea use holds across our study sites, and also other studies in open ocean settings where *Nitrosopelagicus* typically dominates (Santoro et al. 2017 L&O, Wan et al. 2024, GBC). Thus, we are confident that these minor differences in protocols did not impact the outcome of our experiments, or our conclusions. To address this issue, we added a section to the Supplementary Discussion.

We thank the authors for this clarification and including additional details in the Supplementary Discussion. Our primary concern was comparing oxidation rates from an aerobic Schott bottle with a hypoxic exetainer/serum bottle. It was unclear to us if the 500 mL Schott bottles used for the urea incubations with added ammonium pool in the Black Sea were sealed with butyl rubber stoppers in L377-378, especially considering the Schott bottles used in the Angola Gyre were aerobic. However, ref 43 and L380-382 indicate the Black Sea Schott bottles were sealed, sparged, and amended with O_2 . Consider removing the description of the urea incubations with added ammonium pool in the Black Sea from the parentheses and mentioning that both bottles were sealed with butyl rubber stoppers.

The response and description the authors added in L401-403 explaining the rationale for using a Schott bottle in the Angola Gyre address our concerns about the methodological differences. We agree that the authors were able to obtain oxidation rates reflective of actual oxidation rates by mimicking *in situ* concentrations and the minor differences did not impact experimental outcomes. Upon reading the response of the authors, we see rational for the tracer addition at each site and station was mentioned in L411-414. We had incorrectly assumed the description of “all three environments” in L411-414 was describing the 3 sample sites in Kitzinger et al. 2019 based on the structure of the *Process rate experiment* methods section. Our apologies if this caused confusion. Consider moving L410-414 and portions of the reviewer response to the beginning of the *Process rate experiment* methods section before describing site specific methods. This will help to avoid confusion with the methods from Kitzinger et al. 2019, especially for readers unfamiliar with the techniques the authors employed.

4. Ammonium concentrations classically define the environmental niche of ammonia-oxidizing microorganisms. A recent study found the ammonia affinity was higher than the urea affinity in the open ocean providing an explanation for the distribution of urea in this environment (Wan et al., 2024). However, a discussion about the impact of concentrations on niche differentiation is

lacking in the current manuscript, especially while discussing direct versus indirect urea oxidation rates (Figure 2b). For example, the axes for the Gulf of Mexico are an order of magnitude greater than the other sampling sites in Figure 2b making it difficult to make a direct comparison between the sampling sites. It is apparent that the addition of ammonia significantly decreased the urea oxidation rate in the Gulf of Mexico, while the tracer has no impact in the Black Sea and Angola Gyre. However, the Black Sea and Angola Gyre urea oxidation rates are an order of magnitude lower than the presented Gulf of Mexico rates. The different responses may be attributed to the lower *Nitrosopumilus* (0.04 d^{-1}) and *Nitrosopelagicus* (0.002 d^{-1}) NanoSIMS-based growth rates in the Black Sea (line 229-236) compared to the Gulf of Mexico (0.23 d^{-1}). The faster growth rate may enable the Gulf of Mexico AOA population to transition their metabolism away from urea-oxidation over the 24 hour isotope incubations. The Black Sea and Angola Gyre AOA populations may also transition their metabolism towards ammonia-oxidation if the incubation times were corrected for the AOA growth rates. Therefore, we suggest that you consider discussing the impact of concentration and growth rates on direct versus indirect urea oxidation rates.

Generally, we agree that it is possible that the AOA population transitions their metabolism depending on the substrate available. This is the reason why we chose to incubate <24h. Additionally, if as hypothesized the Gulf of Mexico AOA would transition away from urea oxidation, we would see a flattening of the nitrite production from urea over time, however, we see a strong linear trend over time. The same holds true for the hypothesis that the Angola Gyre and Black Sea AOA population transition towards ammonia oxidation. We see a strong linear trend without a lag phase in all our rates. Additionally, in the single cell measurements, we see that even though *Nitrosopumilus* reduces its urea utilization in the presence of ammonium, *Nitrosopelagicus* did not show decreased urea assimilation rates in presence of ammonium. Together with the observed strong correlation of *Nitrosopelagicus* dominating in high urea relative to ammonium availability (concentration based) (and vice versa for *Nitrosopumilus*), we are confident that our observations are a result of inherent physiological traits defining these AOA.

We thank the authors for explaining why you conclude the observations are a result of inherent physiological traits of *Nitrosopelagicus* and *Nitrosopumilus*. We thought the adaption of the AOA populations to the *in situ* concentration and the AOA growth rates might contribute to the differences in indirect and direct urea oxidation observations. AOA in Gulf of Mexico are continuously exposed to ammonium, while AOA in the Black Sea and Angola Gyre are adapted to ammonia concentrations near the limit of detection. The abundance of ammonium in Gulf of Mexico suggests there is no need for AOA to directly utilize urea in the presence of ammonium. Conversely, AOA in the Black Sea and Angola Gyre must utilize urea for continued growth (Supplementary Figure 2). It is clear from the measured *in situ* rates (Fig 2b), single-cell analysis (Fig 3c), and depth distribution of abundance in the Black Sea (Supplemental Figure 2h-m) that the *Nitrosopelagicus* becomes more abundant in water column as ammonium decreases and the availability of urea relative to ammonium increases.

It is possible the slow growth rate of *Nitrosopelagicus* (doubling time ~1 year) may prevent the authors from observing a transitioning from direct urea utilization to indirect utilization. However, the authors point out that the decreased urea utilization by *Nitrosopumilus* and constant utilization by *Nitrosopelagicus* provide an explanation against this argument. This concern has been addressed and no further action is necessary.

5. There are distinct metabolite distributions with depth presented in Extended Data Figure 1, which are not captured in Figure 1b-d or Figure 2a-b. The ammonium concentrations detected in the Black Sea samples are near the limit of detection until ~120m where the ammonium concentrations increase up to 7.4 $\mu\text{M-NH}_4^+$. The distribution of AOA in the Black Sea samples also appears to have an inverse relationship with ammonia availability based on the Thermoproteota ureC RPKM dataset presented in Extended Data Figure 5b. Please, consider presenting the nutrient distributions collected at each sampling station, depth integrated nutrient inventories, or boxplots corresponding to specific depths within the sample site to characterize the environmental gradient within the Black Sea.

We thank the reviewer for the suggestion. We now present ammonium concentration profiles along with ammonia and urea oxidation rates and relative AOA abundance from the Black Sea in Supplementary Figure 2. We also refer to this depth distribution in the main text e.g. L115-116.

We thank the authors for adding depth data from the Black Sea to Supplementary Figure 2. we noticed the figure label only mentions the Angola Gyre. Consider mentioning the Black Sea in the figure label or consider making Supplemental Figure 2h-m into a new Supplemental Figure.

6. The difference in AOA cell concentrations might contribute to the rate measurements presented in this manuscript. Please discuss how bulk rates are dependent on cell concentrations and cell specific rates. This is important because the two order of magnitude difference might be attributed to an order of magnitude difference in cell concentration and an order of magnitude difference in the cell specific rate. Additionally, consider presenting the ammonia-oxidizer cell abundance either in the main text, extended data, or supplemental material.

We agree with the reviewer that both the overall abundance of AOA and their specific activity influence the measured oxidation rates. It appears that in the highly productive Gulf of Mexico, cell specific rates are much higher than in the more oligotrophic Black Sea and Angola Gyre. We have already alluded to this mismatch between AOA abundance and oxidation rates in the previous version of the manuscript, and now specifically mention the difference in specific cell rates. AOA cell numbers are provided both in the main text, and in Supplementary Data 1.

We thank the authors for including the AOA cell numbers in Supplementary Data 1 and adding L194 to address this comment.

7. The axes limits are inconsistent between conditions in many figures making direct comparison difficult. This is especially apparent for the ^{15}N biomass enrichment data of Nitrosopelagicus presented in Figure 3c, which is an order of magnitude lower than other genera. Therefore, we recommend adjusting the axis accordingly.

We took your suggestions into consideration. However, we have tried multiple options of data depiction (e.g. same axes limits, log scale) for this particular plot and found that the current formatting showed the data most clearly. As the axes limits are exactly one magnitude different, the comparison is still possible and we also specifically point out the difference in scales in the figure legend.

We thank the authors for taking the suggestion into consideration and adding the note to the figure legend. The figure communicates the message of the manuscript.

8. Please remove the statement “thus essentially doubling its growth rate” in line 256. The data presented in this manuscript does not demonstrate the growth rate of *Nitrosopelagicus* was doubled via urea and ammonium co-consumption. Previous observations of co-consumption by *Nitrosococcus oceanii* did not report a statistical difference in the growth rate while consuming both ammonia and urea. The statement is not sufficiently supported by the experimental results presented in Figure 3 and should be removed from the manuscript.

We agree that this sentence was not careful enough given the presented evidence and have revised it. It now says “Furthermore, although we cannot unequivocally show that *Nitrosopelagicus* cells were assimilating ammonium and urea simultaneously, if they do, then this strategy could potentially lead to a doubling of their growth rate when both ammonium and urea are available.” (L288-291).

We thank the authors for updating L288-291.

Minor Points (line-by-line):

Line 29-31: The increased contribution of urea relative to NH_4^+ to nitrification in the open ocean has been reported several times before (Arandia-Gorostidi et al., 2024; Tolar et al., 2017; Wan et al., 2024) and does not challenge the current understanding of marine nitrification.

Please tone down novelty claims of your findings.

We have rephrased the abstract taking into consideration the reviewer comments.

We thank the authors for rephrasing the abstract

Line 31-33: The dominance of *Nitrosopelagicus* ammonia oxidizing archaea (AOA) in the open ocean and *Nitrosopumilus* AOA in estuarine areas was previously reported by (Santoro et al., 2015) and Qin et al., 2020. However, physiological traits defining the ecological niche of these ubiquitous marine ammonia-oxidizing archaea have not yet been reported and would represent a major contribution to the field. Please revise manuscript to integrate this rationale.

We have rephrased the abstract taking into consideration the reviewer comments.

We thank the authors for rephrasing the abstract

Line 33-36: The trend of increased urea-N oxidation relative to ammonia oxidation from eutrophic coastal zones to the oligotrophic open ocean was previously demonstrated by (Wan et al., 2024) in the Subtropical North Pacific. Their results were further validated by comparing the urea to NH_4^+ concentration and oxidation rate ratios across the coastal, epipelagic zone, and mesopelagic zone across the global ocean. Please tone down novelty claims of your findings.

We have rephrased the abstract taking into consideration the reviewer comments.

We thank the authors for rephrasing the abstract

Line 40-41: AOA should be presented earlier in the introduction considering the hypothesis presented in the manuscript is the relative availability of ammonium versus urea drives niche differentiation between AOA the Nitrosopelagicus and Nitrosopumilus genus.

We have restructured the introduction based on the reviewer comments, and currently introduce AOA already in the second paragraph of the introduction, after a short first paragraph describing why it is important to understand nitrification in the open ocean. In the revised introduction we have added substantial new text into the introduction regarding what is known about niche differentiation.

We thank the authors for updating the introduction to focus on nitrification, AOA, and niche differentiation!

Line 55-56: While there are still unresolved aspects about the factors shaping the distribution of AOA at the regional scale, the factors shaping the global distribution of urea and ammonia oxidation in the global ocean are known at the global scale (Wan et al., 2024). Although the urea-dependent ammonia-oxidation rate and ammonia-dependent ammonia-oxidation rate decrease from eutrophic coastal zones to the oligotrophic open ocean, the contribution of urea to total ammonia-oxidation increases due to substrate concentrations. Please update the statement in line 55-56 to reflect this understanding. It might be worthwhile to mention areas where this paradigm does not hold.

We thank the reviewer for this comment. In the revised introduction, we now specifically refer to patterns of urea utilization and higher urea availability relative to ammonium in the oligotrophic oceans.

We thank the authors for including information about the patterns of urea availability and utilization in the introduction.

Line 57: The mechanism of urea derived utilization should be mentioned while introducing urea as an AOA substrate. Urea must be transported into the cell where it is hydrolyzed by urease to ammonia and carbon dioxide. The intracellularly derived ammonia can then be utilized for ammonia-oxidation by the ammonia monooxygenase. Consider explicitly explaining this mechanism in the introduction.

We have added this information to the introduction.

We thank the authors for explaining the mechanism of urea-dependent ammonia-oxidation in the introduction.

Line 58-59: Consider using a different reference to introduce urea utilization by marine AOA, since the contribution of urea to total ammonia-oxidation was minor as stated in lines 62-64. The contribution of urea was hypothesized based on genomic surveys (Alonso-Sáez et al., 2012) and the capacity of a marine AOA to utilize urea was demonstrated with the marine strain PS0 isolated from Puget Sound (Qin et al., 2014). The contribution of ammonium and urea to total ammonia oxidation were later compared across pelagic environments where it was found to have a major contribution in Antarctic coastal waters and a minor contribution in temperate coastal waters (Tolar et al., 2017).

We have restructured and revised the introduction to include more studies investigating urea utilization. However in this particular section we are referring to Kitzinger et al. 2019 Nat.

Microbiol., as we are talking about environmental growth rates of AOA on small organic N-compounds, which have only been reported in this paper.

We thank the authors for the clarification and your rational makes sense.

Line 62-66: Recent studies have demonstrated the contribution of urea and ammonia to total ammonia oxidation by mesopelagic and bathypelagic AOA is nearly equivalent (Arandia-Gorostidi et al., 2024; Laperriere et al., 2021; Santoro et al., 2017; Tolar et al., 2017; Wan et al., 2024). Therefore, the importance of urea no longer remains a hypothesis. Please update the statements in line 62 and 65-66 to reflect the current understanding.

We revised the introduction accordingly.

We thank the authors for updating the introduction.

Line 66: The different AOA genera (i.e. *Nitrosopelagicus* and *Nitrosopumilus*) have not yet been introduced. Please refrain from mentioning AOA genera until they have been introduced. Consider moving statement anywhere after line 73.

We have revised and restructured the introduction and now early on introduce the two main genera of AOA (L72-74).

We thank the authors for revising and restructuring the introduction.

Line 72: Please mention carbon fixation pathway as it is important when comparing carbon assimilation across nitrifying lineages.

This is now included in the introduction.

We thank the authors for including a statement about carbon fixation pathways.

Line 74-75: The word “many” is not quantitative and does not denote magnitude. Either remove statement or include percentage of *Nitrosopelagicus* and *Nitrosopumilus* containing urease and urea transport proteins.

We now specify precisely the isolates that encode and utilize urease.

We thank the authors for specifying AOA isolates that encode urease and have confirmed urea utilization.

Line 80-83: This is the crux of the manuscript. The physiological mechanisms underpinning the distribution of *Nitrosopelagicus* and *Nitrosopumilus* remain unknown considering their genomic similarity. This should be highlighted earlier in the introduction, especially the lack of in-situ comparisons.

We agree with the reviewer that this is the main point of the manuscript. We now introduce that the factors driving the different distributions of *Nitrosopumilus* and *Nitrosopelagicus* are unknown early on in the revised introduction (L76-80), before describing the similarities on genomic level.

We thank the authors for including this update to the introduction.

Line 86-94: The use of the phrase “environmental gradient ...” can be easily misconstrued here. Samples were collected from three distinct marine environments, instead of sampling a transect defined by a nutrient gradient. The sampling sites provide a snapshot into the environmental

factors driving niche differentiation there, but nothing about transitions between different environmental conditions. Please specify why each site was chosen and the comparisons that were made between sites.

We rephrased this sentence for clarity. The reasoning for choosing these three sites is given in the introduction.

We thank the authors rephrasing for clarity and updating the introduction

Line 86-94: Contribution of urea to nitrification along a gradient from eutrophic coastal waters to oligotrophic open oceans was recently demonstrated by (Wan et al., 2024). However, the isotopic rates measurements by Wan were not paired with single-cell imaging, metagenomics, and metatranscriptomics. This study provides new insights into the niche differentiation between of *Nitrosopelagicus* and *Nitrosopumilus*. Please focus the results on the niche differentiation.

We agree that the niche differentiation is the key point of our manuscript and we now put more focus on the single cell results throughout. Nevertheless, the additional data (oxidation rates, metagenomics, metatranscriptomics) are essential to support these results and set the scene.

We thank the authors for updating the manuscript to highlight AOA niche differentiation. The updated introduction provides context for the study and the site description, bulk oxidation rates, and -omics datasets provide background for the Black Sea single-cell analysis, which presents the niche differentiation between *Nitrosopumilus* and *Nitrosopelagicus*.

Line 94: Urea utilization by AOA in the Gulf of Mexico has already been published by the authors (Kitzinger et al., 2020, 2019). Results from the Black Sea and Angola Gyre cruise should be compared to the Gulf of Mexico cruise. Consider focusing novelty claims on the results of the Black Sea and Angola Gyre cruise and referencing the published Gulf of Mexico dataset.

We agree that the Gulf of Mexico dataset has a supporting role in the manuscript, however, we have re-analyzed all -omics data for consistency between environments and therefore find it essential to discuss them explicitly.

We thank the authors for this response. It was not clear to us previously that the Gulf of Mexico -omics data had been reanalyzed. The inclusion in the manuscript makes sense and no further action is required.

Line 106-109: There are distinct metabolite distributions with depth presented in Extended Data Figure 1, which are not captured in Figure 1b-d. The ammonium concentrations detected in the Black Sea samples are near the limit of detection until ~120m where the ammonium concentrations increase up to 7.4 $\mu\text{M-NH}_4^+$. The distribution of AOA in the Black Sea samples also appears to have an inverse relationship with ammonia availability based on the Thermoproteota ureC RPKM dataset presented in Extended Data Figure 5b. Consider presenting the nutrient distributions collected at each sampling station, depth integrated nutrient inventories, or boxplots corresponding to specific depths within the sample site.

We adjusted Supplementary Figure 2 and this information to the main text, as described under major point 5.

We thank the authors for adjusting Supplementary Figure 2. As described in response to major point 5, please consider updating the figure title to include “Black Sea” or make Supplementary Figure 2h-m into a separate supplemental figure.

Line 106-109: The manuscript aims to link AOA ecotypes to the relative availability of ammonia and urea in distinct marine ecosystems. However, the environmental niche of ammonia-oxidizing organisms as defined by abundance and oxidation rates is ultimately regulated by their substrate affinity and the environmental nutrient concentration. Please consider (1) presenting the total ammonium + urea concentration alongside the fractional urea abundance using a secondary y-axis and (2) presenting the nutrient data alongside the oxidation rate and AOA relative abundance datasets presented in Figure 2.

We have carefully considered adding the suggested additional panels to the main figures. However, as we already provide all relevant data in the Supplementary Table(s) & Data, we instead chose to show two exemplary depth distributions of ammonium concentrations, oxidation rates and abundance of the different AOA groups for the Black Sea as extra panel of Supplementary Figure 2, as suggested by the reviewer in the comment above.

We thank the authors for taking this suggestion into consideration and including an additional panel in Supplementary Figure 2.

Line 106-109: Consider presenting metabolite concentrations and ratios in Figure 1 with a log axis.

While we have considered this, the resulting plot is very hard to read and thus we have not changed the current presentation.

We thank the authors for taking this suggestion into consideration.

Line 106-109: The x-axis for individual metabolites differs between sampling locations in Extended Data Figure 1. Please maintain a consistent x-axis to ensure the sampling sites can be easily compared.

We considered adjusting the Supplementary Figure 1. However, if we maintain consistent x-axes, the environment-specific depth distributions are not visible (e.g. ammonium concentrations in the Angola Gyre appear to be zero). Instead we now point out the different axis scales in the figure legend.

We thank the authors for taking this suggestion into consideration.

Line 115-118: The Angola Gyre has the lowest total ammonium + urea concentrations presented in this study in addition to having the highest fractional urea abundance. This observation is consistent with data collected throughout the Global Ocean that show an increasing urea:NH₄ + ratio as ammonium and urea concentrations decrease (Wan et al., 2024). Please include information about the total concentrations while discussing the fractional urea abundance.

We have adjusted this section as requested.

We thank the authors for adjusting this section.

Line 119: Please define the term relative availability as it relates to urea-N and NH_4^+ -N. It would be recommended to additionally define availability in terms of concentration considering the relationship between urea: NH_4^+ concentration and rate ratios does not hold for polar marine regions (Damashek et al., 2019; Shiozaki et al., 2021; Tolar et al., 2017). However, there is a distinct correlation in the Global Ocean (Wan et al., 2024).

We now introduce relative urea-N availability in L120, and reference Wan et al. 2024 GBC specifically. However, we feel that a discussion of why this relationship may be less pronounced in polar regions is beyond the scope of the present manuscript.

We thank the authors for referencing the Global Ocean correlation in Wan et al. 2024 GBC. The authors are correct in pointing out that the less pronounced relationship in polar regions is beyond the scope of this manuscript.

Line 121-122: The caveats associated with varying experimental methods across samples and the impact on the subsequent statistical comparisons needs to be discussed. Different methods were used to measure oxidation rates with stable isotope tracers on the Gulf of Mexico, Black Sea, and Angola Gyre cruises. There were also experimental variations between sampling depths on Black Sea cruise. Specific differences include the incubation bottles (serum bottle & exetainer), oxygen addition methods (in-situ & He/N_2 sparge, or sparge & O_2 addition), supplemented metabolites (N_2O & DIC), supplemented metabolite concentration, and supplemented isotope percentages differed between the cruises (lines 338-412). A primary concern is the influence of tracer and oxygen additions on the measured rates, which is not discussed in the manuscript. Please respond to these caveats as part of this review.

While experimental details differed between cruises, urea use has previously been reported to increase towards oligotrophic regions using different experimental setups, and conducted by different authors (e.g. Santoro et al. 2017 L&O; Wan et al. 2024 GBC). Thus, we are confident that minor differences in protocols did not impact the outcome of our experiments, or our conclusions.

Specifically, we would like to clarify that within each cruise, we only compare treatments (i.e. ammonia and urea oxidation) that received the same amendments in regard to oxygen and DIC. Oxygen availability plays a key role for AOA activity (as it is required for activation of ammonia into hydroxylamine and as terminal electron acceptor), and oxygen affinities of AOA are exceptionally high (nM range, Bristow et al. 2016 PNAS). In our experiments, oxygen was adjusted to reflect *in situ* conditions, however, we anticipate that even samples from anoxic depths contained some residual oxygen as even careful handling introduces oxygen contamination (DeBrabandere et al. 2012 J. Microbiol. Methods). The linearity of our rates, also in low oxygen samples, indicates that oxygen was present at higher concentrations than required for AOA activity.

We do not anticipate that differences regarding the addition of DIC or N_2O pools between cruises directly affect the measured nitrification rates. DIC additions were only 10% of the ambient DIC pool (which is unlikely to stimulate DIC fixation activity), and N_2O is not the result of an enzymatic reaction in AOA, therefore it should have a negligible effect on the measured rates.

Finally, the ammonia oxidation rates measured differ by orders of magnitude between the three environments. While differences in experimental setup could have small effects on the

measured oxidation rates, we do not expect them to cause such an order of magnitude difference.

We added an additional section discussing differences between cruises to the Supplementary Discussion.

We thank the authors for adding the Supplementary Discussion. Our detailed response can be found under major point 3.

Line 122: It is difficult to compare the specific tracer and pool additions in Supplementary Table 2. Please include an individual row or column for each metabolite. Consider adding the total ^{15}N percentages in the table.

Supplementary Table 2 was adjusted as suggested.

We thank the authors for updating Supplementary Table 2. It is significantly easier to compare the tracer concentrations between environment and process.

Line 122-129: Lower ammonia-oxidation rates are observed in environments with lower measured NH_4^+ concentrations. However, the ammonia oxidation rates and comparison between ammonium concentration and the ammonia oxidation rate are not presented. Please include and reference these datasets.

Ammonia oxidation rates along with ammonium concentrations as well as urea oxidation rates along with urea concentrations and urea oxidation to total oxidation ratios are now presented in Supplementary Data 1. We now also refer to this in the text.

We thank the authors for including this data in Supplementary Data 1.

Line 130: Please define urea-derived ammonia oxidation and the urea oxidation notation in the introduction instead of the results.

We have made this change as suggested.

We thank the authors for making this change.

Line 132: The urea oxidation rates are not explicitly compared in Figure 2b. Please include and reference these comparisons. Consider adding comparison between urea concentration and urea oxidation rate as well.

It is true that the urea oxidation rates are not explicitly compared in Figure 2b, however, the difference in magnitude is visible by comparing the different panels. We now additionally mention this in the figure legend, and also refer to Supplementary Data 1, where urea oxidation rates along with urea concentrations are presented.

We thank the authors for adding this change to the figure legend and referring to Supplementary Data 1.

Line 133-141: Please present the absolute ammonia and urea oxidation rates in addition to the fractional rates presented in Figure 2a. The oligotrophic Angola Gyre has the lowest measured urea oxidation rates and ammonia concentrations presented in the study. Trends in the Global Ocean demonstrate that decreasing ammonia concentrations in the open ocean correlate with increased the urea: NH_4^+ concentration and oxidation ratios (Wan et al., 2024). The results

presented here further confirm these observations. However, an explicit comparison between concentration, rate, and ratios should be presented here.

The absolute rates can be found in Supplementary Data 1. We now also have an explicit comparison between *in situ* concentrations, rates and ratios in Supplementary Data 1.

We thank the authors for including these comparisons in Supplementary Data 1.

Line 147-148: The statement “as urea must ... via ammonia monooxygenase” should be moved to the introduction.

As suggested, this information is now in the introduction.

We thank the authors for this addition to the introduction.

Line 147: Urea is also broken down to carbon dioxide (CO₂), which can be assimilated. Please mention CO₂ while discussion urea hydrolysis

We now mention that urea is broken down into carbon dioxide in the introduction L59-60.

We thank the authors for describing the products of urea hydrolysis in L59-60.

Line 148-159: The addition of a large background NH₄⁺ to labeling incubation have previously been used to examine whether the ¹⁵N₂O – was produced via direct or indirect oxidation of the ¹⁵N labeled substrates (Kitzinger et al., 2019; Wan et al., 2024). Please update section to reflect the measurement of direct or indirect urea oxidation.

We have now referenced a new Supplementary Discussion paragraph here discussing that these experiments can also be used to examine the direct and indirect oxidation of ¹⁵N-labeled substrates.

We thank the authors for including this Supplementary Discussion about direct versus indirect urea oxidation.

Line 152-159: The axes for the Gulf of Mexico are an order of magnitude greater than the other sampling sites in Figure 2b making it difficult to make a direct comparison between the sampling sites. It is apparent that the addition of ammonia significantly decreased the urea oxidation rate in the Gulf of Mexico, while the tracer has no impact in the Black Sea and Angola Gyre.

However, the Black Sea and Angola Gyre urea oxidation rates are an order of magnitude lower than the presented Gulf of Mexico rates. The different responses may be attributed to the lower Nitrosopumilus (0.04 d⁻¹) and Nitrosopelagicus (0.002 d⁻¹) NanoSIMS-based growth rates in the Black Sea (line 229-236) compared to the Gulf of Mexico (0.23 d⁻¹). The faster growth rate may enable the Gulf of Mexico AOA population to transition their metabolism away from urea-oxidation over the 24-hour isotope incubations. The Black Sea and Angola Gyre AOA populations may also transition their metabolism towards ammonia-oxidation if the incubation times were corrected for the AOA growth rates. Consider discussing the impact of concentration and growth rates on direct versus indirect urea oxidation rates.

Please refer to major point 4, where we have addressed this comment in detail.

Response to the major point 4 comment has been addressed in more detail above.

Line 160: Please change “all known ammonia oxidizers” to “all known ammonia-oxidizing archaea”. All ammonia oxidizers encode an ammonia monooxygenase (AMO) and

hydroxylamine oxidoreductase (HAO) to oxidize ammonia to nitric oxide (Caranto and Lancaster, 2017). However, ammonia-oxidizing bacteria (AOB) and ammonia oxidizing archaea (AOA) have different responses to the nitric oxide (NO) scavenger PTIO, suggesting the mechanism converting nitric oxide (NO) to nitrite may differ between AOB and AOA (Martens-Habbenha et al., 2015).

We agree and apologize for the imprecise wording, which we have now changed accordingly in the revised manuscript.

We thank the authors for updating the phrasing. We are glad imprecise wording was caught in the review process and was updated.

Line 162-165: Please specify that only the Betaproteobacteria ammonia-oxidizing bacteria isolates demonstrated a clear preference for urea, while the marine Gammaproteobacteria AOB isolate did not demonstrate a clear preference for urea or ammonium (Qin et al., 2024). The Gammaproteobacteria are probably most relevant for this study.

We now specify that betaproteobacterial ammonia oxidizers show a preference for urea.

We thank the authors for adding this clarification about differences in urea utilization by AOB, since AOB urea niche differentiation is analogous to the AOA niche differentiation presented in this manuscript.

Line 165-167: Please indicate the increasing importance of urea in oligotrophic regions has been demonstrated several times before with ^{15}N tracer studies. These results further confirm the hypothesis in different marine environments.

We revised the entire manuscript to refer to other studies more frequently.

We thank the authors for referring to these studies more frequently.

Line 174: The referenced Extended Data figure appears to be incorrect. Extended Data Figure 3 is a phylogenetic tree and does not include information about AOA abundance. It appears the intended reference was Extended Data Figure 5.

We apologize for this oversight, neither Supplementary Figure 3 nor Supplementary Figure 5 depict that AOA are the main ammonia oxidizers, but it can be found in the Supplementary Data 1 and 3. We corrected this in the text.

We thank the authors for correcting the reference in the text.

Line 175-178: The difference in cell concentration appears to be driving the measurements presented in this manuscript. Consider presenting the ammonia-oxidizer cell abundance either in the main text, extended data, or supplemental material.

If we understand the reviewer correctly, they are asking for AOA cell numbers that are already given in this section and we now specifically mention differences in single cell activity between environments (L194). The cell numbers can also be found in Supplementary Data 1.

We would like to clarify our comment as we had intended to ask about how the differences between the AOA cell concentration contributes to the difference in total oxidation rates. The authors successfully addressed this commenting on the per cell oxidation rate (L194).

Line 179-181: Please discuss how bulk rates are dependent on cell concentrations and cell specific rates. This is important because the two order of magnitude difference might be attributed to an order of magnitude difference in cell concentration and an order of magnitude difference in the cell specific rate.

Thank you for this comment. We agree that the bulk oxidation rates are influenced by the abundance of AOA cells, and additionally, AOA in nutrient-richer environments may generally have higher cell specific rates compared to those in nutrient deplete environments. We have now addressed this within the manuscript (L194).

We thank the authors for pointing out that AOA in more nutrient-rich environments may have higher cell specific rates and for addressing this comment in L194.

Line 181-188: The contrasting abundance and pangenomics of *Nitrosopumilus* and *Nitrosopelagicus* in coastal estuaries and the open ocean has previously been reported (Qin et al., 2020; Santoro et al., 2017). The results presented in this section further confirm the previous reports of AOA biogeography and genomic inventory. Please reference previous studies.

We now reference Qin et al. 2020 ISMEJ as an example in the main text, as this study investigated marine AOA composition globally.

We thank the authors for referencing one of the suggested manuscripts.

Line 190-191: Consider mentioning the reason for specifically investigating the *dur3* urea transporter, since AOA encode three types of urea transporters (*dur3*-type, ATP dependent, and *yut*-type).

We have adjusted the text to clarify that we investigated *dur3* as it is the most widespread urea transporter in AOA.

We thank the authors for clarifying the reason for selecting *dur3* in the manuscript.

Line 197: Additional organisms contain ureases and urea transporters. Please include other organisms in the relative abundance comparisons in Figure 2c-d.

We now specifically mention that in the Gulf of Mexico (unlike the Angola Gyre and the Black Sea), most *ureC* reads are affiliated with non-AOA, who also dominate *ureC* transcription, and also early on refer to Supplementary Figure 5b. However, we have refrained from including this data in a main text figure, as we feel it is not a key point to this manuscript.

We thank the authors for mentioning the *ureC* reads in the Gulf of Mexico are affiliated with non-AOA in L208-213. We agree that figure updates are not needed after reading the response from the authors.

Line 197: The *ureC* RPKM values presented in Extended Data Figure 5b suggest the Gulf of Mexico AOA population must compete with Cyanobacteria and Proteobacteria for urea. The competition for urea may influence the indirect urea utilization presented in Figure 2b. The Angola Gyre and Black Sea AOA are the dominant urea hydrolyzers and presumably experience minimal urea competition. It is possible these AOA may co-consume ammonia and urea due to the slow growth rates measured with NanoSIMS. It may be worth mentioning that urea competition may influence the observed AOA ecophysiology.

The reviewer correctly points out that in the Gulf of Mexico, *ureC* is predominantly encoded and transcribed by microorganisms other than AOA. Additionally, overall *ureC* transcription is very low compared to the other sampled environments, which we interpret as urea simply not being a major N-source used by the microbial community in this ammonium-rich environment. We now address this in the revised manuscript (L288-291).

We thank the authors mentioning the *ureC* reads in the Gulf of Mexico are affiliated with non-AOA in L208-213.

Line 197: The ratio of urease (*ureC*) and the *dur3*-type urea transporter to the ammonia monooxygenase across AOA lineages is presented in lines 193-211, yet the data is in Extended Data Figure d-g. Consider moving this figure into the main text.

We have considered moving this figure to the main text, but find that it does not contribute enough to the main message of the manuscript to include it there.

We thank the authors for explaining why this figure was not included in the main text.

Line 212: The combined CARD-FISH and NanoSIMS dataset used to evaluate the nitrogen assimilation strategy across marine AOA lineages makes this manuscript unique and furthers our understanding of AOA ecophysiology. The manuscript should be centered around the single-cell N-assimilation measurements of AOA in contrasting marine environments, instead of presenting it at the end.

Thank you for this comment. We think it is important to first establish that only looking at the water mass as a whole (i.e. by measuring bulk oxidation rates and applying -omics approaches) does not help us in understanding the mechanisms of who is using urea. Only when combined with single cell measurements, the observed patterns start to make sense. We have restructured the abstract and the introduction, as well as made changes throughout the manuscript to better highlight the novelty of our study.

We thank the authors want to thank and commend the authors for the effort they put into restructuring this manuscript to better highlight the niche differentiation of *Nitrosopelagicus* and *Nitrosopumilus*. All these updates highlight the novelty of this study.

Line 236: Please maintain a consistent y-axis across microorganisms to ensure easy visual comparisons. It may be worthwhile to present the data on a log axis given the order of magnitude difference between the *Nitrosopelagicus* WCA and *Nitrosopumilus* 15N biomass enrichments in Figure 3a-b.

We have tried multiple options of data depiction for this particular plot and think that this best depicts the data. We find that a log scale plot is harder to read and thus obscures the main finding.

We thank the authors for taking this suggestion into consideration.

Line 236: Please improve contrast of the 15N biomass enrichments in Figure 3a-b to make it easier to distinguish single-cells from the filter background.

We agree that the contrast of the *Nitrosopelagicus* cells is not very high, however the contrast has already been improved by adding an alpha mask (see Methods). The enrichment of

Nitrosopelagicus cells is relatively close to the natural abundance and because we find it important to also show this visually, we opt for not manipulating the contrast any further.

We thank the authors for pointing out that the contrast was not adjusted to demonstrate the enrichment of *Nitrosopelagicus* cells is relatively close to the natural abundance. We agree that it is important to show this visually.

Line 240-252: The absolute ¹⁵N-urea biomass enrichment values in Figure 3 indicate *Nitrosopumilus* and other microorganisms assimilated more urea than *Nitrosopelagicus*. The higher enrichment suggests *Nitrosopumilus* and other microorganisms have higher cell-specific urea assimilation rates and will outcompete *Nitrosopelagicus* in the absence of ammonium. However, the dual substrate (urea + NH₄⁺) dataset suggests *Nitrosopumilus* and other microorganisms repress urea assimilation in the presence of ammonium while *Nitrosopelagicus* continued to assimilate urea at similar rates. The dataset suggests *Nitrosopelagicus* do not have a distinct metabolite preference and may co-consume urea and ammonium, similar to the marine Gammaproteobacteria AOB *Nitrosococcus oceanus* (Qin et al. 2024). Although an additional dual substrate incubation with ¹⁵N-NH₄⁺ would be required to confirm this hypothesis. Please comment on this shortfall.

We agree with the reviewer that *Nitrosopelagicus* appears to not have a distinct metabolite preference on the population level. The question of co-consumption of ammonium and urea by the same cell is interesting, but as pointed out, cannot be fully solved by our experiments. To achieve this would require comparison of single cell N-assimilation in additional experiments where both ¹⁵N-ammonium + ¹⁵N-urea are supplied, in comparison to single labeled experiments. In such experiments, co-consumption would lead to higher ¹⁵N enrichment of individual cells in ¹⁵N-ammonium + ¹⁵N-urea incubations in comparison to single labeled incubations.

In response to this comment, we have now changed the last sentence of this paragraph in the main text (former Line 256, now L288-291).

We thank the authors for updating L288-291 to mention *Nitrosopelagicus* does not appear to have a distinct metabolite preference at the population level.

Line 240-252: Consider comparing urea utilization under the dual substrate incubation across lineages. It appears that *Nitrosopelagicus* may have the greatest assimilation rate under the dual substrate condition. Therefore, *Nitrosopelagicus* may be the predominant urea-oxidizer in the presence of ammonia and urea, which could then explain the direct urea-oxidation measurements in Figure 2b.

It is true that when supplied with ¹⁵N-urea and ¹⁴N-ammonium *Nitrosopelagicus* is higher enriched in ¹⁵N/(¹⁴N+¹⁵N) compared to *Nitrosopumilus* and other microorganisms (see Figure A below). We also consider it likely that the consistent urea oxidation that we observe in the incubation amended with ¹⁴N-ammonium is performed by *Nitrosopelagicus*.

We thank the authors for including a comparison of the ¹⁵N/(¹⁴N+¹⁵N) enrichment across organisms in the ¹⁵N-urea and ¹⁴N-ammonium incubation. It was difficult to compare these enrichments in Fig 3c. I find this figure highly compelling. The significantly higher enrichment of *Nitrosopelagicus* indicates it is the predominant urea-oxidizer in the presence of ammonia and

urea, which could then explain the direct urea-oxidation measurements. It would be nice to see this figure in the manuscript. However, the decision should be at the discretion of the authors.

Figure A: Urea assimilation in the presence of excess ammonium by *Nitrosopelagicus*, *Nitrosopumilus* and other microorganisms. $^{15}\text{N}/(^{14}\text{N} + ^{15}\text{N})$ enrichment of *Nitrosopelagicus* WCA (magenta), *Nitrosopumilus* (blue) and other microorganisms (gray) after incubation with ^{15}N -urea + ^{14}N -ammonium. Boxplots depict the 25–75% quantile range, with the center line depicting the median (50% quantile); whiskers encompass data points within 1.5x the interquartile range. ^{15}N at% enrichment of the substrate pools (ammonium or urea, respectively) was >97%. Note that *Nitrosopumilus* and *Nitrosopelagicus* cells were targeted in separate CARD-FISH NanoSIMS analyses, and thus, the category “other microorganisms” may contain some (not-targeted) *Nitrosopelagicus* and *Nitrosopumilus* cells, respectively. n = number of measurements per group and treatment with number of cells enriched significantly above the detection limit in brackets. *** indicates significant difference between treatments in Mann-Whitney-Wilcoxon tests. *Nitrosopelagicus* was significantly higher enriched in comparison to *Nitrosopumilus* ($W = 452$, P value = 7.5×10^{-4}).

Line 256: Please remove the statement “thus essentially doubling its growth rate”. The data presented in this manuscript does not demonstrate the growth rate of *Nitrosopelagicus* was doubled via urea and ammonium co-consumption. Previous observations of co-consumption by *Nitrosococcus oceanus* did not report a statistical difference in the growth rate while consuming both ammonia and urea (Qin et al., 2024).

Although we do not know whether a single *Nitrosopelagicus* cell is using ammonium and urea simultaneously, the growth rate of the *Nitrosopelagicus* population could be doubled when both ammonium and urea are available. We changed this in the manuscript accordingly.

We thank the authors for adjusting L288-291 and L315 in the manuscript.

Line 258-260: This statement disregards the influence of metabolite concentrations. Please revise the statement.

We have considered revising this concluding paragraph, but as we discuss metabolite concentrations in detail in the main manuscript, we feel it is not needed to bring this up again at this point.

We agree with the response from the authors.

Line 263-264: The underestimation of marine nitrification by approaches only considering ammonia-oxidation has been reported before. Please differentiate this study by highlighting single-cell N-assimilation across AOA lineages.

We adjusted the wording accordingly (L305-307).

We thank the authors for adjusting L305-307

Line 276-277: Please remove the statement “allowing them to double their growth rates”. See comments about Line 256.

We rephrased this sentence. It now says “Furthermore, although we cannot unequivocally show that *Nitrosopelagicus* cells were assimilating ammonium and urea simultaneously, if they do, then this strategy could potentially lead to a doubling of their growth rate when both ammonium and urea are available”. Now L288-291.

We thank the authors for adjusting L288-291 and L315 in the manuscript.

Line 281-282: The demonstration of preferred urea utilization by *Nitrosopelagicus* and concluding urea utilization could differentiate the niche of marine AOA lineages is novel. This finding should be highlighted throughout the manuscript, including the introduction and abstract. Done as suggested.

We thank the authors for updating the abstract and introduction.

Line 288-290: Consider softening the statement “we hypothesize that *Nitrosopelagicus* obtains ammonium from other DON compounds”. The utilization of DON compounds by ammonia-oxidizers has been limited to urea, cyanate, and guanidine in isolates and in-situ utilization of glutamate has been demonstrated.

We have considered revising this statement, but taking into account that ammonia oxidizers have been shown to use urea, cyanate, polyamines and (as recently discovered) guanidine, we think that there might be additional DON compounds marine AOAs are using. As we explicitly state that we are hypothesising here, we have not adjusted this statement.

We agree with your explaining for not adjusting the statement.

Line 325-328: Explicitly mention analytical methods used to measure metabolites in manuscript. We now explicitly mention the used analytical methods, which are all well established standard protocols to measure urea, ammonium, nitrite and nitrate.

We thank the authors for including these well established standard protocols in L363 and L366.

Line 330-331: A brief overview of the experimental methods from the Gulf of Mexico cruise should be presented here.

We now briefly mention how incubations were done in the Gulf of Mexico cruise. However, we feel due to space constraints a more elaborate description of previously published protocols is not needed.

We thank the authors for including this brief description despite the space constraints.

Line 337: Different process rate methods were used for the Gulf of Mexico, Angola Gyre, and Black Sea. Please elaborate on differences between methods.

Please refer to our detailed explanation of how the setups differed and which differences we consider to have an impact in our answer to the reviewer's major points above. Briefly, all protocols followed established incubation protocols and were chosen for each cruise taking into consideration the *in situ* conditions (e.g. oxygen concentrations), the feasibility and also the compatibility with other projects/experiments that were run on the same cruise but are not presented here.

See response to major comment 3.

Line 338-343: Sampling depths, storage duration prior to experiment, and light exposure are not explicitly mentioned. Please consider including.

We now specifically mention storage time and light exposure in the methods section (L379-380). Sampling depths are already specified in Supplementary Data 1.

We thank the authors for including storage time and light exposure.

Line 353-355: State replication used in tracer experiment.

We have now added the replication in the methods. Per time point, one biological replicate was sacrificed.

We thank the authors for including the replication in the methods section.

Line 360-361: It is unclear how the water column oxygen concentrations correspond to the ambient oxygen concentrations in the Schott bottle. Please elaborate on choice.

In the Angola Gyre, ambient oxygen concentrations were $>60 \mu\text{M}$ at all incubation depths. Thus we chose to incubate in Schott bottles as no oxygen manipulations were required. We have added this to the text at L400-403.

We thank the authors for adding L400-403.

Line 363: Duplicate samples were collected for each condition. Please elaborate on using duplicate as opposed to triplicate samples for downstream statistical analysis.

In the Angola Gyre cruise, this was done due to time, water, and incubator space constraints, which precluded us from performing triplicate incubations. This information is now also included in the methods section.

We thank the authors for the addition of L404-405.

Line 370: Please mention the isotope supplementation concentration was chosen taking in situ concentrations into considering and wanting the overall ^{15}N at% to be greater than 90% in the main text.

We respectfully disagree that this is needed within the main text, however we have included the rationale in the methods and made a clearer reference to them as well as the Supplementary Data 1, where the measured labelling percentages are given.

We thank the authors for addressing this comment. As mentioned above, it was not clear to us if L411-414 was referring to the 3 sample sites in Kitzinger et al. 2019 or the sites in this study. See major comment 3.

Line 373-388: Please include all rate calculation equations in main text or supplement to this manuscript.

As suggested, we have now included all rate calculations in the Supplementary Methods of the manuscript.

We thank the authors for including rate calculations in the Supplementary Methods.

Line 392-397: Please elaborate on the use of different methods to quantify initial ^{15}N -urea on the Black Sea and Angola cruise.

We are confident that both methods for determining labeling percentage of the substrate pool are well suited to yield robust and accurate results for our experimental setups. In the Black Sea, *in situ* urea concentrations and urea tracer additions were sufficiently high to confidently quantify with spectrophotometric methods. For the Angola Gyre, the tracer additions were lower, and thus their quantification via the more sensitive IRMS-based approach was better suited.

We thank the authors for commenting on the methodological differences.

Line 397: Consider a paragraph break to separate the statistical analysis from the methods. Done as suggested.

We thank the authors for implementing this suggestion.

Line 409-412: Setting the non-significant slopes to the limit of detection could lead to an overestimate of the measured rate, especially if the true rate is zero. Consider elaborating on this choice.

We find it unlikely that when one biological replicate shows a significant rate, the true rate in a replicate incubation is zero. Instead, there appears to be some biological variability which leads to some replicates falling below the detection limit in the Angola Gyre samples, especially as all rates measured there are very low. Given that our detection limits are close to zero (0.017, 0.040, 0.146 nM day⁻¹), and that we observe some biological variability, we think that averaging by considering non-significant rates as detection limit (which we opt for), gives us a more realistic averaged rate in comparison to setting non-significant rates to zero. However, we would also like to emphasize that if both replicate rates were non-significant we considered the rate to be zero.

We thank the authors for explaining how the biological variability contributes to some rates falling below the limit of detection. I agree with your response. The addition of L457-458 add clarity to the method.

Line 443: The contig assembly and read mapping methods are not explicitly stated here. Please update this section to include package and settings as done in ref. 54. It is implied the metagenomic and metatranscriptomic abundance was evaluated with a read-based method

instead of a MAG-based method. However, the method for calculation abundance should be explicitly mentioned.

We updated this section and now explicitly list the versions for the programs used, all further details regarding the scripts used are found on github (https://github.com/dspeth/aoa_urea/tree/main/marker_gene_databases), which we also reference in the text. Additionally, we clarified the calculation of the abundance (RPKM) calculation. We would like to clarify that for assessing abundance of the key genes (*amoA*, *ureC*, *dur3*) pertaining to AOA, we did not use any assembled metagenomic data, but directly used the trimmed and quality filtered reads. Metagenome assemblies were only used to obtain longer sequences of key genes to calculate phylogenetic trees, and bin metagenome assembled genomes. This latter procedure is detailed in the Supplementary Methods.

Thank you for including program versions and including the relevant scripts in the github repository. The manuscript included: 1) AOA abundance was assessed with metagenomic reads (L485); 2) the RPKM abundance calculation (Equation 1) in the previous draft; and 3) metagenomic assemblies were only used for phylogenetic analysis (L557-560). We would like to acknowledge this oversight in our previous review.

Line 444-446: It is not clear how marker gene abundance was assessed until line 477. The database curation method presented in lines 447-476 is important, but could be moved to supplement and referenced in the main text.

As suggested, we moved the method section on database curation to the Supplementary Methods and referenced this in the main text.

We thank the authors for moving the database curation to the Supplementary methods

Line 480: Please define BSR as bit-score ratio when it is first introduced on line 480. The BSR approach is introduced the first time it is mentioned as “BLAST score ratio (BSR) approach” (L526, previously L458).

We thank the authors for catching my mistake and defining BSR in L526.

Line 595: Please include equations used to calculate cellular ^{15}N -atom% excess and growth rate in the main text or supplement of this manuscript.

We now include the equations in the Supplementary Information of the manuscript.

We thank the authors for including these equations in the Supplementary Information. This will (hopefully) make it easier for others to replicate your calculations.

References

- Arandia-Gorostidi, N., A. L. Jaffe, A. E. Parada, B. J. Kapili, K. L. Casciotti, R. S. R. Salcedo, C. M. J. Baumas and A. E. Dekas (2024). "Urea assimilation and oxidation support activity of phylogenetically diverse microbial communities of the dark ocean." The ISME Journal **18**(1).
- Bristow, L. A., T. Dalsgaard, L. Tiano, D. B. Mills, A. D. Bertagnolli, J. J. Wright, S. J. Hallam, O. Ulloa, D. E. Canfield, N. P. Revsbech and B. Thamdrup (2016). "Ammonium and nitrite oxidation at nanomolar oxygen concentrations in oxygen minimum zone waters." Proc Natl Acad Sci U S A **113**(38): 10601-10606.
- De Brabandere, L., B. Thamdrup, N. P. Revsbech and R. Foadi (2012). "A critical assessment of the occurrence and extend of oxygen contamination during anaerobic incubations utilizing commercially available vials." Journal of Microbiological Methods **88**(1): 147-154.
- Kitzinger, K., C. C. Padilla, H. K. Marchant, P. F. Hach, C. W. Herbold, A. T. Kidane, M. Konneke, S. Littmann, M. Mooshammer, J. Niggemann, S. Petrov, A. Richter, F. J. Stewart, M. Wagner, M. M. M. Kuypers and L. A. Bristow (2019). "Cyanate and urea are substrates for nitrification by Thaumarchaeota in the marine environment." Nat Microbiol **4**(2): 234-243.
- Qin, W., S. P. Wei, Y. Zheng, E. Choi, X. Li, J. Johnston, X. Wan, B. Abrahamson, Z. Flinkstrom, B. Wang, H. Li, L. Hou, Q. Tao, W. W. Chlouber, X. Sun, M. Wells, L. Ngo, K. A. Hunt, H. Urakawa, X. Tao, D. Wang, X. Yan, D. Wang, C. Pan, P. K. Weber, J. Jiang, J. Zhou, Y. Zhang, D. A. Stahl, B. B. Ward, X. Mayali, W. Martens-Habbena and M.-K. H. Winkler (2024). "Ammonia-oxidizing bacteria and archaea exhibit differential nitrogen source preferences." Nature Microbiology.
- Qin, W., Y. Zheng, F. Zhao, Y. Wang, H. Urakawa, W. Martens-Habbena, H. Liu, X. Huang, X. Zhang, T. Nakagawa, D. R. Mende, A. Bollmann, B. Wang, Y. Zhang, S. A. Amin, J. L. Nielsen, K. Mori, R. Takahashi, E. Virginia Armbrust, M. H. Winkler, E. F. DeLong, M. Li, P. H. Lee, J. Zhou, C. Zhang, T. Zhang, D. A. Stahl and A. E. Ingalls (2020). "Alternative strategies of nutrient acquisition and energy conservation map to the biogeography of marine ammonia-oxidizing archaea." ISME J **14**(10): 2595-2609.
- Santoro, A. E., M. A. Saito, T. J. Goepfert, C. H. Lamborg, C. L. Dupont and G. R. Ditullio (2017). "Thaumarchaeal ecotype distributions across the equatorial Pacific Ocean and their potential roles in nitrification and sinking flux attenuation." Limnology and Oceanography **62**(5): 1984-2003.
- Wan, X. S., H. X. Sheng, H. Shen, W. Zou, J. M. Tang, W. Qin, M. Dai, S. J. Kao and B. B. Ward (2024). "Significance of Urea in Sustaining Nitrite Production by Ammonia Oxidizers in the Oligotrophic Ocean." Global Biogeochemical Cycles **38**(10).

Xu, M. N., X. Li, D. Shi, Y. Zhang, M. Dai, T. Huang, P. M. Glibert and S. J. Kao (2019).
"Coupled effect of substrate and light on assimilation and oxidation of regenerated nitrogen in
the euphotic ocean." Limnology and Oceanography **64**(3): 1270-1283.

Zhao, Z., C. Amano, T. Reinthaler, F. Baltar, M. V. Orellana and G. J. Herndl (2024).
"Metaproteomic analysis decodes trophic interactions of microorganisms in the dark ocean."
Nature Communications **15**(1).